# Complement inhibition by a unique cluster of immunomodulatory outer surface proteins of *Borrelia recurrentis*

Florian Röttgerding[1,5], Flavia Reyer[1,5], Eva Gerlach [1], Martin Amborn[1], Nadine Duschek[1], Tilman Gunter Schultze[1], Volker Fingerle [2], Christopher M. Roome[3], Michaela Stumpf[4], Katja Becker[4], Stefan Rahlfs[4], Jude Marek Przyborski [4], Peter Kraiczy [1,6] ✉ & Karin Fritz-Wolf [3,4,6] ✉

*Borrelia recurrentis*, the agent of louse-borne relapsing fever, causes a poverty-associated, infectious disease of high mortality. Here, we identified and characterized five Complement targeting and Host Interacting proteins, ChiA to ChiE displaying immunomodulatory functions. Almost all Chi homologs inhibit complement activation by direct binding of key components, block membrane attack complex formation, and interact with plasminogen. *Borrelia* proteins protect susceptible spirochetes from complement-mediated killing and ChiB, ChiC, and ChiD facilitate serum resistance. X-ray structures of ChiA and ChiB, along with AlphaFold models of ChiC, ChiD, and ChiE, reveal a conserved, compact eight-helix fold with a central hydrophobic pocket and a unique S-domain-feature distinct from all known *Borrelia* proteins. Notably, ChiC and ChiE harbor conserved cysteines forming a reversible disulfide bridge, indicating redox-responsive function. Our findings identify a protein family of functionally related immune evasion factors, advancing our understanding of the underlying mechanisms of complement resistance in this neglected, human pathogenic microorganism.

Louse-borne relapsing fever (LBRF), a neglected vector-borne disease, is solely caused by *Borrelia (B.) recurrentis*[1]. Historically responsible for deadly epidemics, LBRF remains endemic in countries to the Horn of Africa and re-emerged during the refugee crisis in 2015–2016[2–4], raising concern for outbreaks under poor and overcrowded living conditions. Clinically, LBRF manifests with recurrent febrile episodes, systemic inflammation, and severe complications[1–4].

*Borrelia recurrentis* is transmitted by inoculation of spirochete-containing coelomic fluid or feces from crushed lice into skin microabrasions[1,3,4]. Immediately after entering the human host, spirochetes encounter complement as the first line of immune defense[5,6]. To overcome innate and adaptive immunity, *B. recurrentis* has developed at least two major strategies[7,8]: (i) antigenic variation of variable major proteins generating successive waves of antigenically distinct subpopulations under immune pressure, and (ii) complement evasion through recruitment of host-derived regulators such as factor H (FH), C4b-binding protein (C4BP), and C1 esterase inhibitor (C1-Inh), respectively[8]. While antigenic variation allows escape from adaptive immune responses, complement inactivation may significantly support *Borrelia* survival in the bloodstream and systemic spread in the human host.

Complement represents a potent barrier against invasion of pathogenic microorganisms[9]. This system is activated via three

[1]Goethe University Frankfurt, University Hospital, Institute for Medical Microbiology and Infection Control, Frankfurt, Germany. [2]National Reference Center for Borrelia, Bavarian Health and Food Safety Authority, Oberschleissheim, Germany. [3]Max-Planck-Institute for Medical Research, Heidelberg, Germany. [4]Biochemistry and Molecular Biology, Interdisciplinary Research Centre, Justus Liebig University Giessen, Giessen, Germany. [5]These authors contributed equally: Florian Röttgerding, Flavia Reyer. [6]These authors jointly supervised this work: Peter Kraiczy, Karin Fritz-Wolf. ✉e-mail: kraiczy@em.uni-frankfurt.de; karin.fritz-wolf@ernaehrung.uni-giessen.de

pathways: classical (CP), lectin (LP), and alternative (AP) (Fig. S1)[10]. After initiation of the CP by C1q binding to immunoglobulins or surface structures, by carbohydrate recognition (LP) or by spontaneous C3 activation (AP), the C3 convertases C4b2b (CP/LP) and C3bBb (AP) are formed. Subsequent cleavage of C3 into C3b and C3a boost opsonisation of microbes and formation of the C5 convertases C4b2b3b (CP/LP) and C3bBb3b (AP). Upon cleavage of C5, C5b bound to the microbial surface and initiates activation of the terminal pathway (TP) by recruiting C6-C9, forming the pore-forming membrane attack C5b-9 complex or MAC leading to bacterial lysis[10]. To prevent excessive activation, complement is strongly controlled by C1-INH (CP), FH and FHL-1 (AP), and C4BP (CP and LP), vitronectin (Vn) clusterin, and FHR-1 (TP) (Fig. S1). FH, FHL-1, and C4BP, respectively, act as cofactors for factor I (FI), which inactivates C3b and C4b, thereby limiting C3 convertase formation. Assembly of the MAC is terminated by vitronectin, clusterin, and FHR-1, respectively.

Structurally diverse surface-exposed lipoproteins of Lyme disease (LD) and relapsing fever (RF) *borreliae* contribute to complement inhibition[11–13]. Regarding LD spirochetes, the FH and FHL-1 binding proteins CspA and CspZ have been identified as the key complement inhibitors of the AP whereas CspA also blocks TP activation by interaction with C7, C8, C9, and the MAC, respectively[11,14]. Factors involved in complement evasion of RF spirochetes comprises CihC, the FH-binding proteins HcpA, BhCRASP-1 or FhbA, the fibronectin-binding proteins FbpA, FbpB, and FbpC as well as CbiA[8]. CihC and HcpA, both of which exhibit anti-complement properties and potentially contributing to the pathogenesis of *B. recurrentis* protect spirochetes from complement-mediated killing by binding of C1-Inh, C4BP (via CihC), and FH (via HcpA) to terminate CP and AP activation[15,16]. Recently, a novel mode of CP inactivation targeting the formation of the initial C1 complex has been described for diverse RF borreliae and involves the interaction of C1r with CihC, FbpA, FbpB, and FbpC, respectively[17–19]. Thus, recruitment of diverse host regulators represents an ingenious immune evasion strategy of this particular pathogen. Moreover, it has been shown that recruitment of plasminogen and activation to plasmin by urokinase-type activator (uPA) enhances brain and heart invasion of LD and RF borreliae in the murine host[20–22].

In contrast to lipopolysaccharides of Gram-negative bacteria, lipoproteins form a peculiar feature of LD and RF *borreliae* and often serve as serious virulence factors contributing to transmission, adhesion, immune evasion, and persistence[23]. These lipoproteins are characterized by their structural and functional domains consisting of an intrinsically disordered N-terminus ("tether") that harbors the signal peptide, a so-called "lipobox", a conserved cysteine residue for diacylation and a sorting signal. The highly flexible tether links the N-terminal lipid anchor to the rest of the protein that execute the specific functional fold.

Here, we identified a unique gene cluster on megaplasmid lp190 adjacent to the tandemly arranged *chiC* and *hcpA* genes[24,25]. This gene cluster encodes for five distinct outer surface lipoproteins designated as ChiA to ChiE (Complement-targeting and Host-interacting proteins). Our data revealed an unexpected, multifactorial binding profile to certain complement components and plasminogen, thereby impacting complement activation to various degrees. All proteins share a conserved fold comprising a compact α-helical main domain and a distinct S-domain. The structures of ChiA (2.7 Å) and ChiB (1.5 Å), complemented by AlphaFold2 predictions, reveal a central hydrophobic pocket occupied by a phospholipid in ChiB, and a Chi-specific S-domain architecture not observed in other borrelial proteins. Collectively, our data indicate multiple complement-inhibitory activities of these molecules, redox-sensitive cysteine motifs (ChiC, ChiE), and a putative catalytic triad (ChiE).

## Results

### Identification of a unique gene cluster on lp190 of *Borrelia recurrentis*

Previous investigations identified two complement-targeting proteins, HcpA and CihC, both of which are encoded by genes located on the linear, ~190 kbp megaplasmid of *B. recurrentis*[15,16] (Fig. S2A). Our bioinformatic analyses discloses a unique cluster of five homologous genes located downstream to *cihC* and *hcpA* (Fig. S2A). The architecture of the gene cluster and adjacent genes was found to be highly conserved and exhibit significant synteny among *B. recurrentis* strains isolated at different geographic regions between 1990 and 2015[24,26] from relapsing fever (RF) patients but differed from *Borrelia duttonii*, the most closely related RF *Borrelia*[27] (Fig. S2A). These homologs share sequence identities between 33.7% and 56.5% and similarities between 60.1% and 76.1% (Table S1). Sequence analyses revealed high sequence identities/similarities between these proteins and their corresponding orthologs from *B. duttonii* Ly (93.6 to 99.6%) (Fig. S3). Neither CihC nor HcpA exhibit sequence similarities to these proteins and, thus, form two separate phylogenic lineages with other genes located on lp190 (Fig. S2B, Table S1).

### Structure determination of ChiA and ChiB

Initially, we determine the crystal structure of ChiA and ChiB as representative members of the Chi cluster. Both proteins crystallized in a monoclinic (C2) and a tetragonal (P4₂2₁2) space group, with two and one monomer(s) per asymmetric unit, respectively. Native crystals diffracted to 2.7 Å (ChiA) and 1.5 Å (ChiB). Given the low sequence identity to known structures (< 20%) and the lack of AlphaFold2 models, ChiB was solved by molecular replacement and anomalous dispersion; ChiA was resolved using ChiB as a template. Unexpectedly, the initial Fo−Fc electron density map of ChiB showed a well-defined density for a phospholipid. Final models include residues E24-Q284 for ChiB and D29-I265 for ChiA, with data collection and refinement statistics summarized in Table S2.

ChiB (31.8 kDa, 284 residues) consists of eight α-helices and one β-hairpin (P97–L107) forming a compact main domain (α2–α8) and a surface-exposed domain that we refer to as the S-domain. The latter comprises the β-hairpin and the N-terminal portion of helix α4 (A106-K122) (Fig. 1A). A central hydrophobic pocket (-22 Å × 11.5 Å × 8.5 Å), surrounded by hydrophobic and positively charged residues (e.g., R214, R226), accommodates a phospholipid (Fig. 1B). The buried cysteine C266 extends into the pocket, while the phospholipid is hydrogen-bonded by residues R214 and R226.

ChiA shares 38.0% sequence identity with ChiB (RMSD 2 Å over 157 residues, Table S1), adopting a similar fold. However, the α4 helix is shorter, the β-hairpin is rotated, and loop regions between α2–α3 and α5–α6 differ (Fig. 1A, C; Fig. S4). The α3 helix is reduced to one turn in ChiA (L76-I80), and the insertion V150-L155 seen in ChiB is absent. These changes affect the topology of the S-domain. Also, ChiA shows a more negative surface charge compared to ChiB (Fig. S5). Although ChiA possesses a similar hydrophobic pocket, no lipid could be detected. Pocket-lining residues (e.g., Y156 in ChiA vs. Y177 in ChiB) are largely conserved, with key differences at the entrance (e.g., E162/W255 in ChiB vs. R141/G237 in ChiA) (Fig. 1A, C). ChiA contains two cysteines: C115, oriented into the pocket, and surface-exposed C260, which forms a crystallographic disulfide bond with a symmetry-related monomer (C260 and C260'). Additionally, ChiA coordinates a magnesium ion via N69/E75 along with their counterparts N69" and E75" from another symmetry-related monomer (Fig. 1C, D).

### Structural comparison of Chi structures reveals distinctive features

We determined ChiA and ChiB structures by X-ray crystallography and generated AlphaFold2 (AF2) models for all five Chi homologs. The AF2 models align well with X-ray data (ChiA: RMSD = 0.9 Å, 224 atoms;

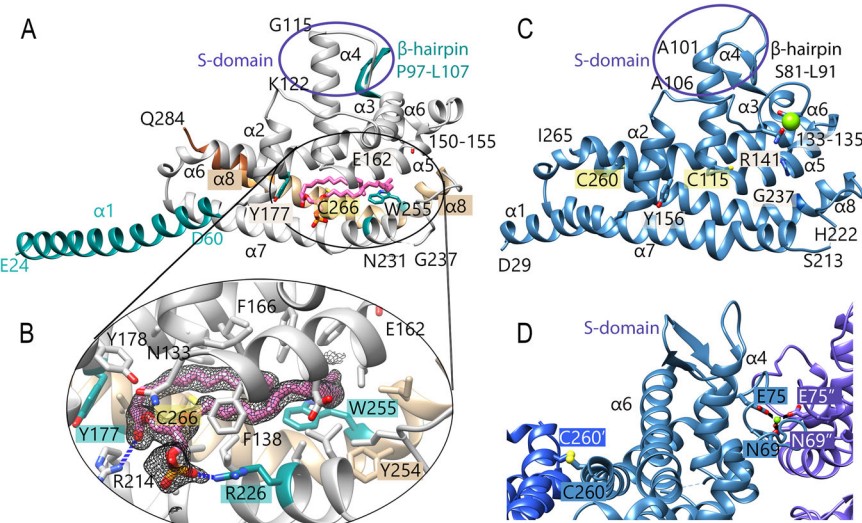

**Fig. 1 | X-ray structures of ChiB and ChiA. A** Overall fold of ChiB with the S-domain indicated by a violet ellipse. The hydrophobic pocket is occupied by a bound phospholipid (magenta sticks), and the insertion region (residues 150–155) is labeled. Mutated residues are highlighted in teal. C-terminal deletion variants that could not be produced in *E. coli* are shown in brown shades: variant 1 (Δ248-284, tan), variant 2 (Δ273-284, sandy brown), and variant 3 (Δ277-284, sienna). **B** Close-up of the ChiB hydrophobic pocket. All residues within 5 Å of the phospholipid are shown; selected residues are labeled for clarity. Hydrogen bonds between the phospholipid (LPP) and pocket residues are shown as blue dashed lines. The electron density (polder map, contoured at 3σ) is depicted in black. **C** Overall structure of ChiA (steel blue), with a Mg²⁺ ion represented as a green sphere. The region corresponding to the ChiB insertion is labeled (residues 133–135). **D** Crystal contacts of ChiA. A disulfide bond between monomer A (steel blue) and a symmetry-related monomer A′ (blue) is shown in ball-and-stick representation. The Mg²⁺ ion is coordinated by N69 and E75 from monomer A and by N69 and E75 from symmetry-related monomer A (violet).

ChiB: RMSD = 0.7 Å, 233 atoms), with minor differences at the α7–α8 junction and N-terminal helix α1 in ChiB (Fig. S6A). Per-residue confidence (pLDDT) exceeds 90 across most regions of the models and is reduced only at the α7–α8 junction and N-terminal regions (Fig. S6B). Together, the strong structural agreement and high model confidence support the use of AF2 models for structural analysis of the remaining homologs.

Superimposing ChiA-ChiE revealed a shared architecture with a helical core (α1–α3, α4–α8) and an extended S-domain formed by a β-hairpin and the N-terminal segment of α4 (Fig. 2A). A conserved hydrophobic pocket in the main domain is flanked by positively charged residues (K118/R197 in ChiA, R214/R226 in ChiB, R206/K218 in ChiC, R211/R223 in ChiD, and K197/K201 in ChiE), which could interact with the negatively charged phosphate group of a lipid, as observed in ChiB (Figs. 1 and S4).

Structurally, Chi proteins can be divided into two groups based on the arrangement of the S-domain (α4, β-hairpin) and the connections between the α2–α3 and α5–α6 helices (Fig. 2B). Group 1 comprises ChiB, ChiC, and ChiD, while group 2 consists of ChiA and ChiE. The S-domain is more pronounced in group 1 because the α4 helix is one turn longer and, in contrast to group 2, the orientation of the beta hairpin is maintained. Furthermore, the α3 helix in group 2 comprises only one turn and has a deletion in the connection between the α5 and α6 helices, as already observed in the comparison between ChiA (group 2) and ChiB (group 1) (Fig. S5).

## Functional properties and redox regulation of Chi proteins
The region between α7 and α8 varies considerably among Chi proteins. In ChiA and ChiB, it appears disordered in the crystal structures, indicating high flexibility, and is also predicted with low confidence by AlphaFold2. ChiE possesses a markedly shorter linker (seven residues) compared to ~30 residues in the other homologs (Figs. 2A, 3A; Fig. S4), forming a distinct surface pocket that provides more open access to the hydrophobic pocket (Fig. 3B).

In ChiC and ChiE, four residues—C165, C260, N257, and D261 in ChiC; C156, C227, H224, and D228 in ChiE—form a

cysteine–histidine–aspartate constellation reminiscent of a catalytic triad, located within a hydrophobic pocket (Fig. 3C, D). The two cysteines (C260/C227) are positioned in close proximity, potentially allowing disulfide bond formation.

To assess redox reactivity, Ellman's assay was performed on oxidized and reduced Chi proteins (Table S3). In ChiA, one reactive thiol was detected, consistent with a surface-exposed cysteine forming a disulfide with a symmetry-related monomer. No free thiols were detected in ChiB, in line with its single cysteine being buried in a hydrophobic pocket. ChiD showed limited thiol reactivity, likely due to partial shielding of its cysteine by K166 and N265. In ChiC, one accessible thiol was observed in the oxidized state, corresponding to the surface-exposed C165, whereas in the reduced form only 1.5 thiols were detected instead of the expected three. This suggests that access to the buried cysteines is restricted by a flexible loop covering the pocket entrance (Fig. 3A). In contrast, ChiE showed two reactive thiols in the reduced form and 0.5 in the oxidized form, consistent with reversible redox switching of C156 and C227 within the hydrophobic pocket. Together, these findings indicate that ChiC and ChiE exhibit redox-responsive cysteine pairs that may undergo reversible disulfide formation.

## Structural comparison with surface proteins of Lyme disease borreliae
Due to the low sequence identity of Chi proteins to known bacterial structures, we performed a structural similarity search using the DALI server. The top matches for all five Chi homologs indicate outer surface proteins from *B. burgdorferi*, including BBE31 (PDB: 6fze)[28], BBA66 (2yn7)[29], BBA73 (4axz)[30], and the BBA64 ortholog from *B. spielmanii* (6hpn)[30], with 13 to 19% sequence identity, Z-scores of 13 to 15, and RMSDs of 3.2–4.2 Å over ~210 residues. All other hits had Z-scores below 10, including the FH-binding protein CspA (PDB: 1w33, 4bl4)[31,32].

InterPro classification[33] assigned all Chi homologs to the PFam54_60 paralogous family[30,34], prompting further comparison to structurally characterized PFam54_60 members (Fig. 4). Structural alignment of ChiB with BBE31, BBA64, BBA66, BBA68 (CspA), BBA69,

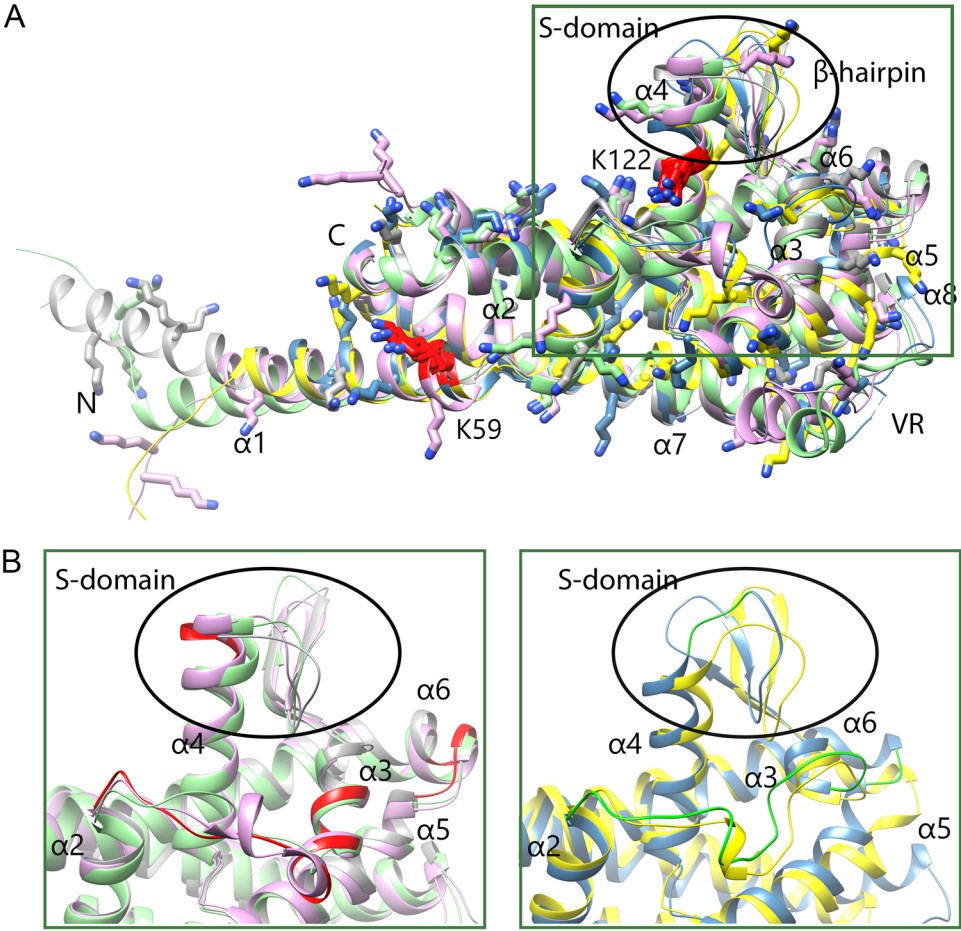

**Fig. 2 | Structural comparison of Chi proteins. A** Superposition of the X-ray structures of ChiA (steel blue) and ChiB (gray) with AlphaFold2-predicted models of ChiC (purple), ChiD (green), and ChiE (yellow). N- and C-termini are labeled N and C, respectively. The variable region between helices α7 and α8 is indicated as VR. All lysine residues are shown as sticks; strictly conserved lysines are highlighted in red, with K59 and K122 of ChiB labeled. A green rectangle marks the region enlarged in (**B**). **B** Close-up view comparing Chi subgroups: Group 1 (ChiB, ChiC, ChiD; left) and Group 2 (ChiA, ChiE; right). Structural differences between groups are highlighted in red (ChiB) and light green (ChiA).

BBA71, and BBA73 using Chimera's MatchMaker tool[35] revealed RMSD values ranging from 4.2 Å (6fze, 203 residues) to 15.3 Å (4bl4, 179 residues). Based on the DALI and MatchMaker results, the compared Pfam54_60 proteins were classified into two groups according to structural similarity: group A (BBE31, BBA64, and BBA66) and group B (BBA68, BBA69, and BBA73) (Figs. 4 and S7A). By using BlastP, these PFam54_60 paralogous proteins did not show significant similarities to the five Chi proteins.

### Assessment of the inhibitory capacity of Chi proteins on complement activation and MAC assembly

To assess the complement inhibitory capacity of the Chi proteins, an ELISA-based approach was conducted. Initially, microtiter plates were immobilized with specific compounds allowing a targeted activation of the respective pathway. After application of the reaction mixtures consisting of NHS pre-incubated with the analyzed protein, formation of the MAC was detected by a neoepitope-specific C5b-9 antibody. Complement inactivation was indicated by low absorbance values. All homologs significantly impaired AP activation (Fig. S8A) whereas ChiE also affected the CP and LP (Fig. S8B, C). The strongest inhibitory effect on the AP was observed for ChiB, ChiD, and ChiE at a final concentration of 1 and 2 μM, respectively (Fig. 5A). Likewise, ChiA and ChiC also inhibited the AP but only at the highest concentration (4 μM) employed (Fig. 5A). ChiE showed a dose-dependent inhibition of the CP and LP at 2 μM and 4 μM, respectively (Fig. 5B, C).

To further analyze TP inactivation, a cell-based hemolytical assay was employed (Fig. 6). After preparation of C5b-6 sensitized sheep erythrocytes, proteins pre-treated with purified C7, C8, and C9 were added. Lysis of erythrocytes was indicated by the release of hemoglobin due to the insertion of the formed MAC. As depicted in Fig. 6A, ChiB, ChiC, ChiD, and ChiE affected the TP even at the lowest concentration (final concentration: 0.5 μM) applied. The strongest inhibition among all proteins and similar to vitronectin, a natural inhibitor of the MAC, was observed for CihC from *B. recurrentis* while CspA from *B. burgdorferi*, a well-characterized inhibitory protein of the TP[11], displayed a weaker inhibitory capacity. Notably, binding to C9 leads to an incomplete MAC assembly[11–13]. Investigating ChiC, ChiD, and ChiE, all proteins strongly impaired C9 polymerization in a dose-dependent fashion of up to a final concentration of 0.2 μM, similar to CspA. In contrast, ChiA did not affect C9 polymerization (Fig. 6B), while ChiB displayed an inhibition only at a final concentration of 7.5 μM (Fig. 6C). Collectively, our findings point towards a differentiated inhibitory capacity of Chi homologs on complement that target certain activation levels including MAC assembly inhibition.

### Interaction of Chi proteins with complement components

To elucidate the molecular mechanism(s) of complement inhibition, interaction of Chi proteins with selected complement components of the AP [C3b, factor B (FB)] and the CP (C1q, C1r, C4, and C4b) as well as C5, FH, and FI was investigated. As demonstrated in Fig. S9A, all Chi

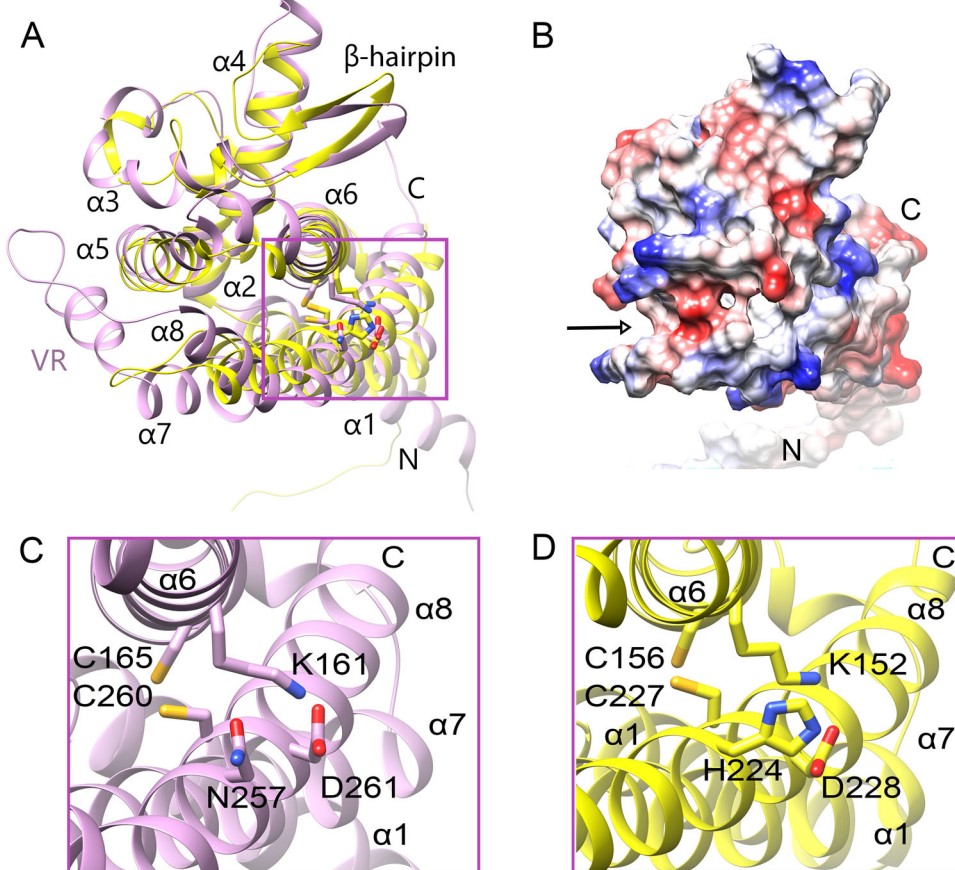

**Fig. 3 | Putative redox-regulated sites of ChiC and ChiE. A** Structural superimposition of ChiC (purple) and ChiE (yellow), viewed from the opposite side compared to Fig. 2A. The variable region (VR) in ChiC, corresponding to a large insertion, is highlighted. The magenta rectangle indicates the conserved cysteine region shown in panels (**C**) and (**D**). **B** Surface electrostatic representation of ChiE (Coulombic potential: red = −10, blue = +10), illustrating the surface pocket created by a deletion between helices α7 and α8 (arrow). **C**, **D** Close-up views of the putative redox-active site in ChiC (**C**) and ChiE (**D**), showing the conserved cysteines and nearby residues forming a protease-like motif.

proteins bound C3b but lower absorbance values were obtained employing ChiA. When binding to FB was assayed, all Chi homologs exhibited very low absorbance values (~0.25), even if statistically significant for ChiB and ChiE (Fig. S9B). In comparison to the C5-binding CbiA protein from *B. miyamotoi*[12] used as a positive control, only ChiB appears to bind C5 to some extent. Furthermore, the inhibitory capacity of ChiE on the CP and LP suggests an interaction with components primarily involved in these pathways. As shown in Fig. S9D–G, ChiE significantly interacted with C1q, C4, and C4b but not with C1r. Our previous studies on CbiA of *B. miyamotoi* HT31 demonstrated binding to FH and FI[12], but none of the Chi proteins interacted with these key components participating in AP regulation (Fig. S9H, I).

Next, the interaction of selected Chi proteins with various components was analyzed in more detail. A dose-dependent binding could be observed for all complement components but the strongest interaction was observed for C3b and ChiB, ChiC, and ChiD, respectively (Fig. S10A–E). A concentration-dependent binding to ChiE could be observed for C1q, C4, C4b, and C5, however, a saturation could not be achieved even at the highest concentration applied (Fig. S10F–K).

As all Chi homologs bound to C3b (Fig. S10A–E), we sought to test whether these proteins exhibit an intrinsic protease activity toward C3b and C3. After incubation with Chi proteins, the degradation pattern of both components was visualized by silver staining (Fig. S11). By monitoring for the appearance of the ~9 kDa C3a fragment, no cleavage products could be detected, suggesting that Chi homologs did not exert C3b/C3 cleavage activity under the tested conditions.

## Mapping of the complement-interacting region within ChiB

To identify complement-interacting regions, we generated ChiB variants carrying targeted deletions or single amino acid substitutions (Figs. 1A, S12A). Residues were selected based on Chi-specific structural features, including the N-terminal helix, the distinct S-domain, residues forming the hydrophobic pocket (Y177, R226, W255), and the C-terminal helix possibly involved in dimerization. Larger deletions removed the N-terminal helix A (aa 1–58), the β-hairpin (aa 97–107), and the C-terminal helix E (Δ248-284, Δ273-284, Δ277-284) to assess their contribution to ligand binding or dimerization. Despite extensive expression trials using different vectors, host strains, and conditions, none of the C-terminal deletion variants could be obtained in soluble form. None of the tested variants including deletion of the protruding N-terminus, the β-hairpin, or the point substitutions showed a measurable effect on complement inhibition (Fig. S12A–D). Only the Y177A substitution slightly reduced inhibitory activity, and no differences in C3b/C5 binding or C9 polymerization were observed (Fig. S12E–G). These findings indicate that several regions, or residues at the C-terminus, may jointly contribute to complement interaction.

## Interaction of Chi proteins with plasminogen

Binding of plasminogen enables spirochetes to penetrate into deeper tissues by acquiring the proteolytic activity of this host-derived protease[12,15,18,36–38]. As depicted in Fig. 7A, binding of plasminogen to all Chi proteins as well as HcpA of *B. recurrentis*[15] could be demonstrated but not to Vsp1 of *B. miyamotoi*, previously shown to lack

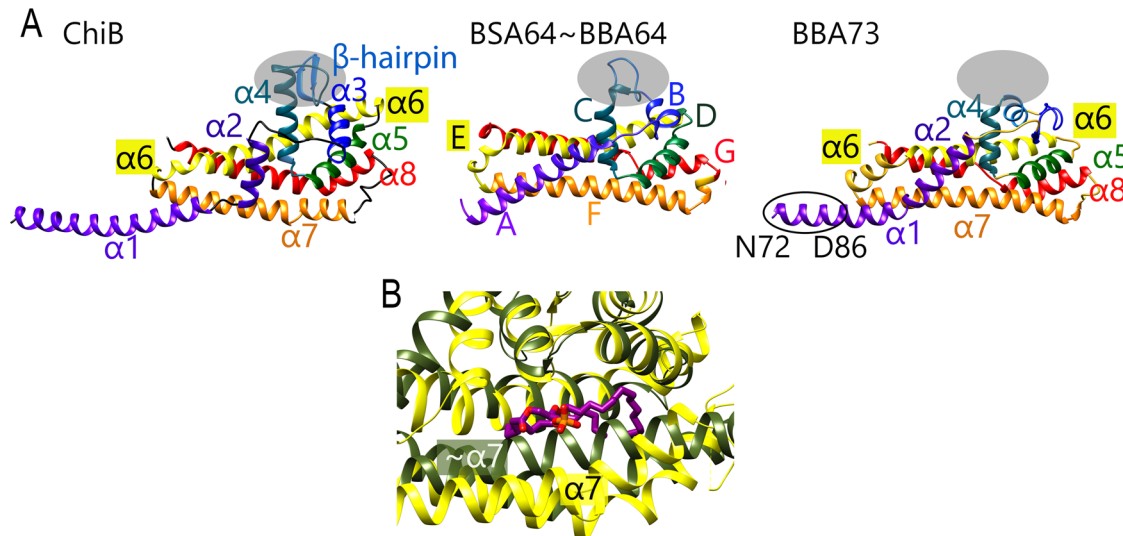

**Fig. 4 | Structural comparison of ChiB with homologous Pfam54_60 proteins.**
**A** Superposition of ChiB (9fqq), BSA64 (6hpn; an ortholog of *B. burgdorferi* BBA64), and BBA73 (4axz), shown in the same orientation using ChiB as the reference. The S-domain, as defined in ChiB, is indicated by a gray ellipse and positioned equivalently in the aligned structures. All proteins adopt a CspA-like fold with varying secondary structure elements. Helices in ChiB are labeled according to this study; BSA64 helices follow the nomenclature from Brangulis et al.[30]. In BBA73, a black ellipse marks the region N72–D86, previously suggested to be involved in dimer formation[82]. **B** Close-up view of the hydrophobic pocket. Superimposed structures of ChiB (yellow) and BBA64 (green; PDB: 4ALY) are shown. The phospholipid bound to ChiB is displayed in stick representation (carbon: magenta; oxygen: red; phosphate: orange). Only ChiB and one Pfam54_60 protein are shown for clarity.

plasminogen binding. Choosing Chi proteins for further analyses, we showed that plasminogen displayed a strong affinity to ChiB to ChiE borrelial proteins exhibiting calculated dissociation constants ($K_d$) of 126.4 nM (± 19.7 nM), 179 nM (± 19.4 nM), 155.8 nM (± 26.6 nM), and 77.3 nM (± 25.8 nM), respectively (Fig. 7C–F). However, no values could be obtained when ChiA was investigated (Fig. 7B).

To gain structural insight into plasminogen binding, we analyzed the distribution of lysine residues in the Chi proteins. Lysines are spread across the protein surface without forming clusters; only K59 and K122 (ChiB numbering) are conserved among all homologs (Figs. 2A and S6). Surface charge analysis likewise revealed no positively charged patches (Fig. S5), suggesting that multiple lysines contribute to plasminogen binding.

Conversion of plasminogen to the active serine protease plasmin is mediated by tissue-type (tPA) or urokinase-type plasminogen activator (uPA)[39]. To demonstrate conversion of plasminogen to active plasmin, immobilized Chi proteins were incubated with the inactive serine protease. Following binding, uPA was added together with the chromogenic substrate S-2251 and cleavage was monitored for up to 24 h. In the presence of uPA, cleavage could be detected for all Chi proteins as well as for HcpA and BBA70[40] (positive controls) while no cleavage occurred when Vsp1[12] and BSA (negative controls) were examined (Fig. S13). As expected, no activation could be monitored when uPA and plasminogen were omitted. A strong activation was demonstrated upon activation of plasminogen in the presence of uPA with the strongest activation achieved with BBA70 as positive control (59.2%) following HcpA (53.4%), ChiE (39.5%), ChiC (34.5%), ChiB (29.5%), ChiD (26.4%), ChiA (21.4%), in relation to plasminogen (100% activity) while Vsp1 reached 11.9% as negative control (Fig. S13). To corroborate the role of lysines in the binding with plasminogen, tranexamic acid, an anti-fibrinolytic lysine analogue was applied. Proteolytic activity of plasminogen was impaired in the presence of tranexamic acid when ChiC and ChiE were assayed but no impact could be observed for ChiA, ChiB, and ChiD, respectively. Further analyses revealed that the cleavage of C3b appeared following activation of plasminogen bound to immobilized Chi proteins (Fig. 7G). Although ChiA possesses a lower plasminogen binding capacity (Fig. 7B), conversion to plasmin appears to be sufficient to cleave C3b. These findings resemble what has been observed for SbiA of *Staphylococcus aureus*[41].

## Chi homologs protect spirochetes from complement-mediated killing and confer serum resistance

To assess the immunomodulatory role of Chi homologs in protecting serum-sensitive *B. garinii* cells from complement-mediated killing, spirochetes were treated with 30% NHS pre-incubated with 10 μM purified Chi proteins (Fig. 8). By counting viable cells after 4 h incubation, all Chi homologs, except ChiA, conferred protection of susceptible spirochetes to NHS. Likewise, the motility and viability of spirochetes were not affected upon incubation with heat-inactivated NHS or BGA66 from *B. bavariensis* PBi, used as a control, as a known inhibitor of the CP, AP, and MAC13. Under the same conditions, approximately 80% of the serum-sensitive cells were killed in the presence of NHS or NHS pre-incubated with 10 μM BSA or Tris/HCl (buffer control) (Fig. 8A). These findings indicate that exogenous Chi proteins protect serum-sensitive *B. garinii* cells from human complement.

To further elucidate the role of Chi proteins for facilitating serum resistance, serum-sensitive *B. garinii* G1 was used to generate a number of gain-of-function strains that ectopically produce individual Chi proteins as well as CihC or HcpA. To confirm expression of the *chi* homologous genes in *B. garinii* G1, RT-PCR was conducted. Initial expression analyses revealed that all borrelial genes analyzed were expressed in the surrogate strain but not in *B. garinii* G1 carrying the basic pKFSS1 shuttle vector (Fig. S14). Moreover, expression of all *chi* genes, as well as *cihC*, *hcpA*, and *flaB* could be demonstrated indicating that the entire *chi* gene cluster was expressed in vitro in the WT strain *B. recurrentis* PAbJ (Fig. S15A). Western blot analyses also confirm synthesis of CihC, HcpA, and ChiB cultivated in BSK medium supplemented with human or rabbit serum (Fig. S15B). Furthermore, sequence analyses identified canonical promoter elements in the upstream regions of each *chi* gene suggesting that these genes, in principle, are expressed in vitro (Fig. S16).

Having demonstrated expression of *chi* genes, we assessed serum survival of gain-of-function *B. garinii* G1 strains by incubating spirochetes in 30% NHS for 4 h (Fig. 8B). Significant levels of serum

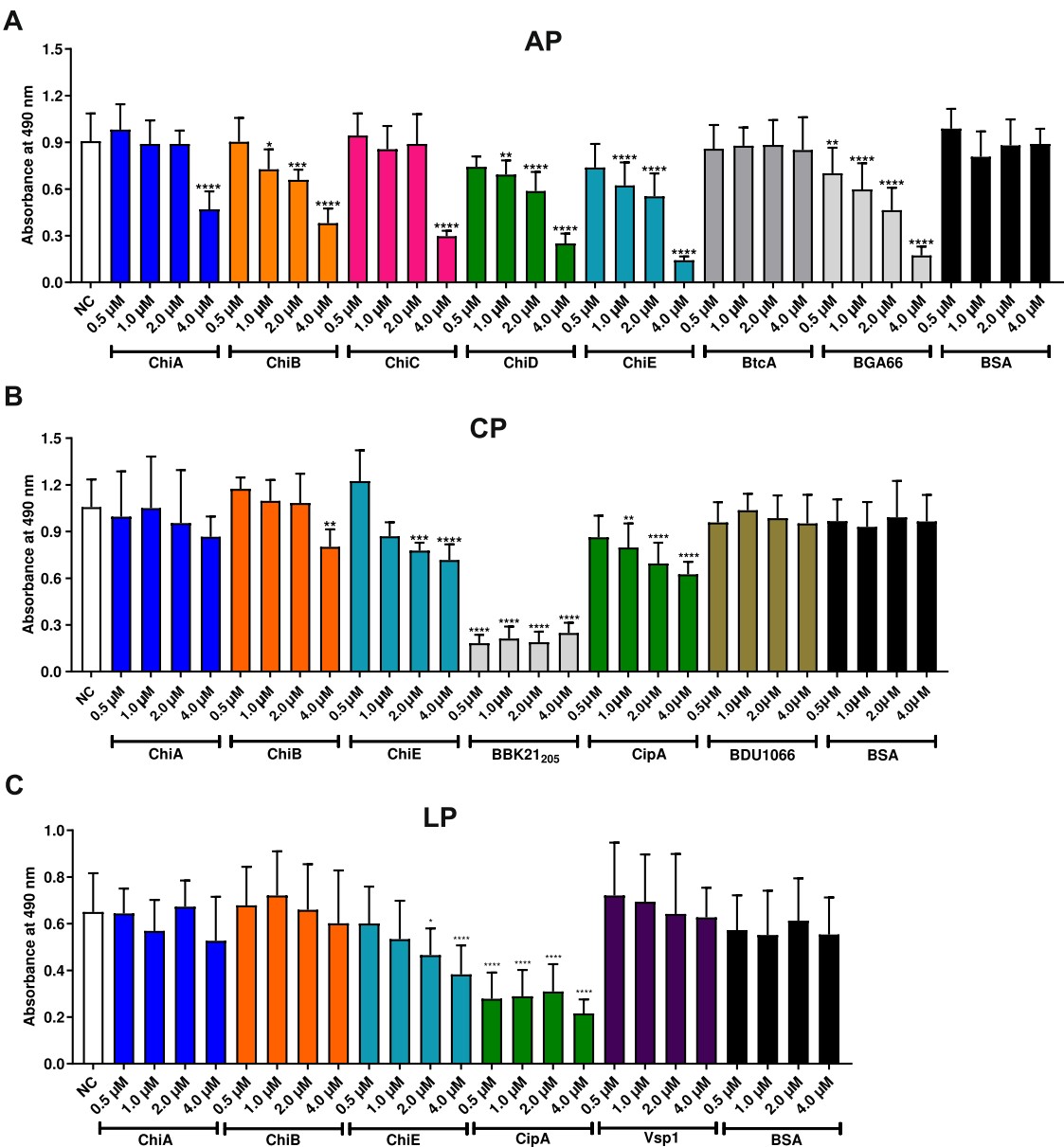

**Fig. 5 | Assessment of the inhibitory capacity of Chi proteins on activation of the AP, CP, and LP.** Dose-dependent inhibition of Chi proteins on the AP (**A**), CP (**B**), and LP (**C**) was assessed by increasing concentration of purified proteins (0.5 to 4 μM). Formation of the MAC was detected by using a monoclonal anti-C5b-9 antibody. Data represent means and standard deviation of at least three different technical replicates each conducted in triplicate ($n = 3$, mean, ±SD). *$p \le 0.1$; **$p \le 0.01$; ***$p \le 0.001$; ****$p \le 0.0001$, n.s., no statistical significance, one-way ANOVA with post-hoc Bonferroni multiple comparison test (confidence interval = 95%). Source data are provided as a Source data file.

resistance comparable to HcpA-producing spirochetes could be observed for the ChiB-, ChiC-, and ChiD-producing spirochetes, respectively. In contrast, ChiA- and ChiE-positive spirochetes were considered serum susceptible as most of the cells were killed during the incubation period. CihC-producing spirochetes appeared to be less protected compared to spirochetes producing HcpA. As expected, no impact on motility and viability could be observed after heat-inactivated NHS (hiNHS) exposure. These findings revealed that ectopically-produced ChiB, ChiC, and ChiD facilitate resistance of serum-sensitive spirochete to human complement.

## Discussion

In this study, we identified a cluster of five genes (Figs. S1 and S2) encoding for proteins that display anti-complement and anti-opsonic properties (Figs. 5, 6, S8, and Table S4). The organization and architecture of this gene cluster are highly conserved among *B. recurrentis*

strains isolated between 1994 and 2015 from clinically confirmed LBRF patients (Fig. S1)[24,26] as well as ancient DNA[42]. The strong conservation over 20 years suggests that this gene cluster has been maintained by selective pressure. Also, the low genetic diversity of *B. recurrentis* might account for an extremely restricted pathogen-vector-host relationship. Comparative genomics identified multiple copies of Chi paralogous genes on the megaplasmid of *B. duttonii* and New World RFB (Fig. S2)[25,43]. Due to their structural similarity to CspA orthologs of LD spirochetes, these encoding proteins were tentatively designated as P35-like proteins, Pfam54_60 proteins or CRASPs without further characterization[25,44].

Noteworthy, our data confirmed the location of the Chi gene cluster on lp190 of *B. recurrentis* A17, a region that was missing in *B. recurrentis* strain A1[27] (Fig. S1). Additional reports suggested that this strain spontaneously lost a portion of the megaplasmid as previously described for *B. turicatae*[45]. Nevertheless, comparative genomic

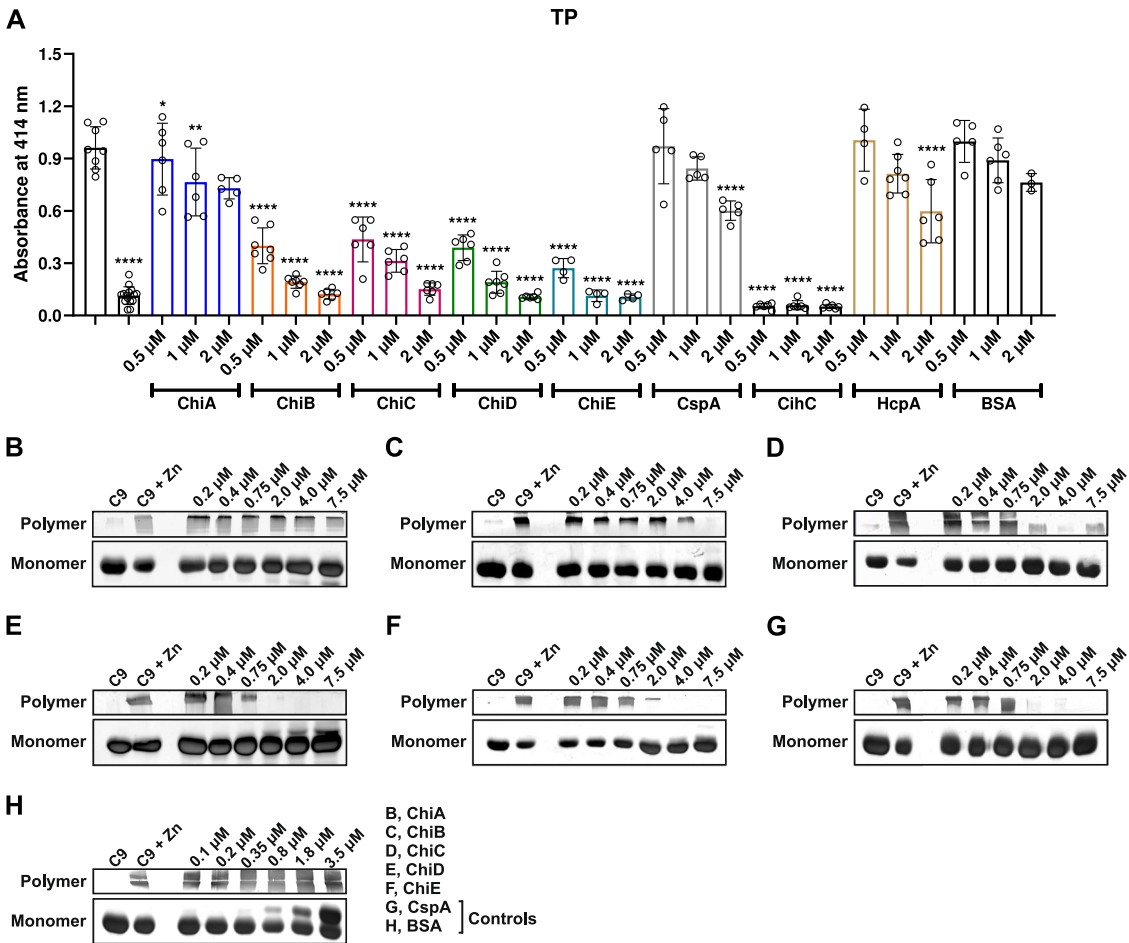

**Fig. 6 | Chi proteins terminate activation of the TP and inhibits C9 polymerization. A** Hemolytic assays were conducted to identify inhibition of the Chi proteins on the TP. Sensitized sheep erythrocytes covered with the preforming C5b-6 complex were incubated with a reaction mixture containing C7, C8, and C9 that was pre-incubated with increasing concentrations of borrelial proteins or BSA (0.5 to 2 μM). Following incubation, hemolysis of erythrocytes was detected at 414 nm. Means of at least three biological replicates ($n = 3$, mean, ±SD). Raw data were analyzed using one-way ANOVA with Bonferroni post test (confidence interval = 95%). *$p < 0.1$; **$p < 0.01$; ***$p < 0.001$; ****$p ≤ 0.0001$. Chi proteins inhibit C9 self-assembly (**B**–**H**). Purified C9 was incubated with increasing concentrations of recombinant proteins or BSA and ZnCl$_2$ were then added to the samples to induced polymerization. As controls, C9 molecules were incubated with and without ZnCl$_2$. Reactions mixtures were then subjected to 7.5% SDS-PAGE and C9 monomers and high molecular weight C9 polymers were visualized by silver staining. Source data are provided as a Source data file.

analyses revealed the presence of lp190 in *B. recurrentis* A1 and A17[16,24]. Proteins encoded by genes located on the megaplasmid are known to play substantial roles in tick colonization and host infection[25,46]. So far, no transcription analyses of these particular genes are available. Here, we show that at least seven genes encoding for CihC and HcpA as well as the five Chi proteins are expressed under in vitro conditions (Fig. S15A). In addition, synthesis of CihC, HcpA, and ChiB could be confirmed in the patient isolate PAbJ grown in the presence of human or rabbit serum (Fig. S15B). Despite the shortcomings of the in vitro data, they set the foundation for future investigations in infected lice or even in blood samples of RF patients to elaborate the importance of these immunomodulatory molecules for bacterial pathogenesis. Of note, antibody responses to at least ChiB (provisionally termed ORF7) and ChiD (provisionally termed ORF9) have been detected in samples from LBRF patients indicating that certain Chi proteins were produced during infection[47]. Along with the observation that proteins encoded by genes located on the megaplasmid are of relevance for *Borrelia* pathogenesis, it can be implied that Chi proteins might take part in mammalian infection. Furthermore, the high sequence identity of the Chi corresponding proteins found in *B. duttonii* (Fig. S3) imply a similar immune evasion mechanism utilized by this particular RFB species.

Structural analyses revealed that Chi proteins adopt a CspA-like fold but differ structurally from other Pfam54_60 members. These proteins fall into two groups differing in the presence and organization of the S-domain (Figs. 4 and S7). In contrast, ChiC and ChiE possess a well-defined S-domain and a hydrophobic pocket containing redox-active cysteines, features absent in other spirochetal complement-interacting proteins such as CspA or BBK32. The ChiB structure also reveals a bound phospholipid within this conserved pocket. In all Chi structures, the pocket is lined with positively charged residues that may interact with negatively charged phosphate groups, suggesting a common binding site for phospholipid-like molecules. Its absence in other Pfam54_60 members, likely due to a shift in helix α7, reflects structural divergence and potentially distinct functional roles (Figs. 1 and 4B). These structural differences may influence how *B. recurrentis* interacts with host factors beyond complement and plasminogen. Moreover, ChiC and ChiE contain conserved cysteines that form a reversible disulfide bond, suggesting redox-sensitive regulation of ligand interaction, as confirmed by structural data and the Ellman assay (Table S4). During infection, *B. recurrentis* may encounter reactive oxygen species such as hydrogen peroxide and hypochlorous acid, which could alter the redox state of these cysteines and thereby

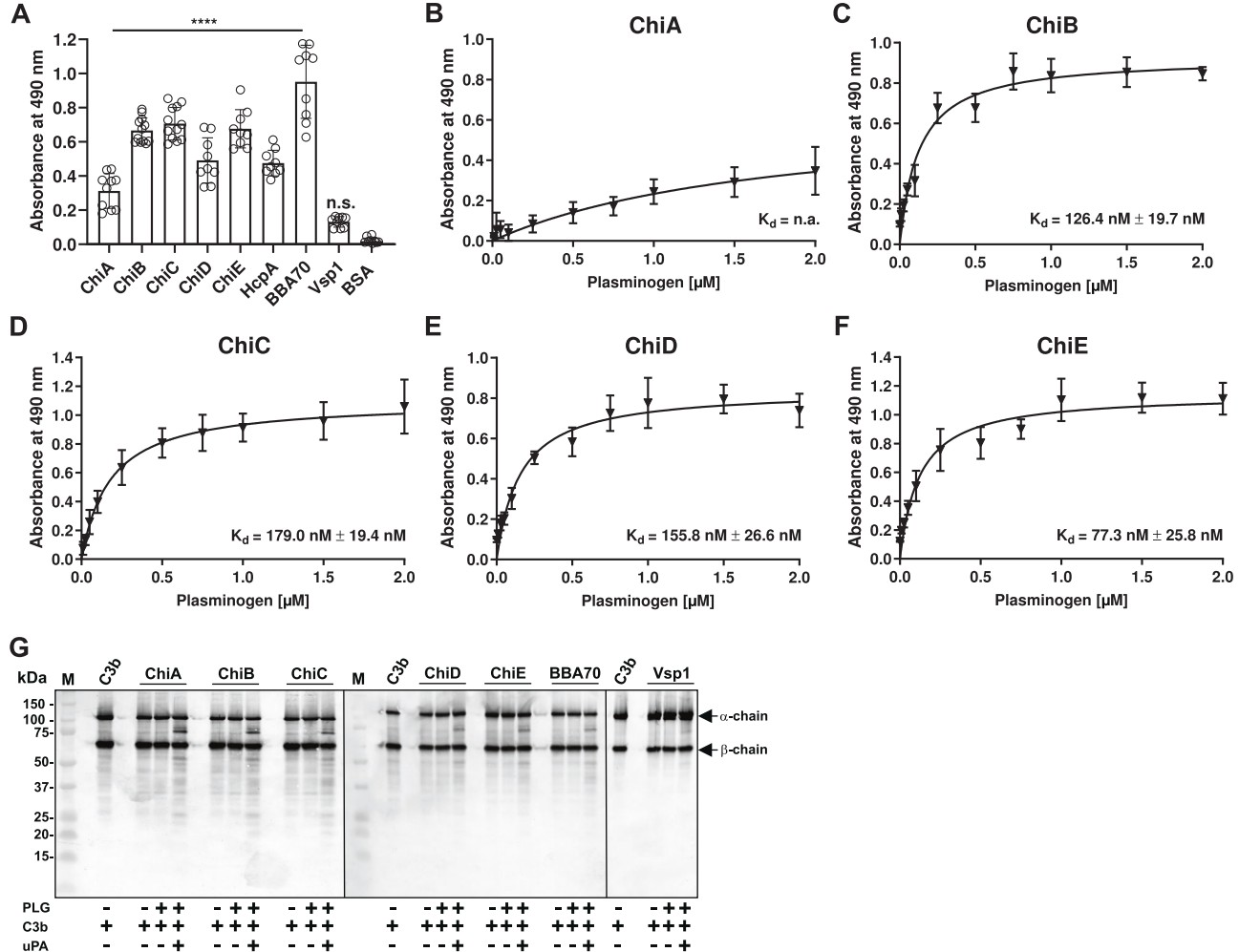

**Fig. 7 | Interaction of Chi proteins with plasminogen. A** Binding of glu-plasminogen to purified borrelial proteins was determined by ELISA. Chi proteins, HcpA and BBA70 (positive controls) or Vsp1 and BSA (negative control) (5 ng/µl each) were immobilized and incubated with 10 ng/µ glu-plasminogen. Protein-protein complexes were detected by using a polyclonal anti-plasminogen antibody (1:1000). Data of at least ten technical replicates ($n = 10$, mean, ±SD) were shown and compared with BSA as negative control. **$p \leq 0.01$, ****$p \leq 0.0001$, n.s., no statistical significance, one-way ANOVA with post-hoc Bonferroni multiple comparison test. **B–F** Dose-dependent binding of glu-plasminogen to three selected Chi proteins. Purified ChiA (**B**), ChiB (**C**), ChiC (**D**), and ChiD (**E**), ChiE (**F**) (5 ng/µl each) were immobilized and incubated with increasing concentrations of glu-plasminogen (0.5 to 2 µM). Binding curve and dissociation constant were calculated via non-linear regression, four-parameter model. Data of three technical replicates ($n = 3$, mean, ±SD) are shown. **G** Degradation of C3b by activated plasminogen bound to Chi proteins. Proteins (10 ng/µl each) immobilized were incubated with plasminogen (10 ng/µl). Thereafter, uPA (25 ng/µl) and C3b (20 ng/µl) were added and C3b degradation products were visualized by Western blotting using a polyclonal anti-C3 antibody (1:1000). n.a., not available. Source data are provided as a Source data file.

modulate protein conformation, potentially contributing to protection of the spirochetes under oxidative stress conditions in the human host. Notably, ChiE also contains residues (H224, C227, D228) arranged in a configuration resembling a catalytic triad (Fig. 3). Despite lacking proteolytic activity against C3 under physiological conditions (Fig. S11), ChiE may mimic or substitute a serine protease within the C3 system, potentially interfering with protease-mediated complement activation without directly cleaving C3.

Additional analyses reveal a functional redundancy of ChiB, ChiC, ChiD, and ChiE inhibiting the AP at ≤4 µM while ChiA displayed a weaker anti-complement activity (Figs. 5 and S8). As ChiA most likely forms homodimers due to an additional cysteine residue, it is conceivable that dimerization impairs the inhibitory activity of this protein. In contrast to what has been observed for other AP-inhibiting borrelial proteins[8,48,49], none of the Chi homologs bound FH indicating that these molecules inactivate the AP by other means (Fig. S9H). Binding of Chi proteins to FB, the catalytic subunit of the AP C3 convertase, appears to be highly unlikely due to the very low absorbance

values measured (Fig. S9B). Hence, the most effective mechanism of AP inhibition mediated by Chi´s targets C3b generation. Breakdown of the AP by interfering with C3b and/or C3bBb has previously been reported for immune evasion molecules SCIN, Efb, Ecb, and Sbi from *Staphylococcus aureus*[50–55]. Concerning inhibition of the CP and LP, solely ChiE terminated both pathways, likely due to the interaction with C1q, C4 or C4b (Figs. 5 and S9D–G).

In contrast to other CP-inhibiting proteins, e.g., CihC of *B. recurrentis*[18] and FbpC orthologs from New World RFB *B. hermsii*[18,19], *B. parkeri*[18], and *B. turicatae*[18], respectively, ChiE did not bind to C1r but to C1q (Fig. S9E, F). This finding suggests that ChiE, like paramyosin of *Schistosoma* spp.[56] exploits a different mode of inhibitory activity on the CP among all known complement-interacting proteins of RFB so far. In addition to their inhibitory capacity at the step of complement initiation, Chi proteins differentially impact the TP. While ChiA showed an attenuated inhibitory activity, ChiC and ChiD possess the strongest effect on C9 polymerization following ChiE and ChiB (Fig. 6). This is in line with the MAC-preventing CspA orthologs of LD spirochetes

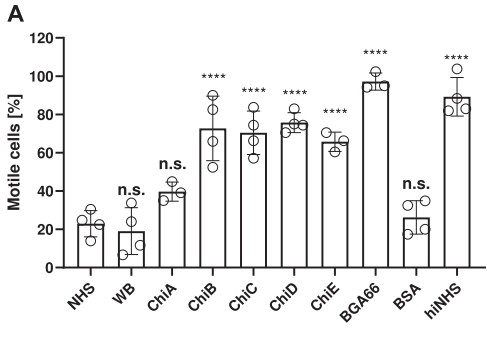

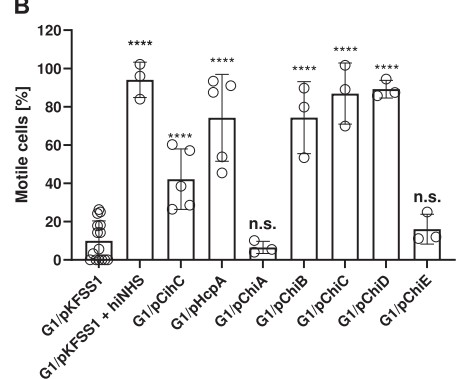

**Fig. 8 | Chi proteins protect spirochetes from complement-mediated killing and facilitate complement resistance.** Protection of spirochetes from complement was assessed by incubating spirochetes with 30% NHS pre-incubated with exogenous purified proteins (final concentration of 10 μM) (**A**). Survival of spirochetes ectopically producing Chi proteins was demonstrated in 30% NHS and compared to the strain carrying the empty shuttle vector (G1/pKFSS1) (**B**). The viability and motility of borrelial cells ($1 \times 10^7$) were determined at 4 h of incubation by dark-field microscopy. Data represent of at least three biological replicates ($n = 3$, mean, ±SD) and compared with NHS as control. ****$p \leq 0.0001$, n.s., no statistical significance, one-way ANOVA with post-hoc Bonferroni multiple comparison test (confidence interval = 95%). Source data are provided as a Source data file.

exhibiting comparable binding properties to C9[11,13] (Fig. 6), as well as paramyosin of *Schistosoma mansonii*[57] and *Trichinella spiralis*[58]. The relevance of multi-functionality on complement was underpinned by the finding that ChiB, ChiC, ChiD, and ChiE protect susceptible spirochetes from complement-mediated killing (Fig. 8A). In addition, at least three gain-of-function strains that ectopically producing individual Chi proteins survived in human serum similar to the CihC- and HcpA-producing strains, respectively[15,16] (Fig. 8B). As expected, ChiA did not protect spirochetes, neither as purified protein nor ectopically produced maybe due to the lowest anti-complement activity (Fig. 5). ChiE displaying inhibitory activity on all three pathways clearly protects susceptible spirochetes from complement-mediated killing comparable to ChiB, ChiC, and ChiD, however, the ChiE-producing strain was killed by human serum (Fig. 8B). Expression of the *chiE* gene in *B. recurrentis* as well as in the gain-of-function strain G1/pChiE (Fig. S14G) most likely suggests that ChiE was (i) not properly folded, (ii) shielded by other outer surface proteins as previously reported for Oms66 and OspA[59], or (iii) sparingly distributed on the spirochetal surface. As ChiB, ChiC, and ChiD promote serum resistance in the surrogate strain, it seems not conceivable that structural changes such as misfolding, shielding by other surface proteins or a restricted abundance might explain the findings collected with the ChiE-producing spirochetes. Thus, employment of a ChiE-specific antibody would help to clarify surface exposure and protein stability, and may therefore contribute to the observed phenotype in the surrogate strain. At this point, we can not explain why ChiE lacked complement-inactivating activity in the transformed spirochetes. Nevertheless, enzymatic removal of surface-exposed proteins including Chi proteins significantly enhances complement sensitivity of *B. recurrentis* as confirmed by an increase of deposited components on the *Borrelia* surface (Figs. S17 and S18).

Identifying the complement-interacting region(s) is essential for understanding the mechanism of serum resistance. Surprisingly, none of the tested modifications including point mutations in the hydrophobic pocket, N-terminal deletions, or removal of the β-hairpin in the S-domain significantly affected complement inhibition or binding to C3b and C5 (Figs. 1A, 2A, and S12). These findings indicate that neither the N-terminal domain, the S-domain, nor the hydrophobic pocket is critical for complement interaction under the tested conditions. In contrast, the failure to express C-terminal deletion variants suggests that this region is essential for protein stability or complement-related function. The conserved hydrophobic pocket may instead serve an alternative role, such as in redox regulation or lipid binding, the relevance of which remains to be elucidated.

Previous investigations underscore the role of plasmin for reversing the effect of opsonisation on the surface of staphylococci or streptococci[60,61]. Moreover, surface-bound plasmin may counteract complement as demonstrated for diverse human pathogenic bacteria[8,49,62,63]. Mechanistically, the interaction of plasminogen with bacterial ligands is often mediated by lysines as observed for ChiC and ChiE but not for ChiA, ChiB, and ChiD (Figs. 7A and S13). As numerous lysines point to the outside of each Chi protein (Fig. 5), we could not identify a region forming a distinct interaction site or even a cluster of positively charged residues that would strongly recommend the replacement of selected residues by in vitro mutagenesis. Nevertheless, activation to plasmin (Fig. S13) supports the notion that acquisition of a broad spectrum protease might be beneficial for *B. recurrentis* to combat innate and adaptive immunity by attenuation of complement, e.g., cleavage of C3b by Chi proteins as shown in Fig. 7G or degradation of anti-*Borrelia* immunoglobulins as demonstrated for *S. aureus*[60]. Not only inactivation of C3b in the fluid phase by Chi-bound plasmin would reduce the formation of C5 convertases but also decreases covalently bound C3b molecules at the borrelial surface. Both scenarios are beneficial for *Borrelia* to combat complement-mediated killing during opsonisation in addition to their properties to directly affect complement activation. It has been shown that recruitment of surface-bound plasmin(ogen) enhances spirochetemia and the ability of LD spirochetes to penetrate the endothelium in the murine hosts[20,21] which is in line with the capability of *B. recurrentis* to survive in high cell densities in blood and to infect multiple organs leading to hepatic dysfunction and cerebral hemorrhages[1].

A limitation of this study is that our analyses were performed under in vitro conditions and not in a physiological host environment. Due to the lack of suitable animal or vector models for *B. recurrentis*, direct in vivo confirmation of the Chi proteins' roles in complement resistance could not be achieved. Nevertheless, our in vitro data provide a mechanistic basis and structural framework for future studies exploring the biological function(s) of Chi proteins during the infection cycle.

Collectively, our data indicate that Chi proteins, while sharing a conserved scaffold with PFam54_60 members, possess unique features including an expanded S-domain, a conserved hydrophobic pocket, and confirmed disulfide-forming cysteines all of which may support additional, yet unknown biological functions. Reflecting the principle

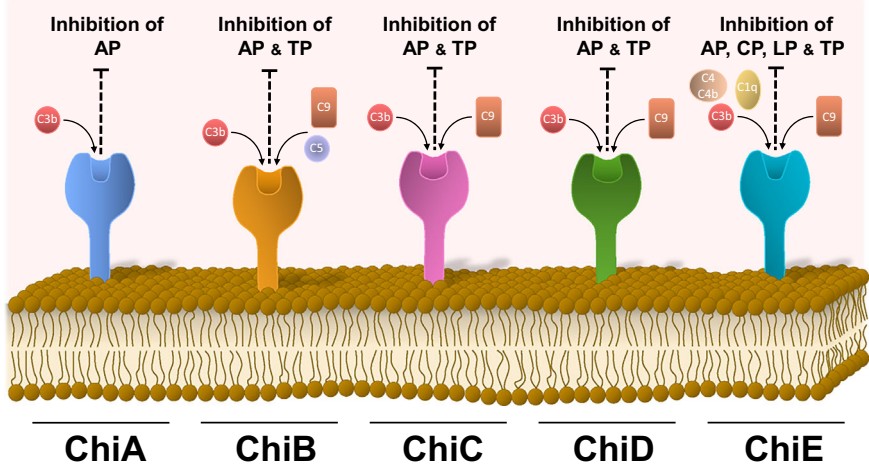

**Fig. 9 | Schematic representation of complement inhibition mediated by Chi proteins.** Inhibition of complement by distinct Chi proteins acting at certain levels of distinct activation pathways. AP alternative pathway, CP classical pathway, TP terminal pathway.

of functional redundancy among bacterial virulence factors, Chi homologs unexpectedly combine multifactorial anti-complement and plasminogen-binding properties (Fig. 9 and Table S4), thus, enabling *B. recurrentis* to evade innate immunity and establish infection. By closing the current gap in our understanding of immune evasion in this particular microorganism, these insights may also illuminate broader strategies employed by other vector-borne pathogens.

## Methods
### Bacterial strains, biological and geographical origin, and culture conditions
*Borrelia recurrentis* A17 (human blood isolate, Addis Ababa, Ethiopia[64]), *B. recurrentis* PAbJ (human blood isolate, Somalia[24]), *B. duttonii* Ly (human blood isolate, Mvumi, Tanzania[64]) and *B. garinii* G1 (CSF isolate, Germany[65]) were cultured until mid-exponential phase ($5 \times 10^7$ cells per ml) at 33 °C in Barbour-Stoenner-Kelly (BSK-H) medium (Bio&SELL, Feucht, Germany) supplemented with 7.4% heat-inactivated rabbit serum (Merck, Darmstadt, Germany). All transformed *B. garinii* strains carrying the respective shuttle vectors were cultured in BSK-H medium supplemented with 100 µg/ml streptomycin (Merck). *Escherichia coli* strains NEB 5-alpha, BL21(DE3), BL21 Star (DE3) (New England Biolabs, Frankfurt, Germany) or M15 (Qiagen, Hilden, Germany) were used as hosts for propagation of respective vectors and for production of recombinant His-tagged proteins were grown at 37 °C in yeast tryptone (YT) broth containing 50 µg/ml ampicillin (Carl Roth GmbH, Karlsruhe, Germany) or 100 µg/ml streptomycin (Merck).

### Human serum, proteins, and antibodies
NHS collected from healthy blood donors was initially tested for the presence of anti-*Borrelia* IgM and IgG antibodies by a commercially available ELISAs (Enzygnost Borreliosis/IgM and Enzygnost® Lyme link VlsE/IgG, Siemens Healthcare Diagnostics Products GmbH, Marburg, Germany)[12]. Only sera considered to be negative were combined to form a serum pool. The total complement activity (CH50) of the serum pool was assessed by employing an ELISA-based assay (WiELISA) for measuring the activity of the classical pathway.

Complement components C1q, C2, C3, C3b, C4, C4b, C5, C9, FB, FH, FI, C4BP, and the C5b-6 complex were purchased from Complement Technology (Tyler, TX, USA). Human glu-plasminogen was purchased from Prolytix (Essex Junction, VT, USA). For the activation of plasminogen, urokinase plasminogen activator (uPA) and the chromogenic substrate D-Val-Leu-Lys *p*-nitroanilide dihydrochloride (S-2251) were used from Merck (Darmstadt, Germany). Polyclonal anti-

plasminogen antibody was purchased from Acris Antibodies (Herford, Germany). Purified vitronectin were obtained from Merck.

The polyclonal anti-C5 antibody was obtained from Complement Technology (Tyler, Texas, USA). Polyclonal antisera raised against complement C3, C4, and FH were from Merck (Darmstadt, Germany) and polyclonal antisera against C1q, C5, FI, and FB as well as the neoepitope-specific monoclonal antibody against the C5b-9 complex were from Quidel (San Diego, USA). The polyclonal anti-plasminogen antibody was purchased from OriGene Technologies (Rockville, MD, USA). For the detection of His-tagged proteins, a mixture of monoclonal anti-His antibodies were used (GE Healthcare, Munich, Germany and Merck, Darmstadt, Germany). All horseradish peroxidase (HRP)-conjugated immunoglobulins were obtained from Agilent Technologies Denmark, Glostrup, Denmark. The monoclonal anti-HcpA and anti-CihC antibodies were described previously[15,16]. The polyclonal rabbit anti-ChiB antibody was produced by a commercial provider (Eurogentec, Seraing, Belgium). For the detection of the periplasmatic FlaB protein, the monoclonal anti-FlaB Ab L41 1C11 was used[66]. Proteinase K was purchased from Merck (Darmstadt, Germany) and Pefabloc SC was from Carl Roth (Karlsruhe, Germany).

### Sequence and phylogenetic analysis
For comparative genomics, *Borrelia* genomes stored under BioProject PRJNA378726, as well as the *B. recurrentis* A1 genome stored under accession number NC_011244 were obtained from NCBI (Source data are provided as a Source data). All genome sequences were then uniformly annotated using Prokka 1.14.6. Searches for homologs were carried out based on resulting FASTA files using Blast p 2.9.0 and visualizations of genomic loci were created using CLC Genomics Workbench version 12.0.2. Lalign (EMBL-EBI, Hinxton, UK) was used for pairwise sequence alignment to calculate protein sequence identity and similarity.

To recognize canonical promoter motifs within the *chi* gene cluster, the YAPP Eukaryotic Core Promoter Predictor (https://www.bioinformatics.org) and Promotech (R. Chevez-Guardado & L. Peña-Castillo, 2021, https://doi.org/10.1186/s13059-021-02514-9) were used. The YAPP tool is created for TATA boxes, initiator elements (INR), downstream core element (DPE) in upstream eukaryotic but also for prokaryotic promoter sequences. YAPP algorithm calculates matrix similarity score for matches with consensus sequences to qualify as promoter elements. True positive predictions display score values that tend to be around 0.5 or higher when using Promotech, a machine-learning-based method.

For phylogenetic analysis, sequences of all respective proteins were aligned using Clustal Omega multiple sequence alignment

(https://www.ebi.ac.uk/jdispatcher/msa/clustalo). Visualization of the tree was drawn using iTOL (https://itol.embl.de/). Sequence analysis of the inserted DNA fragments was performed by using the CLC Sequence Viewer 8.0 (QIAGEN Aarhus A/S, Denmark) and SnapGene Viewer version 7 (GSL Biotech LLC, San Diego, CA, USA).

## Crystallization, data collection, and processing

For crystallization of ChiB, the protein was concentrated in 200 mM NaCl and 50 mM Tris (pH 8.0) to 10 mg/ml. Crystals were grown at 22 °C in sitting drops with the vapor diffusion technique, using a Honeybee 961 crystallization robot. In the drop, we mixed 0.2 µl protein solution with 0.2 µl reservoir solution (1.7 M ammonium citrate, 5 mM DTT). The used heavy atom derivative crystal was produced by soaking a crystal with 1 mM ethylmercury phosphate for 3 h. Before data collection, crystals were soaked in mother liquor with a final concentration of 25% glycerol. ChiA crystals were retained using the seed bead method. The initial crystals (protein concentration 10 mg/ml) were grown in 20% PEG400, 16% PEG4000, 70 mM MgCl₂, 100 mM Tris (pH 8.5) at 4 °C in sitting drops with the vapor diffusion technique, using a Honeybee 961 crystallization robot (Genomic Solutions). The seed crystals were grown at 22 °C in hanging drops. In the drop we mixed 0.2 µl protein solution with 0.2 µl reservoir solution. The reservoir contained 16% PEG4000, 20% PEG400, 50 mM MgCl, 100 mM Tris (pH 8.5). Before data collection, the crystals were soaked in 25% PEG4000, 25% PEG400, 50 mM MgCl₂, 100 mM Tris (pH 8.5), and 10% glycerol. Diffraction data for all crystals were collected at X10SA (detector: Pilatus or EIGER2 16 M) of the Swiss Light Source in Villigen, Switzerland.

The data were collected at 100 °K and processed with XDS[67]. The tetragonal crystals of native ChiB diffracted up to 1.5 Å resolution and obeyed $P4_22_12$ space group symmetry with one monomer in the asymmetric unit. ChiA crystallized in a monoclinic space group (C2) with two monomers in the asymmetric unit and diffracted up to 2.7 Å. During refinement, we omitted 10% (ChiA) or 8% (ChiB) of the reflections, which were used for calculation of an $R_{free}$ value.

## Structure determination

The three-dimensional structure of ChiB was solved through a combination of molecular replacement and anomalous dispersion techniques. A search model was constructed using SWISS-MODEL[68] through homology modeling. The outer surface protein BBA66 from *Borrelia burgdorferi* (2yn7)[29], sharing a sequence identity of 14% with ChiB, was employed as a template for this purpose. However, owing to the limited sequence similarity, this approach did not yield a solution.

To address this issue, we soaked the ChiB crystals with ethylmercury phosphate, varying the duration and concentration of the soak. Only one dataset exhibited an anomalous signal, detectable up to 4 Å. Merging of the native and heavy atom data sets of ChiB was performed using XDS[67]. The asymmetric unit of the tetragonal crystals contains a single monomer. Notably, ChiB has only one methionine and one cysteine residue, indicating a single heavy atom site. Utilizing AutoSol, implemented in the program package Phenix[69], the native data and anomalous scattering data from the ethylmercury phosphate dataset (sad peak, 1 site) yielded a Bayesian correlation coefficient of 16% and an $R_{free}$ of 54%. Despite these poor values, three helices were visible in the electron density calculated from the results of the heavy atom dataset. We then successfully manually fitted a partial model (150 amino acids) based on 2yn7 to this helix density. The $R_{free}$ value of this new solution was 41%. Subsequent application of autobuild (implemented in the program package Phenix)[69] to the high-resolution dataset (1.5 Å) resulted in a CC of 82% and an $R_{free}$ of 26%. After several rounds of manual rebuilding and subsequent refinement, we revealed a model of ChiB, comprising residues E24 to N231 and G237 to Q284 with a final $R_{free}$ of 21% and good stereochemistry. In addition to

several solvent molecules, the initial Fo–Fc electron density map revealed clear density that could not be satisfactorily explained by buffer components present during purification and crystallization (Tris, citrate, glycerol, DTT) and was most consistent with a phospholipid-like molecule. The density contained a prominent peak consistent with a phosphate group and extended into two elongated hydrophobic chains with additional density features compatible with carbonyl groups. Based on its size and geometry, the ligand was modeled as LPP and refined with standard stereochemical restraints. As no lipid was introduced during purification or crystallization, the most plausible explanation is co-purification of a bacterial phospholipid during heterologous expression in *E. coli*.

The sequence identity between ChiA and ChiB is only 38.7%. However, the structure of ChiA could be solved by using the refined model of ChiB as a template for molecular replacement. The final ChiA model includes residues D29 to S213 and H222 to I265 and was refined to an $R_{free}$ value of 30%. The final statistics of ChiA and ChiB are summarized in Table S2.

The PHENIX program suite served for reflection phasing and structure refinement[69]. The interactive graphics program Coot[70] was used for model building, the superpositions of the structures were made using SSM Superposition[71], which is implemented in the Coot program package or with routines included in the UCSF Chimera package[71]. Molecular graphics images were produced using the UCSF Chimera package[35].

## Modeling of ChiA, ChiB, ChiC, ChiD, and ChiE by AlphaFold2

We used the protein sequences of ChiA, ChiB, ChiC, ChiD, and ChiE from *B. recurrentis* A1 (GenBank accession number NC_011244 and Source data file) to perform structural predictions using the multimer model of Alphafold2. A comparative analysis between the predicted rank-ordered PDB structures and the structures obtained using the Amber relax option revealed only minor differences. Therefore, we selected the best ranked amber-relaxed structures. To assess which regions of the predicted structures are trustworthy, the heatmaps obtained from Alphafold2 were analyzed using UCSF ChimeraX[72]. In addition, a structural comparison was made between the X-ray structures of ChiA and ChiB and the corresponding AlphaFold2 structures. All structures were obtained using the AlphaFold2 program[73] available at the MPI for Medical Research in Heidelberg, Germany; monomers were modeled with version v2.2.0 using the multimode option, and dimers were modeled with version v2.2.0 and v2.3.1.

## Determination of free thiol groups using Ellman's reagent (DTNB assay)

To quantify accessible thiol groups, 200 µl of purified Chi proteins (0.2–6 mg/ml) were incubated with 5 mM reducing agent (DTT or β-mercaptoethanol) for 30 min at 37 °C. Meanwhile, ZebaSpin 2 ml desalting columns (Thermo Scientific) were prepared according to the manufacturer's instructions. Columns were placed in 15 ml collection tubes and centrifuged three times at 1000 × *g* for 2 min at 4 °C with 1 ml of 150 mM potassium phosphate buffer (pH 8.0) per wash. After equilibration, 200 µl of reduced protein sample mixed with 40 µl of phosphate buffer (pH 8.0) was applied to the column and centrifuged at 1000 × *g* for 2 min at 4 °C. The eluate represented the desalted, reduced protein.

For DTNB assays, 20–60 µl of this eluate was transferred to a 96-well half-area plate in duplicates. Wells were adjusted to 80 µl total volume with phosphate buffer (pH 8.0) and DTNB added to a final concentration of 400 µM. Absorbance was recorded at 412 nm immediately and at 1, 2, 5, 10, and 20 min using a Tecan plate reader. Protein concentrations were determined by Bradford assay on remaining sample. To assess spontaneous oxidation, the remaining reduced protein was stored at 4 °C for 4–5 days, then processed again as above for DTNB reactivity (Table S3).

## Purification of His6-tagged proteins

To purify His$_6$-tagged proteins, *E. coli* cells carrying the appropriate plasmid were grown in YT-broth supplemented with 50 μg/ml ampicillin (Merck, Darmstadt, Germany) to an OD$_{600}$ of 0.5. Thereafter, protein production was induced by adding 0.2 mM IPTG for 4 h at room temperature and cells were then harvested by centrifugation. The sedimented cells were stored at −80 °C until use and then resuspended in lysis buffer (50 mM NaH$_2$PO$_4$, 300 mM NaCl, 10 mM imidazole, pH 6.8) supplemented with 1 mg/ml lysozyme (Merck, Darmstadt, Germany) and incubated for 30 min on ice. *E. coli* cells were then disrupted by homogenization using a MiCCRA D-9 disperser (Art Prozess- & Labortechnik GmbH; Heitersheim, Germany) following sonification six times for 30 s each with a Sonifier 450 (Branson Ultrasonics, Danbury, CT). After centrifugation, the supernatant was passed through a 0.45 μm filter and proteins were purified by affinity chromatography using NEBExpress Ni resin (New England Biolabs, Frankfurt, Germany) with increasing imidazole concentrations of 50 to 300 mM. For buffer exchange, 50 mM Tris (pH 8.0) combined with an ultrafiltration centrifugal device (cut off 10,000, Pierce) was used. The purity and size of recombinant proteins were then analyzed by subjecting 20 μl of the column eluates on a 10% Tris/Tricine SDS-PAGE following silver staining[40]. The protein concentration of each protein was determined by employing the Pierce BCA protein assay kit (Thermo Fisher Scientific, Rockford, IL, USA).

For the determination of free thiol groups (see below), *E. coli* cells producing Chi proteins were resuspended in lysis buffer containing 0.5 mM DDT. After incubation on ice for 30 min, cells were disrupted by homogenisation and sonication. Following centrifugation, proteins were purified by IMAC in the presence of 0.5 mM DTT with increasing imidazole concentrations of 50 to 300 mM. To obtain reduced proteins, fractions collected were concentrated in the presence of 50 mM Tris (pH 8.0) containing 0.5 mM DDT by using ultrafiltration centrifugal devices and proteins were then stored at −20 °C before use.

## SDS-PAGE and Western blot analysis

Purified His$_6$-tagged proteins or whole cell lysates were separated to 10% Tris/Tricine SDS-PAGE under reducing conditions and transferred to nitrocellulose membranes[18]. Briefly, the membranes were blocked with 5% nonfat dry milk in TBS containing 0.1% Tween 20 (TBS-T). After three wash steps with TBS-T, membranes were incubated with appropriate antibodies followed by horseradish peroxidase-conjugated anti-mouse or anti-rabbit immunoglobulins. Protein-antigen complexes were detected by tetramethylbenzidine as substrate. Images of the gels and nitrocellulose membranes were processed by using a GS-900 calibrated densitometer (Bio-Rad, Hercules, CA, USA) and the Image Lab version 6.1 (Bio-Rad).

## Enzyme-linked immunosorbent assay

To detect binding of complement components or plasminogen, microtiter plates (Nunc MaxiSorp, Thermo Fisher Scientific) were coated with 100 μl of purified His$_6$-tagged proteins (5 μg/ml) or BSA (5 μg/ml) in PBS at 4 °C overnight[74]. Between every incubation step, wells were washed three times with PBS containing 0.05% (v/v) Tween 20 (PBS-T). After blocking with Blocking Buffer III BSA (AppliChem, Darmstadt, Germany) or with PBS containing 0.2% gelatine (w/v) (AppliChem, Darmstadt, Germany), complement components (5 μg/ml each) or glu-plasminogen (10 μg/ml) in PBS was added. Binding of complement components or glu-plasminogen were then assessed by utilizing specific primary antibodies (dilution 1:1000). Following incubation for 1 h at RT, HRP-conjugated anti-goat or anti-mouse IgG (dilution 1:1000) were added and protein complexes were visualized using *o*-phenylenediamine (Merck, Darmstadt, Germany). The absorbance was read at 490 nm employing the PowerWave HT spectrophotometer (Bio-Tek Instruments, Winooski, VT, USA).

To determine dose-dependency, borrelial His$_6$-tagged proteins were immobilized (5 μg/ml) and incubated with increasing amounts of the respective complement components or glu-plasminogen. The antigen-antibody complexes were detected by using appropriate anti-complement antibodies as described above.

## Complement inactivation assays

A modified ELISA-based approach (WiELISA) was applied to assess the inhibitory capacity of bacterial proteins on the alternative (AP), classical (CP), Lectin pathway (LP)[75]. Briefly, Nunc MaxiSorp 96-well microtiter plates were coated with either LPS (10 μg/ml) (Hycult Biotech, Beutelsbach, Germany) for the AP, human IgM (3 μg/ml) (Merck, Darmstadt, Germany) for the CP, or mannan (100 μg/ml) (Merck, Darmstadt, Germany) for the LP at 4 °C overnight. Following three wash steps with TBS containing 0.5% (v/v) Tween 20 (TBS-T), the wells were blocked with PBS-T containing 1% BSA for 1 h at RT. NHS (15% for the AP, 1% for the CP, and 2% for the LP) was then pre-incubated with a final concentration of 4 μM (initial analyses) or increasing concentrations (0.5, 1, and 4 μM) (dose dependence analyses) of purified His$_6$-tagged proteins for 15 min at RT before being added to the wells to initiate complement activation of the respective pathway. After washing with TBS-T, a neoepitope-specific, monoclonal anti-C5b-9 antibody (1:500) was added to detect formation of the MAC as the final activation step of the cascade. Following incubation for 1 h at RT, wells were washed thoroughly with TBS-T and incubated with HRP-conjugated anti-mouse immunoglobulins (1:1000) at RT for 1 h. All reactions were developed applying tetramethylbenzidine as substrate.

In order to examine the inhibitory potential of the Chi proteins on the terminal pathway, a hemolytic assay was conducted[75]. Briefly, sensitized sheep erythrocytes ($1.5 \times 10^7$ cells) (kindly provided by Dr. Michael Kirschfink (emer.), Institute of Immunology, Heidelberg, Germany) were pre-incubated with C5b-6 (1.5 μg/ml) for 10 min at RT. In parallel, complement C7 (2 μg/ml), C8 (0.4 μg/ml), and C9 (2 μg/ml) were pre-incubated with or without purified His$_6$-tagged proteins (0.5, 1, and 2 μM) for 5 min at RT. The pre-incubated proteins were then added to the C5b-6 coated sheep erythrocytes. Following incubation for 30 min at 37 °C, erythrocytes were sedimented by centrifugation and the supernatants were transferred to a microtiter plate. The hemolysis of the erythrocytes was then determined by measuring the absorbance of the supernatants at 414 nm.

Concerning the controls utilized, previously characterized His-tagged proteins originated from LD or RF *borreliae* as well as *Acinetobacter baumanii* were produced and purified using the same protocol as described above. These control proteins were selected according to their capability to inhibit the respective complement pathway as follows: BGA66 from *B. bavariensis*[13] (AP inactivation), CihC from *B. recurrentis*[18] and the C-terminal fragment of BBK32 (BBK32$_{205}$) from *B. burgdorferi*[18] (CP inactivation), CipA from *Acinetobacter baumanii*[76] (LP inactivation), and CspA from *B. burgdorferi*[11] and CihC from *B. recurrentis*[18] (TP inactivation). As negative controls, BtcA from *B. turicatae*[38] (AP inactivation), BDU1066 from *B. recurrentis* (CP inactivation), and Vsp1 from *B. miyamotoi*[37] (LP inactivation), and HcpA from *B. recurrentis*[15] (TP inactivation) were chosen. In addition, vitronectin (Vn) was included in the cell-based hemolytic assay as a natural inhibitor of the TP as it binds to the preassembled Cb5-7, C5b-8, and C5b-9 complexes and thereby prevent MAC formation[77].

## Determination of the inhibitory capacity of Chi proteins on C9 polymerization

To assess the inhibitory capacity of the purified borrelial proteins on C9 polymerization, increasing concentrations (final concentrations 0.005 μg/μl to 0.22 μg/μl or 0.2 μM to 9 μM) of the Chi proteins of *B. recurrentis*, CspA of *B. burgdorferi* (positive control), and BSA (negative control) was incubated with C9 (0.06 μg/μl or 0.9 μM) for 40 min at

37 °C[11,13]. Thereafter, auto-polymerization of C9 was induced by adding 50 μM ZnCl$_2$ to each reaction mixture and incubated for 2 h at 37 °C. As additional controls, purified C9 was incubated with or without ZnCl$_2$. Reaction mixtures were then subjected to 8% Tris/Tricin-SDS gels and monomeric and polymeric C9 molecules were visualized by silver staining.

## Plasmin(ogen) activation assay

Activation of glu-plasminogen by uPA and cleavage of the chromogenic substrate D-Val-Leu-Lys-p-nitroanilide dihydrochloride was entirely described previously[40,78]. In brief, microtiter plates were immobilized with 100 μl of His$_6$-tagged proteins or BSA (5 ng/μl each) in PBS overnight following incubation with 10 ng/μl of glu-plasminogen for 1 h at room temperature. After three wash steps, each well was incubated with 0.3 μg/μl S-2251 in 50 mM Tris/HCl (pH 7.5), 300 mM NaCl, and 0.003% Triton X-100. Finally, 4 μl of 2.5 ng/μl urokinase plasminogen activator (uPA) were added to each well to activate protein-bound plasminogen. Further reactions containing 50 mM tranexamic acid, a lysine analogue with high affinity to the lysine binding sites of plasminogen to assess the role of lysines on binding of plasminogen to Chi proteins. As additional controls, reaction mixtures were prepared in which plasminogen or uPA were omitted. For long time measurement (24 h), microtiter plates were sealed, placed in an ELISA reader and incubated at 37 °C. The absorbance was measured every 30 min at 405 nm. To calculate the dissociation constant for the binding of plasminogen to Chi proteins, the non-linear regression model (four parameters dose-response curve) with a variable slope (Hill slope) was selected in GraphPad Prism 10.2.2.

## C3b degradation by activated plasmin bound to Chi proteins

Chi proteins, BBA70 of *B. burgdorferi*, Vsp1 of *B. recurrentis*, and BSA (10 ng/μl each), respectively, were immobilized on microtiter plates in PBS overnight. After three wash steps, each well was incubated with glu-plasminogen (10 ng/μl) for 1 h at room temperature. Following washing, uPA (25 ng/μl) and C3b (20 ng/μl) were added to each well and reactions were incubated overnight at 37 °C. Samples were separated by Tris/Tricine-SDS-PAGE and transferred onto a nitrocellulose membrane. C3b cleavage products were detected by Western blotting employing a polyclonal anti-C3 antibody.

## Generation of expression and shuttle vectors for the electroporation of spirochetes

The generation of vectors producing N-terminally His$_6$-tagged proteins used as controls for this study was previously described and includes CihC and HcpA of *B. recurrentis* A17[15,16], BtcA of *B. turicatae*[38], CbiA and Vsp1 of *B. miyamotoi* LB-2001[12,37], CspA and BBA70 of *B. burgdorferi* LW2[40,74], BGA66 of *B. bavariensis* PBi[13], and CipA of *Acinetobacter baumannii* 19606[79]. As a further control, a C-terminal fragment of BBK32 of *B. burgdorferi* B31 known to inhibit CP activation[80] was generated by PCR. First, the *bbk32* gene was amplified using primers BBK32 Bam_FP and BBK32 Hind_RP (Table S5), and vector pMal-c/BBK32 as template (kindly provided by Yi-Pin Lin, Department of Infectious Diseases & Global Health, Cummings School of Veterinary Medicine, Tufts University, North Grafton, USA). The amplified DNA fragment was digested with appropriate restriction endonucleases and re-cloned into the expression vector pQE-30 Xa (Qiagen, Hilden, Germany). The resulting plasmid pQE-BBK32 served as template and oligonucleotides BBK32-205 BamHI and pQE-RP (Table S5) for a subsequent PCR amplification to engineer a His-tagged BBK32$_{205}$ fragment containing the C-terminal amino acids 205 to 356. After digestion, the DNA fragment was ligated into pQE-30 Xa and the resulting plasmid was transformed into *E. coli* BL21 Star (DE3) cells. In addition, BDU1066 located on lp165 of *B. duttonii* Ly and supposed to display similar complement-inhibitory functions as the factor H-binding CspA protein of *B. burgdorferi* B31[25] was also included in this

study as a further control. The open reading frame (lacking the putative lipoprotein signal sequence) of the BDU1066 encoding gene of *B. duttonii* Ly was amplified by PCR using oligonucleotides Bre_1066_FP_Bam and Bre_1066_RP_Sal (Table S5). After digestion with BamHI and SalI, the DNA fragment was cloned into pQE-30 Xa. The resulting plasmid pQE-Bdu_1066 was then used to transform *E. coli* BL21 Star (DE3) cells. Plasmids isolated from selected clones were sequenced to ensure that no mutations were incorporated during PCR and the cloning procedure. Expression vectors encoding for N-terminally His$_6$-tagged ChiA, ChiB, ChiC, ChiD, and ChiE of *B. recurrentis* A17 were kindly provided by Reinhard Wallich (emer.), Institute of Immunology, University of Heidelberg, Germany. In addition, to increase the yield and purity of CihC, the encoding gene was re-cloned into the pET-16b expression vector (Merck, Darmstadt, Germany) by using primers CihC_Nde_FP and CihC_Bam_RP (Table S5). The resulting vector pET-CihC producing a N-terminally located His$_6$-tagged CihC protein was then transformed into *E. coli* M15 cells according to the manufacturer's instructions.

To introduce deletions as well as single and double amino acid substitutions in ChiB, site-directed mutagenesis was conducted[40]. Briefly, PCR was carried out for 18 cycles (95 °C for 15 s, 60 °C for 15 s and 72 °C for 90 s) using 50 ng/μl pQE-ChiB, 125 ng each of the oligonucleotides (Table S5), and 4 U PCRBIO VeriFi polymerase (PCR Biosystems, London, UK). Following incubation with 10 U *Dpn*I (New England Biolabs, Frankfurt, Germany) to eliminate the remaining vector DNA, reactions were used to transform *E. coli* NEB 5-alpha cells. To generate truncated ChiB proteins lacking either the N- or the C-terminus or a ChiB protein lacking the loop region, PCR was performed with specific oligonucleotides (Table S5) and the PCRBIO HiFi polymerase (PCR Biosystems, London, UK) applying the identical conditions as stated above.

Shuttle vectors harboring the genes encoding for ChiA, ChiB, ChiC, ChiD or ChiE were generated by PCR amplification of the respective genes along with their potential promoters at the 5´end. Each gene was amplified by using genomic DNA from *B. recurrentis* A17 as template with oligonucleotides listed in Table S5. Following amplification and digestion with the respective endonucleases, each DNA fragment was cloned into the shuttle vector pKFSS1. Plasmids were prepared from presumptive *E. coli* clones with the Monarch plasmid kit (New England Biolabs, Frankfurt, Germany) and DNA inserts were Sanger sequenced by a commercial provider (Eurofins Genomics, Ebersberg, Germany).

## Transformation and characterization of serum-sensitive *B. garinii* strains ectopically producing individual Chi protein of *B. recurrentis* A17

A high-passage, non-infectious *B. garinii* strain G1 selected as surrogate strain was grown in 100 ml BSK-H medium and harvested at mid-exponential phase ($5 \times 10^7$ to $1 \times 10^8$ cells/ml). Electrocompetent cells were prepared as described previously with slight modifications[75]. Briefly, 50 μl aliquots of competent *B. garinii* G1 cells were electroporated at 12.5 kV/cm in ice-cold 2-mm cuvettes with 20 μg of plasmid DNA. After electroporation, spirochetes were immediately transferred into 10 ml of BSK-H medium without antibiotics. Following incubation for 18 h at 33 °C, the cell suspension was further diluted by adding 90 ml BSK-H medium containing streptomycin (25 μg/ml) and 200 μl aliquots were seeded into 96-well cell culture plates. After six to eight weeks, streptomycin-resistant clones were macroscopically detected by a color change of the BSK-H medium. Individual clones selected were expanded in 1 ml of fresh BSK medium without antibiotic selection for 7 days, and then transferred into 10 ml of fresh BSK-H medium containing streptomycin (50 μg/ml). Selected clones were then further characterized by amplifying the inserted genes using primers M13 For and M13 Rev (Table S5). To confirm that no mutations were introduced during selection of positive clones, whole genomic DNA was isolated

from the transformed spirochetes using the QIAamp DNA Mini kit (Qiagen, Hilden, Germany). The purified DNA containing the shuttle vectors were used for transformation of *E. coli* NEB 5-alpha cells (New England Biolabs, Frankfurt, Germany) and plasmids purified from selected clones were sequenced as mentioned above.

## Gene expression analyses

The RNA from spirochetes grown at mid-logarithmic phase was extracted by using the RNAprotect Bacteria Reagent and the RNeasy Mini Kit (Qiagen, Hilden, Germany) according to the manufacturer´s instructions. Thereafter, the purified RNA was treated twice with RNAse-free DNase, and proteins were removed by applying the Monarch RNA Cleanup kit (New England Biolabs, Frankfurt, Germany). The cDNA synthesis of the isolated RNA (1 μg) was then performed by using the LunaScript RT Supermix Kit (New England Biolabs, Frankfurt, Germany) following qPCR with 50 ng of cDNA and 10 μM of appropriate oligonucleotides (Table S5) according to the manufacturer´s instructions. Briefly, reactions were incubated initially at 95 °C for 60 s followed by 40 to 45 cycles of 95 °C (15 s), 60 °C (30 s), and 60 to 95 °C on a Roche Lightcycler 480 (Roche Diagnostics, Rotkreuz, Switzerland). Results were then analyzed using LinregPCR software[81], and the relative values of the genes of interest (*chiA, chiB, chiC, chiD, chiE, cihC, hcpA, flaB*, 16S rRNA) were compared to the values of the control sample (cDNA synthesized from spirochetes carrying the empty shuttle vector) by employing the $2^{\Delta Ct}$-method.

## Serum protection assay

Protection from complement-mediated lysis mediated by recombinant proteins was assessed by pre-incubation of 25 μl NHS with Chi proteins, BGA66 of *B. bavariensis* PBi[13] or BSA (10 μM each) for 15 min at 37 °C with gentle agitation. The pre-incubated serum samples were then adjusted to 100 μl with BSK-H medium. As additional reaction mixtures, native NHS (not pre-treated), heat-inactivated NHS and a Tris/HCl-buffer control were also included. In parallel, $1 \times 10^7$ spirochetes of serum-sensitive *B. garinii* G1 were sedimented by centrifugation and resuspended in either the pre-treated serum samples or the controls. All reaction mixtures were then incubated for 4 h at 37 °C with gentle agitation. The percentage of motile and viable cells was determined by dark field microscopy after 4 h, respectively. Spirochetes in nine microscopy fields were counted by using Glasstic slides 10 (KOVA International Inc., CA, USA). At least three independent biological replicates were performed and ±SEM was determined by using GraphPad Prism version 7.

## Serum bactericidal assay

Spirochetes grown at mid-logarithmic phase were sediment by centrifugation and resuspended in 500 μl BSK-H medium. Reaction mixtures consisting of 75 μl highly viable spirochetes ($1 \times 10^7$) and 25 μl of NHS (30%) were incubated at 37 °C with gentle agitation. The percentage of motile cells was determined as described above.

## In situ proteinase K treatment and immunofluorescence microscopy

To obtain complement-susceptible *B. recurrentis* A17 cells, a protease accessibility assay was performed. Spirochetes ($6 \times 10^6$ cells) were incubated with or without proteinase K (200 μg/ml) for 40 min at RT and the proteolytic activity was then terminated by adding Pefabloc SC (3 mM). Following sedimentation, cells were carefully washed twice with PBS and resuspended in 100 μl PBS containing 1% BSA (PBSA). Proteinase K-treated spirochetes were either lysed by sonication for Western blot analysis or incubated for 30 min at 33 °C with either 50 μl NHS or 50 μl hiNHS. After sedimentation, spirochetes were diluted 1:20 in PBSA and aliquots of 12 μl were spotted on diagnostic slides (Waldemar Knittel Glasbearbeitungs GmbH, Braunschweig, Germany).

Slides were allowed to air-dried overnight and thereafter incubated for 10 min at RT with 40% glyoxal solution. After fixation, slides were incubated for 1 h at 33 °C in a humidified chamber with either a polyclonal anti-C3 antibody (dilution of 1:1000) or a monoclonal C5b-9 antibody (dilution of 1:50), respectively. Following four washes with PBS, the slides were incubated for 1 h at 33 °C with 1:2000 dilutions of Alexa 488-conjugated secondary antibodies (Life Technologies, Carlsbad, CA, USA). To stain *Borrelia* DNA, slides were washed four times with PBS and incubated with 40 μl of a DAPI solution (2 μg/ml) for 10 min at 4 °C. After mounting with fluorescence mounting medium (Dako), complement components deposited on the spirochetal surface were visualized by using an Axio Imager M2 fluorescence microscope (Zeiss, Oberkochen, Germany) equipped with a Spot RT3 camera (Visitron Systems, Puchheim, Germany). Fluorescent images were acquired by using 63x objective (Zeiss Plan-Apochromat) and processed and analyzed using Visiview (Visitron Systems GmbH, Puchheim, Germany).

## Statistical analysis

Statistical analyses were performed using one-way ANOVA followed by Bonferroni's post hoc test, using GraphPad Prism (GraphPad Software, San Diego, CA, USA). A *p* value of <0.05 was considered statistically significant. Significance levels are indicated as follows: *$p < 0.05$, **$p < 0.01$, ***$p < 0.001$, and ****$p < 0.0001$.

## Ethics declarations

Collection of blood samples and consent documents was approved by the ethics committee at the University Hospital of Frankfurt (control numbers 160/10 and 222/14), Goethe University of Frankfurt am Main. All healthy blood donors provided written informed consent in accordance with the Declaration of Helsinki.

## Reporting summary

Further information on research design is available in the Nature Portfolio Reporting Summary linked to this article.

## Data availability

ChiA and ChiB structures are deposited in the Protein Data Bank under accession codes 28LI (ChiA) and 28LK (ChiB). Sequences of *B. recurrentis* strains related to this article are available via NCBI GenBank under the following accession codes: A1 (NC_011244); A11 (NZ_CP169977); A17 (NZ_CP169973); PAbJ (NZ_CP169965.1); PAbN (NZ_CP169954.1); PBek (NZ_CP169961.1); PMaC (SRX2631581); PUfA (SRX2631582). Additional genomes used in this study are available under the following accession codes: *B. duttonii* Ly (CP000979.1); *B. hermsii* HS1 (NZ_CP014350.1); *B. miyamotoi* CA17-2241 (CP021873.1); *B. miyamotoi* LB-2001 (NC_022079.2). All other data supporting the findings of this study are available within the paper and its Supplementary Information. Source data are provided with this paper.

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

## Acknowledgements

The authors gratefully acknowledge the skillful and excellent technical assistance of Martyna Olesiuk. We are also indebted to Prof. Dr. Reinhard Wallich (emer.) and Dr. Michael Kirschfink (emer.), Institute of Immunology, University Hospital of Heidelberg, Germany, who kindly provided Chi expression vectors and sheep erythrocytes. We also thank Sally Cutler, School of Health and Bioscience, University of East London, Stratford, London, UK, for providing *Borrelia recurrentis* A17. In particular, we are very grateful to Ilme Schlichting for the continued support over many years. Diffraction data were collected at beamline X10SA, Swiss Light Source, Paul Scherrer Institute, Villigen, Switzerland, and the authors thank the beamline staff for the excellent setup. Furthermore,

we thank Wolfgang Kabsch for his support in processing the difficult ChiA-data. This work forms part of the doctoral thesis of F.R.O., E.G., M.A., and F.R.E. This work was supported by the LOEWE Center DRUID (Novel Drug Targets against Poverty-Related and Neglected Tropical Infectious Diseases), LOEWE/1/10/519/03/03.001(0016)/53, project C3 (P.K.) and E3 (J.M.P., K.B., and S.R.).

## Author contributions

Conceptualization (P.K., K.F.W. and K.B.); Data curation (F.R., F.R.E., E.G., N.D., T.G.S., P.K. and K.F.W.); Formal analysis (P.K. and K.F.W.); Funding acquisition (P.K., S.R., and K.B.); Investigation (F.R., F.R.E., E.G., M.A., F.R.E., N.D., P.K. and K.F.W.); Methodology (P.K., F.R.E., K.F.W., M.S., N.D. and C.M.R.); Project administration (P.K.); Resources (V.F., J.M.P., K.B., P.K. and K.F.W.); Software (K.F.W. and C.M.R.); Supervision (P.K. and K.F.W.); Validation (F.R., F.R.E., P.K. and K.F.W.); Visualization (F.R., T.G.S., F.R.E., N.D., P.K. and K.F.W.); Writing—original draft (P.K. and K.F.W.); Writing—review & editing (P.K., K.F.W., F.R., E.G., M.A., F.R.E., N.D., T.G.S., V.F., C.M.R., M.S., K.B., S.R. and J.M.P.).

## Funding

## Competing interests

The authors declare no competing interests.
