## [Transparent Peer Review file · Nature Communications]

Complement inhibition by a unique cluster of immunomodulatory outer surface proteins of *Borrelia recurrentis*

Corresponding Author: Dr Karin Fritz-Wolf

Version 0:

Reviewer comments:

Reviewer #1

(Remarks to the Author)

Rottgerding and others have provided extensive biochemical and structural analysis of a unique cluster of immunomodulatory outer surface proteins of *Borrelia recurrentis* (Br), an understudied pathogen that causes louse-borne relapsing fever with a high incidence of mortality in resource limited regions of Africa. The authors have primarily relied on biochemical and structural analysis of a new cluster of proteins encoded on 190kb megaplasmid of Br downstream of previously identified complement-targeting proteins HcpA and CihC. The new cluster comprising of ChiA to ChiE (Complement inhibiting and Host interacting proteins) were shown to inhibit complement activation and MAC assembly; activation of complement components of AP by all homologs and that CP and LP was also activated by ChiE. Chi proteins were also shown to terminate the activation of TP and inhibit C9 polymerization with demonstrated binding to different complement components. Chi proteins were also shown to bind to plasminogen with the structure of ChiA and ChiB determined to have distinctive features ChiA having two cysteines with an ability to coordinate Mg ion. Overall structural analysis help divide Chi proteins into two groups with group 1 comprising of ChiB, ChiC and ChiD and group 2 consists of ChiA and ChiE with other structural features revealing binding to plasminogen except for ChiC. Additional comparative analysis with LD borreliae proteins also revealed homology to PFam54 -60 paralogous gene family among others. Serum-sensitive *B. garinii* cells expressing one or more Chi proteins to be resistant to complement except for ChiE which the authors attribute to misfolding. A large array of data sets corroborating the key findings have also been provided as part of the supplementary information. While biochemical and structural studies help advance the role of Chi proteins in providing resistance to complement, these studies lack the biological and physiological relevance as to the how one or more Chi proteins in Br confer resistance and whether there is a regulated expression one or more Chi proteins to respond the complement during the different stages of the infectious cycle of these spirochetes.

A major limitation of these studies is lack of the role of these proteins being assessed under conditions that are physiological or biological that limits the relevance of this study. The list of major concerns are as below

1. There were no studies reported directly establishing the relevance of Chi proteins in protecting Br from being inactivated by compliment. While ectopic studies in *B. garinii* offers select advantages, the levels of protein expression all Chi proteins presumably at equimolar or non-equimolar levels as would be expected in Br was not established. While technically it is a challenge to express all 5 proteins in Bg as in Br, the authors could exploit Br strains that have been compared to determine if all Chi proteins are expressed at the same time or in the presence of different signals as would be expected in the vector and vertebrate hosts.

2. Do all 5 Chi proteins regulated by different promoters responds to external signals in a similar fashion or is there a common signal that drive all of them or specifically one of the 5 Chi proteins?. Since there is a sequence correlation to PFam54 -60 paralogous gene family which are known to encode for genes that are differentially expressed in LD spirochetes, would there be a significant difference in the levels of expression of Chi proteins in Br. What is the likelihood of only ChiE being expressed relatively more and compensate for other Chi proteins functionally similar to the proteins of PFam54 -60 paralogous gene family that are known to compensate in mutants lacking one or more proteins. The functional significance of Chi proteins at physiological level is a key deficiency that need to be addressed. Are the Chi proteins expressed on the surface of the Br and if so does enzymatic removal of surface exposed proteins enhance complement sensitivity of Br strains?

3. Are there any amino acid changes between Chi proteins in each of the Br strains that may also provide information as to the functional significance of these proteins?. The significance of the lack of proteolytic activity of Chi proteins to degrade

complement C3 and C3b has not been discussed or brought into relevance in the context of effects of Chi proteins in providing resistance to complement.

4. The physiological relevance of structural differences of Chi proteins compared to other known spirochetal proteins that confer pathogen survival has not been discussed to leverage the major findings of this study. Does ROS affect complement sensitivity of Br due to redox sensitive changes in ChiC and ChiE proteins?.

5. What are the unique structural features of ChiC and ChiE that are different from those spirochetal proteins that interact with complement and how do these proteins alter interaction of Br with the host (other than complement and plasminogen binding). These aspects of the study can be expanded in the discussion part of the manuscript to focus the structural differences which is the mainstay of this study.

Reviewer #2

(Remarks to the Author)

This manuscript describes the functions and structures of five proteins from *Borrelia recurrentis*, designated ChiA to ChiE. *Borrelia recurrentis* is the causative agent of louse-borne relapsing fever, a poverty-associated infectious disease with high mortality. The topic of this work is important due to its relevance to human health. The proteins ChiA to ChiE have not been previously characterized, and the authors propose that these proteins contribute to the pathogen's virulence by functioning as immune evasion factors.

The manuscript addresses significant scientific questions and presents novel, original and numerous results. However, the overall organization of the manuscript is somewhat disordered. The authors seem to assume that all readers are advanced experts in spirochete biology. While this assumption could be appropriate for a specialized journal, it can hinder comprehension for a broader scientific audience. For instance, the Introduction lacks a few essential sentences on spirochetal lipoproteins, which are central to the manuscript's focus.

Also, the positive and negative control proteins/compounds used in all assays should be clearly characterized.

The Results section is also disorganized, making the paper difficult to follow. Reordering the subsections—placing structural findings before those on function—could enhance clarity. A similar lack of organization is present in the Materials and Methods section.

In this study, the ChiA to ChiE proteins were functionally characterized using various *in vitro* assays. The structural analyses combined crystallographic methods with *in silico* modeling. Both approaches yield interesting, novel, and potentially valuable results. However, the quality of some data and interpretations is not entirely convincing.

Detailed comments:

The introduction is scarce and does not cover the content of the publication. The authors supplement the information and introduce new names, phenomena and processes in the results or discussion sections e.g.:

Line 92, 93 Structurally diverse proteins of Lyme disease (LD) and relapsing fever (RF) borreliae contribute to the inhibition of the terminal pathway (TP), this information should be expanded and moved to the introduction, together with the information about CspA (line 96)

Complement is not sufficiently described as well, as there are binding studies on FB, FH, FI and C3b (line 107) etc., which are not mentioned in the introduction. It would be also advisable to have a comprehensive graphical diagram illustrating e.g. the complement system in the introduction.

Vsp1 (line 121), uPA (line 125) are not introduced before

Types of CP inhibitory mode of complement interacting proteins, opsonization from the discussion should be also be in the introduction part.

Line 73. Identification of a unique gene cluster on lp190 of *Borrelia recurrentis*

This section, along with the associated figures and tables, should include a comparison of sequence similarity/identity between the corresponding proteins in *B. recurrentis* and *B. duttonii*. Some Chi protein sequences for example from *B. duttonii* GenBank entry (CP000979) share over 99% identity with presented *Borrelia recurrentis* ChiA to ChiE. This supports that both *Borrelia* species utilize similar immune evasion mechanisms.

Line 84: Assessment of the inhibitory capacity of Chi proteins on complement activation and MAC assembly. Brief explanation of the assay's principle and mechanism would greatly aid reader understanding.

Line 88, Figure 1 and corresponding Materials and Methods section (line 526). The amount of ChiA to ChiE used in the assays is not clearly stated. Please clarify whether the reported concentration refers to the final concentration in the assay or to a stock concentration that was subsequently mixed with NHS. Please provide a clearer and more detailed description of the negative and positive controls. Figure 1 – Figure 1b includes the data from Figure 1a. One of Figure 1 panels could be placed in the Supplementary data.

Line 92. Inhibition of terminal pathway. Brief explanation of the assay's principle and mechanism would greatly aid reader understanding.

Figure 2, Results and corresponding Materials and Methods section (line 538). Please explain mentioning vitronectin. The amount of ChiA to ChiE used in the assays is not clearly stated. Please clarify whether the reported concentration refers to the final concentration in the assay or to a stock concentration that was subsequently mixed with complement compounds. Please clarify it. Please provide a clearer and more detailed description of the negative and positive controls.

Figure 2 B-H – please use the same format for protein concentration. It is difficult to determine how 0.5-10ug of tested proteins relates to the molar concentrations used in Figure 2A. Please use consistent concentration format throughout the manuscript.

Line 106. "The very low absorbance values suggested that none of the Chi homologs interacted with FB..." and Figure S2. Authors inconsistently interpret the values of absorbance in the context of binding. In Figure S2A, absorbance value for ChiA is ~0.3 and is interpreted as fine binding, while similar result in Figure S2B for a few Chi proteins means no binding, and in Figure S2G means strong binding (BSA levels are comparable at these panels, A490 is ~0.1-0.2). This interpretation decreases the reliability of the presented results. Description of control proteins is missing.

Line 112 and corresponding S3J-K Figure. It is difficult to determine how 5ng/ul of the protein in the Figure S2 relates to the molar amount of proteins used in S3 Figure. Figure S3F-H: ChiD binding to C5 was considered insignificant in the previous Figure S2, but in Figure S3 is significant and has the same level as ChiE and ChiB. Lack of consistent results. Why there is increasing signal for controls with BSA in Figure S3J-K?

Line 115. According to Figure S3, the saturation is not reached at most cases except for S3A,B,C,D.

Line 115: "...binding is evident C1q, C4, C4b and C5..." According to the Figure S3J-K it is not evident, rather it is unspecific binding due to the high concentrations of proteins applied. Why the concentrations up to 200nM were used only in Figure S3J-K but not in the other assays from Figure S3? What would be the results then?

Line 117. Interaction of Chi proteins with plasminogen. Please provide a more detailed description of the method used for KD calculation in the Materials and Methods section. It is only briefly mentioned in the Figure 3 legend.

Line 125. Conversion of plasminogen to the active serine protease plasmin and Figure S4. Are the values of A405 averaged if the experiments shown in the Figure S4 were performed in duplicate? Authors mention that "At least three independent experiments were conducted, each in triplicate". What are the results? The proteolytic activity upon binding to Chi proteins could be described as, for example % of protease activity of positive controls after 24h, in a supplementary table.

Line 134. Please describe more the role of lysines and tranexamic acid in the assay with plasminogen.

Figure S4, Why in the assays with ChiA and ChiC, the proteolytic activity in the presence of TXA is almost as high as in its absence? Why in the assays with ChiA and ChiC, the proteolytic activity in the absence of plasminogen is much higher than corresponding proteolytic activity in the absence of plasminogen in the rest of assays from on Figure?

Line 135 and Fig3G – the results from this figure must be commented more comprehensively. Why results for ChiA in Figure 3G show strong plasmin activity as compared to the rest of Chi proteins, while the previous results for ChiA in Figure 3 and Figure S4 indicated something opposite?

Line 141: I do not understand the statement "The lack of alphafold model" as AlphaFold2 can predict structures never previously seen in the PDB, i.e. novel protein folds (Bordin et al., 2023; Barrio-Hernandez et al., 2023; Durairaj et al., 2023). Therefore, it is unclear why authors have not attempted (at least not described) to generate AlphaFold2 predictions from sequence and use this model for molecular replacement in the first place instead of the homology model with such a low identity. The authors should firstly describe the outcome of an attempt of use alphafold predicted model for molecular replacement, before anomalous measurement, autoSol and autobuild stages. Also, in my opinion reviewers should have access to both the MTZ file (containing structure factor amplitudes and phases) and the PDB file (containing the atomic coordinates of the refined model). These files are essential for assessing the quality and validity of the reported structure, especially, when deposit structure release is hold for publication.

In view of the very high proportion of RSRZ outliers in both structures, it is not clear whether this is due to the lack of electron density for the amino acid side chains or to their poor modeling in electron density. This issue is not addressed in the text and also could not be checked by reviewers in the structural data.

Line 180. Functional properties and redox regulation of Chi proteins... "All Chi proteins, except ChiC, bind to plasminogen"... In the previous sections the authors showed that ChiC binds plasminogen and assessed Kd = 179nM.

This section and related Figures contain mostly speculations, should be shortened, edited and placed in the discussion.

Line 210 and Figure S8. ~9kDa product of digestion is out of range of presented SDS-PAGE picture.

Figure S8. ChiA lane contains double band for ChiA. Could this influence the results in the other presented experiments like, for example, problems with assessment of Kd for plasminogen binding?

Line 215. Structural comparison with surface proteins of LD borreliae. This section and related Figures contain mostly speculations, should be shortened, edited and placed in the discussion.

Line 238. Selected residues potentially involved in the host-protein interaction – it is not specified on what basis these residues were selected

Lines 250 to 273. Section: Chi homologs protect spirochetes from complement-mediated killing and confer serum resistance. This section presents really important findings on the biological relevance of the ChiA–E proteins. However, the results are described too casually and in a disorganized manner. The reader should be clearly informed about which *Borrelia* strain was used in each specific experiment.

Line 260. RNA expression not always means that the encoded protein is produced. Please be more careful when drawing conclusions and describe your results more precisely.

Line 264. Please comment on the fact that the bands for the same proteins, CihC and HcpA, produced in different cells, show different molecular weights in Figure S12B

Lines 276-290. This section of the discussion includes overly long descriptions of the results. Please shorten these summaries and focus more on interpretation and implications.

Line 302. "...antibody responses to at least ChiB and ChiD have been detected in samples from LBRF"... Please provide more information helping the reader to identify the discussed literature data on ChiB and ChiD, concerning citation 40. For example, include in brackets the corresponding ORF names or other identifiers for ChiB and ChiD as described in reference 40.

Line 356 – Was the serine protease activity of the Chi proteins was tested e.g. with another substrates eg. available commercially and based on fluorophores, to investigate if they are active ?

Line 370 – This notion is not fully supported by the results from the Figure S4.

Minor comments:

Line 26, 66: 2.71Å is medium resolution structure, whereas 1.5Å is indeed a high resolution structure, maybe it would be better to write “X-ray structures of ChiA and ChiB”

Line 58: CihC and HcpA, have been described so far in *B. recurrentis* as complement targeting proteins that bind C1-Inh, C4BP, and FH, respectively – what exactly bind what?

Line 333: probably “as” not “than”, the whole sentence in the line 333 is also hard to understand

Line 62: there is “chiC gene” which is probably a mistake as the gene corresponds to CihC protein, the same mistake is in the 263 line

Line 24-25: Sentence “Borrelia proteins protect susceptible spirochetes from complement-mediated killing and three molecules facilitate serum resistance” should be edited. The authors should name the proteins and molecules.

Figure S10. The Figure legend needs correction.

Reviewer #3

(Remarks to the Author)

Version 1:

Reviewer comments:

Reviewer #2

(Remarks to the Author)

1. The introduction has been significantly improved. The rest of the manuscript has been somewhat improved, but is still difficult to read, as mentioned in the first version of this review.
2. Panels A–C of Figure 1 should be moved to the Supplementary Data, as they duplicate content already shown in panels D–F. This recommendation was also made in the first review.
3. Figure 6 was not improved (the format of concentration units in the Figure was not unified).
4. Supplementary Figure 8 was not improved and shows inconsistent data. The controls are not clearly described. For example, panel S.Fig.8.C – BtcA marked as significant. The authors write that “At the time when these assays were conducted, no control protein of bacterial origin was available that interact with FB, C1q, C4, and C4b, respectively”. Perhaps better controls are now available. The results presented in S. Figure 8 are main basis of Figure 9 which summarizes the proposed role of the studied proteins. In the light of the above, Figure 9 is not very credible.
5. Supplementary Figure 9 was not improved. Supplementary Figure 9 G – data on ChiD binding to C5 are still inconsistent with the Supplementary Figure S8. The authors respond that they have no explanation for this observation. In my opinion, such inexplicable and inconsistent results should not be published. Both the tested proteins and BSA appear to bind to complement components nonspecifically. The results presented in Supplementary Figure S9 also form the main basis of Figure 9, which summarizes the proposed role of the tested proteins. In the light of the above, Figure 9 is not very credible.
6. Supplementary Figure S11 and measurements in replicas. Although an Excel table with the raw data was included, the standard deviations in this table are still missing.
7. Supplementary Figure S11 – the authors did not respond the question “Why in the assays with ChiA and ChiC, the proteolytic activity in the presence of TXA is almost as high as in its absence?” The authors still claim that: “Our new generated data revealed that TXA strongly influenced binding of plasminogen to ChiA, ChiB, and ChiD, and to some extent also to ChiC, and ChiE indicating that lysines do play a role in binding of plasminogen” and “Proteolytic activity of plasminogen was impaired when ChiB, ChiD, and ChiE was assayed indicating that lysines are involved in binding to all five Chi proteins “. In the discussion: “Mechanistically, the interaction of plasminogen with bacterial ligands is often mediated by lysines as also observed for all Chi homologs (Fig. 7A and S11)”. The results confirming these claims are missing. The Supplementary Figure 11 shows something opposite. The difference in signal recorded with and without TXA for controls is clear while there is no difference in signal for ChiA, ChiB and really weak difference for ChiD, so lysines are not involved in binding.

Points 4-7 point out some really serious shortcomings of this manuscript.

8. The authors still claim that: “The ChiB structure was solved in 2018, at a time when AlphaFold2 had not yet been released. The first AlphaFold2 release appeared at the end of 2020, and public access to the prediction server became available only in mid-2021. Because of the low sequence identity to known structures, the homology model used for molecular replacement was not sufficient to solve the structure, so we performed experimental phasing using anomalous measurements, followed by AutoSol and AutoBuild. We fully agree that an AlphaFold2-based approach would have greatly facilitated structure determination, and we would have gladly used this tool had it been available at the time.”

Although the authors clarified the rationale behind their structure determination approach, they did not address the reviewer’s suggestion to use the AlphaFold model for MR. Given that it is now 2025, the authors should perform molecular replacement on the 2018 dataset using AlphaFold-generated model to assess whether the experimental model differs from the predicted one and, if so, in what ways. They should also articulate the possible added value provided by the experimental model. Readers are not aware that the structure was originally determined in 2018, and this inconsistency in the structural strategy is immediately noticeable. Moreover, refining predicted models using experimental data is now an

established practice.

9. The authors still claim that: "We agree with the reviewer that access to both the MTZ files and the PDB files is important for evaluating the quality of the reported structures, especially when the public release is on hold until publication. The ChiA and ChiB structures have been deposited in the Protein Data Bank under accession codes 9FSD and 9FQQ, respectively, including the corresponding MTZ files. The entries are currently on hold due to ongoing competitive work in the Borrelia field. We would have no objection to making these files available to the reviewers. However, the validation reports generated by the PDB are generally sufficient to assess the quality of the deposited structures. In the ChiB structure (1.5 Å), only a few RSRZ outliers (18 residues out of 284) were detected. These are mainly located in surface regions with weak side-chain density, particularly in the N-terminal helix, which is consistent with the higher local B-factors.

The ChiA structure (2.7 Å) crystallized in the monoclinic space group C2 with two monomers in the asymmetric unit connected by a disulfide bond. As ChiA is rapidly oxidized, crystallization was challenging; well-diffracting crystals were obtained only after seeding and in the absence of reducing agents. Minor flexibility between the monomers likely contributed to the moderate resolution. Space-group determination was initially ambiguous but confirmed unambiguously during final refinement. The data were processed with XDS, and quality was assessed with xtriage (Phenix). The relatively high Wilson B-factor (63.5; Supplementary Table 2) indicates increased overall flexibility, consistent with the higher number of RSRZ outliers (97 residues out of 530). These residues are mainly located in the N-terminal helix, the S-domain, and the connecting loops between helices. Nevertheless, the overall model is well defined, supported by an R_{free} of 30% and its close agreement with the AlphaFold2 prediction."

I thought that this journal requires all raw data discussed in the manuscript to be available for review. The validation report for the ChiA structure indicates a model of suboptimal quality in terms of side-chain geometry and bond angles. I understand that the data are not ideal and that the structure displays substantial flexibility, but under such circumstances tightening the geometric restraints appears to be a reasonable approach to improve the model. Moreover such a high overall B-factor of the structure may just as well indicate an error in data processing or refinement. The validation report itself rather suggests that provided structure needs significant improvement.

Reviewer #3

(Remarks to the Author)

Reviewer #4

(Remarks to the Author)

I have uploaded the information requested from me as a separate document.

Version 2:

Reviewer comments:

Reviewer #4

(Remarks to the Author)

The authors have responded adequately to all of the comments raised in my prior review. I have no new comments on the revised submission.

We would like to thank the reviewers for their constructive and valuable comments, which have greatly helped us to improve our manuscript. We also appreciate the time and effort that the editorial team and reviewers have invested in our manuscript. We believe that the revisions have substantially enhanced the overall quality and readability of the manuscript. We hope that the revised version will now be suitable for publication in *Nature Communications*. Below, please find our detailed point-by-point response to the reviewers' comments.

Reviewer #1 (Remarks to the Author):

Rottgerding and others have provided extensive biochemical and structural analysis of a unique cluster of immunomodulatory outer surface proteins of *Borrelia recurrentis* (Br), an understudied pathogen that causes louse-borne relapsing fever with a high incidence of mortality in resource limited regions of Africa. The authors have primarily relied on biochemical and structural analysis of a new cluster of proteins encoded on 190kb megaplasmid of Br downstream of previously identified complement-targeting proteins HcpA and CihC. The new cluster comprising of ChiA to ChiE (Complement inhibiting and Host interacting proteins) were shown to inhibit complement activation and MAC assembly; activation of complement components of AP by all homologs and that CP and LP was also activated by ChiE. Chi proteins were also shown to terminate the activation of TP and inhibit C9 polymerization with demonstrated binding to different complement components. Chi proteins were also shown to bind to plasminogen with the structure of ChiA and ChiB determined to have distinctive features ChiA having two cysteines with an ability to coordinate Mg ion. Overall structural analysis help divide Chi proteins into two groups with group 1 comprising of ChiB, ChiC and ChiD and group 2 consists of ChiA and ChiE with other structural features revealing binding to plasminogen except for ChiC. Additional comparative analysis with LD borreliae proteins also revealed homology to PFam54 -60 paralogous gene family among others. Serum-sensitive *B. garinii* cells expressing one or more Chi proteins to be resistant to complement except for ChiE which the authors attribute to misfolding. A large array of data sets corroborating the key findings have also been provided as part of the supplementary information. While biochemical and structural studies help advance the role of Chi proteins in providing resistance to complement, these studies lack the biological and physiological relevance as to the how one or more Chi proteins in Br confer resistance and whether there is a regulated expression one or more Chi proteins to respond the complement during the different stages of the infectious cycle of these spirochetes. A major limitation of these studies is lack of the role of these proteins being assessed under conditions that are physiological or biological that limits the relevance of this study. The list of major concerns are as below

1. There were no studies reported directly establishing the relevance of Chi proteins in protecting Br from being inactivated by complement. While ectopic studies in *B. garinii* offers select advantages, the levels of protein expression all Chi proteins presumably at equimolar or non-equimolar levels as would be expected in Br was not established. While technically it is a challenge to express all 5 proteins in Bg as in Br, the authors could exploit Br strains that have been compared to determine if all Chi proteins are expressed at the same time or in the presence of different signals as would be expected in the vector and vertebrate hosts.

We thank the reviewer for his/her important remark regarding the relevance of the Chi proteins *in vivo*. We are aware that we currently cannot entirely solve this issue satisfactorily due to technical limitations and missing infection models as well as genetically tools established for *B. recurrentis*. The very restricted infectious cycle of *B. recurrentis* in which the body louse acts as only verified

vector and humans as the only host, makes such *in vivo* studies almost impossible due to, for example, ethical concerns. A recent study investigating different mice strains for *Borrelia* infection clearly showed that only CC046 mice could be experimentally infected with *in vitro* grown spirochetes but instead of multiple relapses, only a single spirochetemia could be observed in the infected mice exhibiting low titers of spirochetes in plasma (between 4.0×10^4 to 4.0×10^6 /ml) (Rogovskyy et al., 2021, Infect. Immun. 89:e00048, doi: 10.1128/IAI.00048-21). To the best of our knowledge, no *Pediculus humanus corporis* model has been established so far to conducted gene expression analyses of *B. recurrentis* in the vector. As suggested by the reviewer, we sought to get more information about potential signals influencing gene expression of Chi proteins under different *in vitro* condition. As already done extensively for LD spirochetes in the last decade to analyze gene expression under different external signals including temperature, *B. recurrentis* A17 was grown at room temperature and elevated temperatures (35 °C and 37 °C). We also investigate temperature shifts (35 °C → 33 °C and 35 °C → 37 °C) and collected cell lysates after the cell population expanded to collected sufficient material for Western blotting. Cell lysates were than used for the analyses of Chi, HcpA, and ChiB as described in **Supplementary figure S14**. As shown in the figure below, all three proteins were detectable independent of the temperature indicating that these molecules were continuously produced by *in vitro* grown spirochetes. In addition to temperature, other signals also did not impact production of these three proteins when *B. recurrentis* was grown in the presence of two different culture media supplemented with either rabbit or human serum (supplementary figure **S14B**).

Influence of different growth temperatures on the *in vitro* synthesis of selected complement-targeting proteins by *B. recurrentis* A17. Spirochetes were grown in BSK-H medium at each of the indicated temperatures and whole-cell lysates were generated. Detection of ChiC (A), HcpA (B), and ChiB (C) by Western blot analyses. Lysates (20 µg each) as well as purified ChiC, HcpA, and ChiB (500 ng each), respectively, were subjected to Tris/tricine-SDS-PAGE and transferred to nitrocellulose. Proteins were then detected by applying a mAb directed to CihC that also recognizes ChiC-homologs FbpA (45.8 kDa) and FbpB (46.9 kDa), a mAb directed to HcpA, and a polyclonal rabbit anti-ChiB antibody directed to ChiB, respectively. Molecular mass standards (M) are shown at the left and the respective borrelial proteins are indicated to the right of the arrow.

As also mentioned by this reviewer, we cannot completely rule out that each individual *chi* gene could differentially be expressed *in vivo* either in the vector or the human host. Nevertheless, our analyses revealed that each gene is expressed in the surrogate strain as well in the WT *B. recurrentis* strain showing that the gain-of-function strains showed similar expression levels. As also mentioned

by this reviewer that gene expression does not necessarily correlate with the production of the respective protein in the WT or the genetically modified strains as suggested for ChiE (please see our responses below). In addition, appropriate tools for manipulating *B. recurrentis* to selectively knock out individual chi genes are not at the horizon yet, so we try to overcome this obvious limitation by using a well-established, highly serum susceptible *B. garinii* surrogate to collect reliable *in vitro* data as possible to underscore the role of this cluster of homologous proteins for complement resistance.

2. Do all 5 Chi proteins regulated by different promoters responds to external signals in a similar fashion or is there a common signal that drive all of them or specifically one of the 5 Chi proteins?

We scan the non-coding regions of each *chi* gene for canonical promotor elements using the YAPP eukaryotic core promotor predictor searching for TATA boxes, initiators, and for putative synergistic combinations. The expectation is that synergistic combinations are less like to occur by chance than single elements, and that two or more weak elements can combine to make a functional promoter. The program is created for scanning sequences of putative determined promoter regions in eukaryotic but also for prokaryotic promoter sequences. As shown below, putative promoter sequences could be identified in the upstream region of all five *chi* genes. If sequence variation within these regions might account for a differential gene expression as described for other bacterial promoters appears to be a matter of speculation as there are no information available yet from *B. recurrentis* or other Old World relapsing fever spirochetes. It has been known that the precise sequences and spacing between these elements can vary slightly between different bacterial species or even between different promoters within the same bacterium.

ChiA gene:

5' **taa**agatgtttaataaatgatataaaaagctaataaaatatattgatatctagg**tcaaatt**aatattaactatatgtaataatgattattaa
ttaataacagcaacaatttgaataaattgtttattttaataataaggagagatt**ttg**-3'

ChiB gene:

5' **tag**taagaatttatgttgattatgaatcaattaaatataattataaataactattggattaaacatgactgaggaaaagtttcccttggtgatcgat
aattag**cag**ttttcaat**ttg**-3'

Core Promoter Element Matches

Motif	Pos	Score	Seq	TSS
INR	100	0.95	TCAGTTT	102
TATA	37	0.94	TTTATAAATACT	66
TATA	29	0.81	TAAATATATTTA	58
DPE	98	0.91	AGTCA	70

Synergistic Combination Matches

Motif	Pos	Seq	Motif	Pos	Seq	Combined Score	TSS
TATA	37	TTTATAAATACT	DPE	98	AGTCA	1.85	70

ChiC gene:

5' **taa**aagggttaatttggattattaatcaattaaaatactattaattaataataccaagaaaaatttcttggttatcaataatt**agtt**agttt
ataattagataatgaggagaatatt**ttg**-3'

Core Promoter Element Matches

Motif	Pos	Score	Seq	TSS
INR	22	0.94	TCAATTT	24
INR	87	0.82	TTAGTTT	89
INR	5	0.82	TTAATTT	7
TATA	24	0.85	AATTTAAAAATA	53
TATA	44	0.84	TAAATATAACCA	73
DPE	100	0.93	AGATA	72
DPE	85	0.88	AGTTA	57
DPE	71	0.87	GGTTA	43
DPE	3	0.87	GGTTA	-25

Synergistic Combination Matches

Motif	Pos	Seq	Motif	Pos	Seq	Combined Score	TSS
TATA	24	AATTTAAAAATA	DPE	85	AGTTA	1.73	57
TATA	44	TAAATATAACCA	DPE	100	AGATA	1.76	72

ChiD gene:

5' **taa**tgttaattataatcaattaatatatttaggacacttttaattaataataccaagaaaaatttctt**ggtt**atattaattagccagtt
tgtaattaatgataattgaataataaggaggatatt**tga**-3'

Core Promoter Element Matches

Motif	Pos	Score	Seq	TSS
INR	89	0.97	CCAGTTT	91
TATA	6	0.92	ATTATAAATCAA	35
TATA	45	0.84	TAAATATAACCA	74
TATA	36	0.82	CTTTTAAATTA	65
TATA	19	0.81	TAAATATATTTA	48
DPE	31	0.95	GGACA	3
DPE	125	0.92	GGATA	97
DPE	73	0.87	GGTTA	45

Synergistic Combination Matches

Motif	Pos	Seq	Motif	Pos	Seq	Combined Score	TSS
TATA	6	ATTATAAATCAA	DPE	73	GGTTA	1.79	45
TATA	19	TAAATATATTTA	DPE	73	GGTTA	1.68	45

ChiE gene:

5' **tag**gagattaatgttaattataaatcaattaatatatttaaactgttaattaataataacaaggtaagaaa**tacttggtt**atcgta
attagccagtttgaatt**agata**ataaggaagatatt**ttg**-3'

Core Promoter Element Matches					
Motif	Pos	Score	Seq	TSS	
INR	97	0.97	CCAGTTT	99	
INR	75	0.80	TACTTG	77	
TATA	14	0.92	ATTATAATCAA	43	
TATA	33	0.85	TATTTAAACAC	62	
TATA	53	0.85	TAAATATAACA	82	
TATA	27	0.81	TAAATATATTA	56	
DPE	110	0.93	AGATA	82	
DPE	122	0.93	AGATA	94	
DPE	81	0.87	GGTTA	53	

Synergistic Combination Matches							
Motif	Pos	Seq	Motif	Pos	Seq	Combined Score	TSS
TATA	14	ATTATAATCAA	DPE	81	GGTTA	1.79	53
TATA	27	TAAATATATTA	DPE	81	GGTTA	1.68	53
TATA	33	TATTTAAACAC	DPE	81	GGTTA	1.72	53
TATA	53	TAAATATAACA	DPE	110	AGATA	1.78	82
TATA	53	TAAATATAACA	INR	75	TACTTG	1.65	73

stop codon (red), synergistic region (green); start codon (light blue)

In addition, we also run Promotech (<https://github.com/BioinformaticsLabAtMUN/PromoTech>), a machine-learning-based method for promoter recognition (R. Chevez-Guardado & L. Peña-Castillo, 2021, <https://doi.org/10.1186/s13059-021-02514-9>) to predict potential promoters in the 5' upstream region of each *chi* gene. This species independent, bacterial promoter detection method was initially trained with data set of promoter sequences of nine distinct bacteria species including spirochetes (*Leptospira interrogans*). Of note, Promotech predicts promoters in the proximity of actual promoters but are unable to recognize the exact genomic location of an actual promoter. True positives predictions display score values that tend to be around 0.5 or higher by setting a default threshold of 0.5 which is recommend. As summarizes below, promoter sequences could be found in all five *chi* genes

ChiA gene:

	chrom	start	end	score	strand	sequence
0	sequenz-promotech-ChiA	8	47	0.62794	+	GTTTAAATAAATGATATAAAAAGCTAATAAAATATTATTGA
1	sequenz-promotech-ChiA	11	50	0.64037	-	ATATCAATATATTTTATTAGCTTTTATATCATTTATTTA
2	sequenz-promotech-ChiA	57	96	0.64770	-	AATTAATAATCATTATTAACATATAGTTAATATTAATTTG

ChiB gene:

	chrom	start	end	score	strand	sequence
0	sequenz-promotech-ChiB	11	50	0.61688	-	AGTATTTATAAATATATTTAATTGATTCATAATCAACATA

ChiC gene:

	chrom	start	end	score	strand	sequence
0	sequenz-promotech-ChiC	67	106	0.65374	+	TTCCTTGGTTATCAATAATTAGTTAGTTTATAATTAGATA
1	sequenz-promotech-ChiC	68	107	0.71418	+	TCCTTGGTTATCAATAATTAGTTAGTTTATAATTAGATAA

ChiD gene:

	chrom	start	end	score	strand	sequence
0	sequenz-promotech-ChiD	0	39	0.68232	-	AGTGCCTAAATATATTTAATTGATTATAATTAACATTA
1	sequenz-promotech-ChiD	1	40	0.69834	-	AAGTGCCTAAATATATTTAATTGATTATAATTAACATT
2	seauenz-promotech-ChiD	42	81	0.72711	-	TATAACCAAGAAAAATTTTCCTTGGTTATATTTAATTTA

ChiE gene:

	chrom	start	end	score	strand	sequence
0	sequenz-promotech-ChiE	6	45	0.66549	+	ATTAATGTTAATTATAAATCAATTAATATATTTAAACA
1	sequenz-promotech-ChiE	8	47	0.70451	-	AGTGTTTTAAATATATTTAATTGATTTATAATTAACATTA
2	sequenz-promotech-ChiE	9	48	0.68325	-	CAGTGTTTTAAATATATTTAATTGATTTATAATTAACATT
3	sequenz-promotech-ChiE	23	62	0.74323	+	ATCAATTAATATATTTAAACACTGTTAAATTAATATA
4	sequenz-promotech-ChiE	50	89	0.63862	-	GATAACCAAGTAATTTCTTACCTTGTTTATATTTAATTTA

Since there is a sequence correlation to PFam54 -60 paralogous gene family which are known to encode for genes that are differentially expressed in LD spirochetes, would there be a significant difference in the levels of expression of Chi proteins in Br.

As suggested, we cannot completely rule out that single or multiple genes within this cluster could be induced by certain external as yet unknown signals. We could not detect differences in gene expression under *in vitro* condition.

Regarding the organization of the PFam54 gene family on lp54 of LD spirochetes, it should be noted that this particular gene cluster on the megaplasmid lp190 in *B. recurrentis* is organized in a different manner, thus, data collected from previous studies performed with Lyme disease spirochetes should be viewed in caution and cannot be taken one to one (Ojaimi et al., 2003, *Infect. Immun.* 71:1689-1705, doi: 10.1128/IAI.71.4.1689-1705.2003; Tokarz et al., 2004, *Infect. Immun.* 72:5419-5432, doi: 10.1128/IAI.72.9.5419-5432.2004).

What is the likelihood of only ChiE being expressed relatively more and compensate for other Chi proteins functionally similar to the proteins of PFam54 -60 paralogous gene family that are known to compensate in mutants lacking one or more proteins.

This is an excellent point raised by the Reviewer. Whether an increased expression of ChiE compensates the function of at least ChiB, ChiC, and ChiD, respectively, all of which protect spirochetes from complement-mediated killing is somewhat speculative and is, to be honest, beyond the scope of the current study. As mentioned above, tools for genetic manipulations of *B. recurrentis* are not available so far to delete particular genes of interest for further functional analyses.

Also of note and already mentioned in the original manuscript, the high conservation of the *chi* gene cluster and adjacent genes (including *hcpA* and *cihC*) suggested that these genes are maintained by a strong selective pressure. The significant synteny of the gene cluster exhibiting a very low genetic diversity observed among *B. recurrentis* strains isolated from relapsing fever patients at different time points (1990 and 2015) and geographic regions might account for an extreme restricted pathogen-vector-host adaptation.

In addition, human pathogenic microorganisms like spirochetes developed means to compensate functions relevant for infectivity, pathogenesis or even immune evasion. This can be accomplished for example by gene duplication that could also be suspected for the *chi* genes. Regarding compensation of proteins functionality by other homologous proteins, we and others clearly demonstrated that the complement-inhibitory function of the key Factor H-binding protein CspA of *B. burgdorferi* could not be compensated by other, structural highly similar members of the PFam54 protein family located on lp54 of Lyme disease spirochetes (Brooks et al., 2005, *J. Immunol.* 175:3299-3308, doi: 10.4049/jimmunol.175.5.3299; Kenedy et al. 2009; *Infect. Immun.* 77:2773-2782, doi: 10.1128/IAI.00318-09). Despite some structural similarities with members of the Pfam54 protein family, Chi proteins largely differ with regard to their complement inhibitory function and could, in principle, compensate the complement-inhibitory function of other proteins of the cluster.

The functional significance of Chi proteins at physiological level is a key deficiency that need to be addressed.

We would like to kindly draw the reviewer's attention to our responses dealing with the physiological relevance of Chi proteins (see above).

Are the Chi proteins expressed on the surface of the Br and if so does enzymatic removal of surface exposed proteins enhance complement sensitivity of Br strains?

Each Chi protein carries a typical N-terminal domain structure which is composed of the signal sequence, a lipobox with a conserved cysteine residue (red), a sorting signal and a highly flexible tether:

These characteristics are in general attributed to *Borrelia* outer surface lipoproteins (Zückert, W.R., 2015, *Biochim Biophys Acta*, 1843:1509-1516, doi: 10.1016/j.bbamcr.2014.04.022) and, thus, it can be expected that Chi proteins might be exposed to the outer surface of *B. recurrentis* in a similar manner. In addition, we showed in our previous study, that at least ChiB and ChiD elicit an immune response during an infection caused by *B. recurrentis* (Röttgerding et al., *Front Cell Infect Microbiol.* 2022, 12:983770, doi: 10.3389/fcimb.2022.983770). These findings point to the location of Chi proteins to the outer leaflet of the borrelial membrane.

As suggested by reviewer #1, we performed an *in situ* protease accessibility assay to remove proteins exposed to the outer leaflet of the spirochetal membrane. Highly viable spirochetes of *B. recurrentis* A17 (6×10^6 cells/ μ l) were incubated with 200 μ g/ml (+ PK) or without proteinase K (- PK) for 40 min at RT. After inactivation of proteinase K with Pefabloc for 20 min at RT, cell lysates were generated which were then subjected to SDS-PAGE and further processed for silver staining (A) or Western blotting (B). Specific mAb directed to FlaB and CihC, respectively, or a polyclonal rabbit anti-ChiB Ab were employed to detect borrelial proteins or fragments thereof.

Surface exposition of native CihC and ChiB proteins. After proteinase K incubation, cell lysates were subjected to 10% T/T-SDS-PAGE and proteins were visualized by silver staining **(A)** and Western blotting **(B)**. For the detection of protease-susceptible complement-interacting proteins, CihC and ChiB were analyzed. As a control, the periplasmatic FlaB protein investigated. Selected proteins were detected by a monoclonal anti-FlaB Ab (1:100), the monoclonal anti-CihC Ab (1:10), and the polyclonal rabbit anti-ChiB Ab (1:1000), respectively.

Our analyses revealed that ChiB and CihC were highly susceptible to proteolytic degradation by proteinase K while the periplasmatic FlaB protein remains unaffected during treatment, indicating that both, ChiB and CihC, are exposed to the outer surface of *B. recurrentis* A17.

Having demonstrated proteolytic digestion of these lipoproteins, proteinase K-treated cells were incubated with 50% NHS or 50% hiNHS to analyze deposition of activated C3 and the formation of the MAC (C5b-9 complex) on the spirochetal surface by using immunofluorescence microscopy. As controls, wild type cells of *B. recurrentis* A17 were also treated with NHS and hiNHS, respectively, under identical conditions.

Shedding of outer surface proteins results in a strong complement activation following deposition of C3 **(A)** and formation of the MAC **(B)**. As expected, no complement deposition could be observed on spirochetes incubated with hiNHS. In addition, neither C3 nor MAC were detected on WT cells incubated with NHS indicating that these spirochetes resist complement activation. Almost all complement-affected cells are characterized by the formation of blebs as an indicator of cell destruction. These findings suggest that spirochetes lacking complement-protecting outer surface proteins are highly susceptible to complement-mediated lysis.

Deposition of activated complement components on proteinase K-treated spirochetes. Complement activation on proteinase K-treated and untreated *B. recurrentis* A17 were detected after incubation of cells (6×10^6 cells/ μ l) with NHS or hiNHS. After fixation, deposition of C3 (**A**) and the MAC (**B**) were visualized with a polyclonal anti-C3 Ab (1:1000) and a monoclonal anti-C5b-9 Ab (1: 50). Spirochetal DNA (blue) was stained with DAPI. All scale bars are equal to 10 μ m. Spirochetes were observed at a magnification of 1,000 and the data were recorded with an Axio Imager M2 fluorescence microscope (Zeiss) equipped with a Spot RT3 camera (Visitron Systems).

3. Are there any amino acid changes between Chi proteins in each of the Br strains that may also provide information as to the functional significance of these proteins?

We thank this reviewer for bringing this interesting point to our attention. As suggested, we performed additional sequence comparisons using Clustal Omega with available sequences of diverse *B. recurrentis* strains. These analyses clearly reveal that no intra-specific differences exist between the five Chi proteins from eight isolates including strains A1, A11, A17, PAbJ, PAbN, PBek, MAC, and UFA (see below). All homologs possess **100 % sequence identity** with their respective counterpart and, thus confirm the strong conservation of this protein cluster on lp169 over more than 20 years as the first *B. recurrentis* strains (A1, A11, and A17) were isolated in 1994 while the most recently isolated strains (PAbJ, PAbN, PBek, MAC, and UFA) were collected in 2015 in Germany from refugees developed louse-borne relapsing fever. Moreover, the strong sequence conservation of these genes more likely suggest, that each Chi protein display the same anti-complement function in these *B. recurrentis* strains.

Sequence comparison of Chi homologs of *Borrelia recurrentis* strains

Sequence alignments of Chi homologs were conducted by employing Clustal Omega (Madeira et al., Nucleic Acid Res. 2024, 52:W521-W525, <https://doi.org/10.1093/nar/gkae241>) with default setting for proteins and order set to 'input'. To generate the alignment, available sequences from *B. recurrentis* strains A1, A11, A17, PAbJ, PAbN, PBek, MAC, and UFA were used. Differences for strain A11, which can be observed in the 5' region of the homologues ChiD and ChiE, are a result of mutations from the start codon to alternative start codons and might therefore be an artifact of automatic annotation.

The significance of the lack of proteolytic activity of Chi proteins to degrade complement C3 and C3b has not been discussed or brought into relevance in the context of effects of Chi proteins in providing resistance to complement.

We thank the reviewer for pointing this out. Due to word limitation, we were unable to discuss all points in depth. In the revised version we included the following sentences for clarification as suggested:

Line 411-415:“ Not only inactivation of C3b in the fluid phase by Chi-bound plasmin would reduce the formation of C5 convertases but also decreases covalently bound C3b molecules at the borrelial surface. Both scenarios are beneficial for *Borrelia* to combat complement-mediated killing during opsonisation in addition to their properties to directly affect complement activation. “

4. The physiological relevance of structural differences of Chi proteins compared to other known spirochetal proteins that confer pathogen survival has not been discussed to leverage the major findings of this study. Does ROS affect complement sensitivity of Br due to redox sensitive changes in ChiC and ChiE proteins?.

We thank the reviewer for this valuable suggestion. The possible connection between oxidative stress and complement sensitivity is indeed an interesting aspect that we will further address in our ongoing studies. ChiC and ChiE contain conserved cysteine residues that form a reversible disulfide bond, indicating potential redox-sensitive regulation. Although ROS-dependent effects were not investigated in this work, such mechanisms might influence protein conformation or ligand interaction during infection. Reactive oxygen species such as hydrogen peroxide and hypochlorous acid are produced by neutrophils and macrophages and are known to oxidize bacterial proteins, thereby affecting their biological function. It is tempting to speculate whether ChiE provide some level of protection for *B. recurrentis*. We have revised the discussion accordingly and now include the following statement:

Line 382-387:“ ChiC and ChiE contain conserved cysteines that form a reversible disulfide bond, suggesting redox-sensitive regulation of ligand interaction, as confirmed by structural data and the Ellman assay. During infection, *B. recurrentis* may encounters reactive oxygen species such as hydrogen peroxide and hypochlorous acid, which could alter the redox state of these cysteines and thereby modulate protein conformation potentially contributing to protection of the spirochetes under oxidative stress conditions in the human host.”

5. What are the unique structural features of ChiC and ChiE that are different from those spirochetal proteins that interact with complement and how do these proteins alter interaction of Br with the host (other than complement and plasminogen binding). These aspects of the study can be expanded in the discussion part of the manuscript to focus the structural differences which is the mainstay of this study.

We thank the reviewer for bringing this point to our attention. ChiC and ChiE display distinctive structural features compared with other spirochaetal complement-interacting proteins. Specifically, they contain a well-defined, surface-exposed S-domain and a hydrophobic pocket harbouring redox-active cysteines—features absent in already known complement-inhibiting proteins of Lyme (CspA, CspZ, BGA66, BGA71, BBK32) or relapsing fever borreliae CihC, FbpA, FbpB, FbpC, HcpA , FhbA, BhCRASP-1). These unique elements distinguish Chi proteins from other Pfam54_60 members and may contribute to additional host interactions beyond complement and plasminogen binding. We, thus expand the discussion accordingly:

(Line 371-381)“Although Chi proteins adopt a CspA-like fold, they differ structurally from other Pfam54_60 members. As shown by the structural comparisons (**Fig. 4 and S7**), Pfam54_60 proteins fall into two groups differing in the presence and organization of the S-domain. In contrast to these, ChiC and ChiE possess a well-defined S-domain and a hydrophobic pocket containing redox-active cysteines, features absent in other spirochetal complement-interacting proteins such as CspA or BBK32. The ChiB structure also reveals a bound phospholipid within this conserved pocket. In all Chi structures, the pocket is lined with positively charged residues that may interact with negatively charged phosphate groups, suggesting a common binding site for phospholipid-like molecules. Its absence in other Pfam54_60 members, likely due to a shift in helix α 7, reflects structural divergence and potentially distinct functional roles (**Fig. 4 and 7B**). These structural differences may influence how *B. recurrentis* interacts with host factors beyond complement and plasminogen binding.”

Reviewer #2 (Remarks to the Author):

This manuscript describes the functions and structures of five proteins from *Borrelia recurrentis*, designated ChiA to ChiE. *Borrelia recurrentis* is the causative agent of louse-borne relapsing fever, a poverty-associated infectious disease with high mortality. The topic of this work is important due to its relevance to human health. The proteins ChiA to ChiE have not been previously characterized, and the authors propose that these proteins contribute to the pathogen's virulence by functioning as immune evasion factors.

The manuscript addresses significant scientific questions and presents novel, original and numerous results.

However, the overall organization of the manuscript is somewhat disordered. The authors seem to assume that all readers are advanced experts in spirochete biology. While this assumption could be appropriate for a specialized journal, it can hinder comprehension for a broader scientific audience. For instance, the Introduction lacks a few essential sentences on spirochetal lipoproteins, which are central to the manuscript's focus.

We thank the reviewer for raising this point. As suggested, we expand the introduction (see below) by adding important information regarding spirochetal lipoproteins, the complement system as well as a short description of the contributing factors for immune evasion of LD and RF borreliae.

Regarding the description of lipoproteins, the following sentences have been incorporated:

Line 78-84: "In contrast to lipopolysaccharides of Gram-negative bacteria, lipoproteins form a peculiar feature of LD and RF *borreliae* and often serve as serious virulence factors contributing to transmission, adhesion, immune evasion, and persistence²³. These lipoproteins are characterized by their structural and functional domains consisting of an intrinsically disordered N-terminus ("thether") that harbors the signal peptide, a so-called "lipobox", a conserved cysteine residue for diacylation and a sorting signal. The highly flexible tether links the N-terminal lipid anchor to the rest of the protein that execute the specific functional fold."

Also, **the positive and negative control proteins/compounds** used in all assays should be clearly characterized.

We thank the reviewer(s) for raising this particular point. All positive and negative controls have now been clearly described. We also incorporated a detailed description of each protein used either negative or positive control into the Material and Methods section as follows:

Line 657-667: "Concerning the controls utilized, previously characterized His-tagged proteins originated from LD or RF *borreliae* as well as *Acinetobacter baumannii* were produced and purified using the same protocol as described above. These control proteins were selected according to their capability to inhibit the respective complement pathway as follows: BGA66 from *B. bavariensis*¹³ (AP inactivation), CihC from *B. recurrentis*¹⁸ and the C-terminal fragment of BBK32 (BBK32₂₀₅) from *B. burgdorferi*¹⁸ (CP inactivation), CipA from *Acinetobacter baumannii*⁷⁷ (LP inactivation), and CspA from *B. burgdorferi*¹¹ and CihC from *B. recurrentis*¹⁸ (TP inactivation). As negative controls, BtcA from *B. turicatae*³⁸ (AP inactivation), BDU1066 from *B. recurrentis* (CP inactivation), and Vsp1 from *B. miyamotoi*³⁷ (LP inactivation), and HcpA from *B. recurrentis*¹⁵ (TP inactivation) were chosen. In addition, vitronectin (Vn) was included in the cell-based hemolytic assay as a natural inhibitor of the TP as it binds to the preassembled Cb5-7, C5b-8, and C5b-9 complexes and thereby prevent MAC formation⁷⁸."

The Results section is also disorganized, making the paper difficult to follow. Reordering the

subsections—placing structural findings before those on function—could enhance clarity. A similar lack of organization is present in the Materials and Methods section.

We thank the reviewer for this helpful suggestion and agree with the comment. We have therefore reorganized both the Results and the Materials and Methods sections to improve readability. In particular, the structural results are now presented at the beginning of the Results section, following the short description of the identification of the unique gene cluster. All corresponding changes are indicated in the revised version with track changes.

In this study, the ChiA to ChiE proteins were functionally characterized using various in vitro assays. The structural analyses combined crystallographic methods with in silico modeling. Both approaches yield interesting, novel, and potentially valuable results. However, the quality of some data and interpretations is not entirely convincing.

Detailed comments:

The introduction is scarce and does not cover the content of the publication. The authors supplement the information and introduce new names, phenomena and processes in the results or discussion sections e.g.:

Line 92, 93 Structurally diverse proteins of Lyme disease (LD) and relapsing fever (RF) borreliae contribute to the inhibition of the terminal pathway (TP), this information should be expanded and moved to the introduction, together with the information about CspA (line 96)

We thank the reviewer for this remark, we have expanded this part of the introduction as follows:

Line 63-77: “Structurally diverse surface-exposed lipoproteins of Lyme disease (LD) and relapsing fever (RF) *borreliae* contribute to complement inhibition¹¹⁻¹³. Regarding LD spirochetes, the FH and FHL-1 binding proteins CspA and CspZ have been identified as the key complement inhibitors of the AP whereas CspA also block TP activation by interaction with C7, C8, C9, and the MAC, respectively^{11,14}. Factors involved in complement evasion of RF spirochetes comprises CihC, the FH-binding proteins HcpA, BhCRASP-1 or FhbA, the fibronectin-binding proteins FbpA, FbpB, and FbpC as well as CbiA⁸. CihC and HcpA, both of which exhibit anti-complement properties and potentially contributing to the pathogenesis of *B. recurrentis* protect spirochetes from complement-mediated killing by binding of C1-Inh, C4BP (via CihC), and FH (via HcpA) to terminate CP and AP activation^{15,16}. Recently, a novel mode of CP inactivation targeting the formation of the initial C1 complex has been described for diverse RF borreliae and involves the interaction of C1r with CihC, FbpA, FbpB, and FbpC, respectively¹⁷⁻¹⁹. Thus, recruitment of diverse host regulators represents an ingenious immune evasion strategy of this particular pathogen. Moreover, it has been shown that recruitment of plasminogen and activation to plasmin by urokinase-type activator (uPA) enhances brain and heart invasion of LD and RF borreliae in the murine host²⁰⁻²².”

Complement is not sufficiently described as well, as there are binding studies on FB, FH, FI and C3b (line 107) etc., which are not mentioned in the introduction. It would be also advisable to have a comprehensive graphical diagram illustrating e.g. the complement system in the introduction.

As suggest, we now included additional information on complement into the introduction and added a separate figure to the supplements (indicated as **Fig. S1** in the revised version) to increase the readability and clarity of the manuscript. This figure illustrates activation and regulation of complement in a more comprehensive manner. Due to limited word counts, therefore we must reduce this part to the necessary information but include a detail description on complement in the

legend to supplementary figure S1 to allow unfamiliar readers to easier understand the impact of the borrelial proteins investigated herein on complement. This part was now modified as follows:

Line 51-62: “Complement represents a potent barrier against invasion of pathogenic microorganisms⁹. This system is activated via three pathways: classical (CP), lectin (LP), and alternative (AP) (**Fig. S1**)¹⁰. After initiation of the CP by C1q binding to immunoglobulins or surface structures, by carbohydrate recognition (LP) or by spontaneous C3 activation (AP), the C3 convertases C4b2b (CP/LP) and C3bBb (AP) are formed. Subsequent cleavage of C3 into C3b and C3a boost opsonisation of microbes and formation of the C5 convertases C4b2b3b (CP/LP) and C3bBb3b (AP). Upon cleavage of C5, C5b bound to the microbial surface and initiates activation of the terminal pathway (TP) by recruiting C6–C9, forming the pore-forming membrane attack C5b-9 complex or MAC leading to bacterial lysis¹⁰. To prevent excessive activation, complement is strongly controlled by C1-INH (CP), FH and FHL-1 (AP), and C4BP (CP and LP), vitronectin (Vn) clusterin, and FHR-1 (TP) (**Fig. S1**). FH, FHL-1, and C4BP, respectively, act as cofactors for factor I, which inactivates C3b and C4b, thereby limiting C3 convertase formation. Assembly of the MAC is terminate by vitronectin, clusterin, and FHR-1, respectively.”

Vsp1 (line121) , uPA (line 125) are not introduced before

To introduce Vsp1 and uPA we added the following sentences:

Line 75-77: “Moreover, it has been shown that recruitment of plasminogen and activation to plasmin by urokinase-type activator (uPA) enhances brain and heart invasion of LD and RF borreliae in the murine host²⁰⁻²².”

Line 234-236: “As depicted in **Fig. 7A**, binding of plasminogen to all Chi proteins as well as HcpA of *B. recurrentis*¹⁵ could be demonstrated but not to Vsp1 of *B. miyamotoi*, previously shown to lack plasminogen binding.”

Types of CP inhibitory mode of complement interacting proteins, opsonization from the discussion should be also be in the introduction part.

As suggested, the following sentence has been added to the introduction:

Line 71-73: “Recently, a novel mode of CP inactivation targeting the formation of the initial C1 complex has been described for diverse RF borreliae and involves the interaction of C1r with CihC, FbpA, FbpB, and FbpC, respectively¹⁷⁻¹⁹.”

Line 73. Identification of a unique gene cluster on lp190 of *Borrelia recurrentis*

This section, along with the associated figures and tables, should include a comparison of sequence similarity/identity between the corresponding proteins in *B. recurrentis* and *B. duttonii*. Some Chi protein sequences for example from *B. duttonii* GenBank entry (CP000979) share over 99% identity with presented *Borrelia recurrentis* ChiA to ChiE. This supports that both *Borrelia* species utilize similar immune evasion mechanisms.

We thank this reviewer for this valuable suggestion and draw our attention to this important point. As already mention by this reviewer, the corresponding Chi orthologs in *B. duttonii* (BDU_1021 to BDU_1015) exhibit high sequence identities and similarities (93.6 to 99.6). These additional sequence analyses have now been incorporated in the revised version of the manuscript as a separate supplementary figure (**Fig. S3**). We also refer to this figure in the result and discussion as follows:

Line 105-106: “Sequence analyses revealed high sequence identities/similarities between Chi proteins and their corresponding orthologs from *B. duttonii* Ly (93.6 to 99.6 %) (**Fig. S3**).”

and

Line 334-335: “Furthermore, the high sequence identity of the Chi corresponding proteins found in *B. duttonii* (Fig. S3) imply a similar immune evasion mechanism utilized by this particular RF species.”

Line 84: Assessment of the inhibitory capacity of Chi proteins on complement activation and MAC assembly. Brief explanation of the assay's principle and mechanism would greatly aid reader understanding.

To clarify this point we slightly modify this part of the manuscript and incorporated the following sentences:

Line 187-191: “To assess the complement inhibitory capacity of the Chi proteins, an ELISA-based approach was conducted. Initially, microtiter plates were immobilized with specific compounds allowing a targeted activation of the respective pathway. After application of the reaction mixtures consisting of NHS pre-incubated with the analysed protein, formation of the MAC was detected by a neoepitope-specific C5b-9 antibody. Complement inactivation was indicated by low absorbance values.”

Please note that a detailed description of this assay was also presented in the Materials & Methods section.

Line 634-647: “A modified ELISA-based approach (WiELISA) was applied to assess the inhibitory capacity of bacterial proteins on the alternative (AP), classical (CP), Lectin pathway (LP) as described previously⁷⁶. Nunc MaxiSorp 96-well microtiter plates were coated with either LPS (10 µg/ml) (Hycult Biotech, Beutelsbach, Germany) for the AP, human IgM (3 µg/ml) (Merck, Darmstadt, Germany) for the CP, or mannan (100 µg/ml) (Merck, Darmstadt, Germany) for the LP at 4 °C overnight. Following three wash steps with TBS containing 0.5 % (v/v) Tween20 (TBS-T), the wells were blocked with PBS-T containing 1 % BSA for 1 h at RT. NHS (15 % for the AP, 1 % for the CP, and 2 % for the LP) was then pre-incubated with a final concentration of 4 µM (initial analyses) or increasing concentrations (0.5, 1, and 4 µM) (dose dependence analyses) of purified His₆-tagged proteins for 15 min at RT before being added to the wells to initiate complement activation of the respective pathway. After washing with TBS-T, a neoepitope-specific, monoclonal anti-C5b-9 antibody (1:500) was added to detect formation of the MAC as the final activation step of the cascade. Following incubation for 1 h at RT, wells were washed thoroughly with TBS-T and incubated with HRP-conjugated anti-mouse immunoglobulins (1:1,000) at RT for 1 h. All reactions were developed applying tetramethylbenzidine as substrate.”

Line 88, Figure 1 and corresponding Materials and Methods section (line 526). The amount of ChiA to ChiE used in the assays is not clearly stated. Please clarify whether the reported concentration refers to the final concentration in the assay or to a stock concentration that was subsequently mixed with NHS. Please provide a clearer and more detailed description of the negative and positive controls. Figure1 – Figure 1b includes the data from Figure 1a. One of Figure 1 panels could be placed in the Supplementary data.

The reported concentrations in the manuscript (main text and Materials and Methods) refers to the final concentrations of the proteins employed in the respective assay. To clarify that final concentrations have been used, we slightly modified this part as follows:

Line 193-196: “The strongest inhibitory effect on the AP was observed for ChiB, ChiD, and ChiE at a final concentration of 1 and 2 μM , respectively. Likewise, ChiA and ChiC also inhibited the AP but only at the highest concentration (4 μM) employed (**Fig. 5D**). ChiE showed a dose-dependent inhibition of the CP and LP at 2 μM and 4 μM , respectively (**Fig. 5E and F**).”

and

Line 640-643: “NHS (15 % for the AP, 1 % for the CP, and 2 % for the LP) was then pre-incubated with a final concentration of 4 μM (initial analyses) or increasing concentrations (0.5, 1, and 4 μM) (dose dependence analyses) of purified His₆-tagged proteins for 15 min at RT before being added to the wells to initiate complement activation of the respective pathway.”

Line 92. Inhibition of terminal pathway. Brief explanation of the assay's principle and mechanism would greatly aid reader understanding.

To make this part more reader-friendly and understandable, we added the following sentence:

Line 197-200: “After preparation of C5b-6 sensitized sheep erythrocytes, proteins pretreated with purified C7, C8, and C9 were added. Lysis of erythrocytes was indicated by the release of hemoglobin due to the insertion of the formed MAC. “

Figure 2, Results and corresponding Materials and Methods section (line 538). Please explain mentioning vitronectin.

As suggested, vitronectin was shortly mentioned in the introduction (**Line 58-62:** “To prevent excessive activation, complement is strongly controlled by C1-INH (CP), FH and FHL-1 (AP), and C4BP (CP and LP), vitronectin (Vn) clusterin, and FHR-1 (TP) (**Fig. S1**). FH, FHL-1, and C4BP, respectively, act as cofactors for factor I, which inactivates C3b and C4b, thereby limiting C3 convertase formation. Assembly of the MAC is terminate by vitronectin, clusterin, and FHR-1, respectively.” and the figure legend of supplementary figure 1. In the results (**line 202**), we added a short note that vitronectin serve as natural inhibitor of the MAC.

The amount of ChiA to ChiE used in the assays is not clearly stated. Please clarify whether the reported concentration refers to the final concentration in the assay or to a stock concentration that was subsequently mixed with complement compounds. Please clarify it.

We now incorporated the final concentration used as suggested by the reviewer(s) to clarify this point as follows:

Line 204-207: “Investigating ChiC, ChiD, and ChiE, all proteins strongly impaired C9 polymerisation in a dose-dependent fashion of up to a final concentration of 0.2 μM , similar to CspA. In contrast, ChiA did not affect C9 polymerisation (**Fig. 6B**), while ChiB displayed an inhibition only at a final concentration of 7.5 μM (**Fig. 6C**).”

and in the Materials and Methods section as well:

Line 670-673: “To assess the inhibitory capacity of the purified borrelial proteins on C9 polymerisation, increasing concentrations (final concentrations 0.005 $\mu\text{g}/\mu\text{l}$ to 0.22 $\mu\text{g}/\mu\text{l}$ or 0.2 μM to 9 μM) of the Chi proteins of *B. recurrentis*, CspA of *B. burgdorferi* (positive control), and BSA (negative control) was incubated with C9 (0.06 $\mu\text{g}/\mu\text{l}$ or 0.9 μM) as previously described^{11,13}.”

Please provide a clearer and more detailed description of the negative and positive controls.

Done (please see our comment above)

Line 201-204: "The strongest inhibition among all proteins and similar to vitronectin, a natural inhibitor of the MAC, was observed for CihC from *B. recurrentis* while CspA from *B. burgdorferi*, a well-characterized inhibitory protein of the TP¹¹, displayed a weaker inhibitory capacity."

Figure 2 B-H – please use the same format for protein concentration. It is difficult to determine how 0.5-10ug of tested proteins relates to the molar concentrations used in Figure 2A. Please use consistent concentration format throughout the manuscript.

We completely agree with the reviewer(s) to use consistent concentrations for both assays to make the data more comparable and transparent. As suggested, consistent concentrations for the C9 polymerisation assay has been presented in the text and in **Figure 6** as well as **Fig. S12**.

Line 106. "The very low absorbance values suggested that none of the Chi homologs interacted with FB..." and Figure S2. Authors inconsistently interpret the values of absorbance in the context of binding. In Figure S2A, absorbance value for ChiA is ~0.3 and is interpreted as fine binding, while similar result in Figure S2B for a few Chi proteins means no binding, and in Figure S2G means strong binding (BSA levels are comparable at these panels, A490 is ~0.1-0.2). This interpretation decreases the reliability of the presented results. Description of control proteins is missing.

We thank the reviewer(s) for pointing out the inconsistently interpretation of the ELISA data obtained with the different binding analyses, in particular the data presented in figure S2A, S2B, and S2G (new numbering: **S8A, S8B, and S8G**). As mentioned in the text and the legend to figure S2 (new **figure S8**), BSA was always used as a negative control for calculating statistical significance for interaction of the five Chi proteins with the respective complement component. However, the absorbance values obtained with BSA differed from ≤ 0.1 to ~ 0.5 depending on the anti-complement antibody used (monoclonal versus polyclonal), thus making the interpretation of the data collected with C4, C4b, and C5 somewhat difficult. To overcome this constrains, we included previously well-characterized control proteins, e.g. CbiA from the relapsing fever spirochetes *B. miyamotoi* (C3, C5 and FH binding), the C-terminal fragment of BBK32 from LD spirochete *B. burgdorferi* (C1r binding) or CipA from *Acinetobacter baumannii* (FI binding). At the time when these assays were conducted, no control protein of bacterial origin was available that interact with FB, C1q, C4, and C4b, respectively. However, the high absorbance values obtained from the binding analyses of ChiE with C1q indicate that both molecules appears to interact. In contrast, the low absorbance values achieved from the binding analyses of ChiE with C4 and C4b are more difficult to interpret despite the statistical significance calculated.

To address this point, we tone down our interpretation in this part of the manuscript as follows:

Line 211-217: "To elucidate the molecular mechanism(s) of complement inhibition, interaction of Chi proteins with selected complement components of the AP (C3b, FB) and the CP (C1q, C1r, C4, and C4b) as well as C5, FH, and FI was investigated. As demonstrated in **Fig. S8A**, all Chi proteins bound C3b but lower absorbance values were obtained employing ChiA. When binding to FB was assayed, all Chi homologs exhibited low absorbance values (~ 0.25) as well, even if statistically significant for ChiB and

ChiE (**Fig. S8B**). In comparison to the C5-binding CbiA protein from *B. miyamotoi*¹² used as a positive control, only ChiB and ChiE appears to bind C5 to some extent.”

Line 112 and corresponding S3J-K Figure. It is difficult to determine how 5ng/ul of the protein in the Figure S2 relates to the molar amount of proteins used in S3 Figure.

In general, a standard protocol was used in which the same protein concentration of 5 ng/μl was used for the bait protein (Chi's, and controls) and the respective captured ligand (complement component) as a practical guidance. These data were presented in Figure S2 (new figure S8) as mentioned by the reviewer(s) and served as basis for the dose-dependent binding analyses presented in figure S3 (new **figure S9**). The molar concentration used in figure S2 (new **figure S8**) correspond to 28 nM for C3b, 26 nM for C5, 12 nM for C1q, 24 nM for C4, and 28 nM for C4b, respectively. Regarding Figure S3A-H (new figure **S9A-H**), lower concentrations of C3b and C5 was used to trace the increase of the absorbance values and to end up with similar concentrations applied for C3b and C5 in Figure S2A and C (new Figure **S8A**).

Figure S3F-H: ChiD binding to C5 was considered insignificant in the previous Figure S2, but in Figure S3 is significant and has the same level as ChiE and ChiB. Lack of consistent results.

We thank and agree with the reviewers' assessment and to point to the inconsistent results of these assays. Although the conditions have not been changed when these assays were performed in terms of the proteins, primary and secondary antibodies, buffers etc., consumables (microtiter plates, tips) used, we, however, observed lower absorbance values for BSA in the dose-dependent binding assays which were then considered significant but were insignificant in the standard ELISA (ChiD and ChiE). Honestly, we have no explanation for these observation yet, even if all these experiments were repeated.

Why there is increasing signal for controls with BSA in Figure S3J-K?

For consistency, we decided to present our data using BSA as a control in each assay (please see new figure S9J-K), although we are aware that BSA could bind to some extent C4 and C4b indicating that BSA is not an inert protein as always thought, and, in principle cannot be used for all applications. To make our study as transparent as possible we included these data as well. Overall, the increasing signals could easily be explained by the interaction of BSA with C4 as well as with C4b.

Line 115. According to Figure S3, the saturation is not reached at most cases except for S3A,B,C,D. Line 115:..”binding is evident C1q, C4, C4b and C5....” According to the Figure S3J-K it is not evident, rather it is unspecific binding due to the high concentrations of proteins applied. Why the concentrations up to 200nM were used only in Figure S3J-K but not in the other assays from FigureS3? What would be the results then?

By performing these assays, we sought to figure out whether a plateau was achieved when a 10 fold higher concentrations (up to 200 nM) was only applied for the binding of C4/C4b to ChiE, respectively. In consideration of these data, it is not entirely clarified whether C4 and C4b temporarily interact with ChiE under the conditions employed (see new **figure S9**).

Regarding binding of C1q to ChiE, we increased the concentration of C1q of up to 200 nM, however, no saturation could be achieved even at the highest concentration applied. Application of concentrations up to 25 nM, an increase in unspecific binding of BSA could be detected as already shown for the binding of C4 and C4b to ChiE (Fig. S9J and K). The data with C1q have now been

incorporated in **Figure S9** for transparency. The respective part was slightly modified the respective sentence as follows:

Line 224-226: “A concentration-dependent binding to ChiE could be observed for C1q, C4, C4b, and C5, however, a saturation could not be achieved even at the highest concentration applied (**Fig. S9F to K**).”

Line 117. Interaction of Chi proteins with plasminogen. Please provide a more detailed description of the method used for KD calculation in the Materials and Methods section. It is only briefly mentioned in the Figure 3 legend.

As suggested by the reviewer(s) we briefly explain which method/tool was used to calculate the dissociation constant as follows:

Line 690- 692: “To calculate the dissociation constant for the binding of plasminogen to Chi-proteins, the nonlinear regression model (four parameters dose-response curve) with a variable slope (Hill slope) was selected in GraphPad Prism 10.2.2.”

Please note that we have also corrected the figure legend accordingly:

Line 1158-1159: “Binding curve and dissociation constant were calculated via nonlinear regression, four-parameter model.”

Line 125. Conversion of plasminogen to the active serine protease plasmin and Figure S4. Are the values of A405 averaged if the experiments shown in the Figure S4 were performed in duplicate?

The plasminogen activation assays (now presented in **Figure S11**) were performed in triplicate at three different days.

Authors mention that “At least three independent experiments were conducted, each in triplicate”. What are the results?

We apologize for the confusion this may cause. These experiments were conducted in triplicate on different days to obtain three independent data sets where each reaction mixture was performed in triplicate. To create the graphics the means without the standard deviations are presented. The raw data of these assays are now incorporated in the Data Source File.

The proteolytic activity upon binding to Chi proteins could be described as, for example % of protease activity of positive controls after 24h, in a supplementary table.

As suggested by the reviewer(s), we calculate the % of protease activity in relation to plasminogen as control (100 % activity) for each protein analysed after 24 h of incubation (see table below). As expected, the highest protease activity was obtained with BBA70 used as positive control (59.2 %) while Vsp1 with 11.9 % showed the lowest percentage.

Protein	Percentage of protease activity Values expressed in % in relation to plasminogen as control
ChiA	21.4
ChiB	29.5
ChiC	34.5
ChiD	26.3
ChiE	39.5

HcpA	53.4
BBA70	59.2
Vsp1	11.9
Plasminogen	100.0

We decided to include these data into the main text of the revised manuscript:

Line 253-256: “A strong activation was demonstrated upon activation of plasminogen in the presence of uPA with the strongest activation achieved with BBA70 as positive control (59.2%) following HcpA (53.4 %), ChiE (39.5 %), ChiC (34.5%), ChiB (29.5 %), ChiD (26.4 %), ChiA (21.4 %), in relation to plasminogen (100% activity) while Vsp1 reached 11.9 % as negative control (**Fig. S11**).”

Line 134. Please describe more the role of lysines and tranexamic acid in the assay with plasminogen.

Tranexamic acid is structural similar to lysine, and binds to the same lysine-rich sites on plasminogen that are crucial for its interaction with fibrin and other molecules, and thus, effectively blocks plasminogen activation.

Due to word limitation we slightly modified to respective sentence in the main text as follows:

Line 256-259: “To corroborate the role of lysines in the binding with plasminogen, tranexamic acid, an anti-fibrinolytic lysine analogue was applied. Proteolytic activity of plasminogen was impaired when ChiB, ChiD, and ChiE was assayed indicating that lysines are involved in binding to all five Chi proteins.”

and add additional information in the Materials and Methods section:

Line 685-687: “Further reactions containing 50 mM tranexamic acid, a lysine analogue with high affinity to the lysine binding sites of plasminogen, to assess the role of lysine on binding of plasminogen to Chi proteins.”

Figure S4, Why in the assays with ChiA and ChiC, the proteolytic activity in the presence of TXA is almost as high as in its absence?

Our new generated data revealed that TXA strongly influenced binding of plasminogen to ChiA, ChiB, and ChiD, and to some extent also to ChiC, and ChiE indicating that lysines do play a role in binding of plasminogen (please see **figure S11**).

Why in the assays with ChiA and ChiC, the proteolytic activity in the absence of plasminogen is much higher than corresponding proteolytic activity in the absence of plasminogen in the rest of assays from on Figure?

To clarify this point, we have repeated the plasminogen activation assays to collected new data sets. As shown in **figure S11**, the proteolytic activity in the absence of plasminogen is in all assays in the expected range.

Line 135 and Fig3G – the results from this figure must be commented more comprehensively. Why results for ChiA in Figure 3G show strong plasmin activity as compared to the rest of Chi proteins, while the previous results for ChiA in Figure 3 and Figure S4 indicated something opposite?

Regarding the data collected with ChiA, we observed a lower binding capacity of this protein to plasminogen compared to the other Chi`s proteins (**Figs. 7 and S11**). This means that there is sufficient proteolytic activity present to degrade C3b. Degradation of C3b by activated plasmin

bound to Chi's proteins was measured after 24 h of incubation, thus, cleavage products are generated continuously over time and then detected by Western blot analyses. The activation assays and the Western blot analyses cannot be directly compared.

To clarify this point, we added the following sentence to the results:

Line 261-263: "Although ChiA possesses a lower plasminogen binding capacity (**Fig. 7B**), conversion to plasmin appears to be sufficient to cleave C3b. These findings resemble what has been observed for SbiA of *Staphylococcus aureus*⁴¹. "

Line 141: I do not understand the statement "The lack of alphafold model" as AlphaFold2 can predict structures never previously seen in the PDB, i.e. novel protein folds (Bordin et al., 2023; Barrio-Hernandez et al., 2023; Durairaj et al., 2023). Therefore, it is unclear why authors have not attempted (at least not described) to generate AlphaFold2 predictions from sequence and use this model for molecular replacement in the first place instead of the homology model with such a low identity. The authors should firstly describe the outcome of an attempt of use alphafold predicted model for molecular replacement, before anomalous measurement, autoSol and autobuild stages.

We thank the reviewer for this comment and the opportunity to clarify this point. The ChiB structure was solved in 2018, at a time when AlphaFold2 had not yet been released. The first AlphaFold2 release appeared at the end of 2020, and public access to the prediction server became available only in mid-2021. Because of the low sequence identity to known structures, the homology model used for molecular replacement was not sufficient to solve the structure, so we performed experimental phasing using anomalous measurements, followed by AutoSol and AutoBuild. We fully agree that an AlphaFold2-based approach would have greatly facilitated structure determination, and we would have gladly used this tool had it been available at the time.

Also, in my opinion reviewers should have access to both the MTZ file (containing structure factor amplitudes and phases) and the PDB file (containing the atomic coordinates of the refined model). These files are essential for assessing the quality and validity of the reported structure, especially, when deposit structure release is hold for publication.

In view of the very high proportion of RSRZ outliers in both structures, it is not clear whether this is due to the lack of electron density for the amino acid side chains or to their poor modeling in electron density. This issue is not addressed in the text and also could not be checked by reviewers in the structural data.

We agree with the reviewer that access to both the MTZ files and the PDB files is important for evaluating the quality of the reported structures, especially when the public release is on hold until publication. The ChiA and ChiB structures have been deposited in the Protein Data Bank under accession codes 9FSD and 9FQQ, respectively, including the corresponding MTZ files. The entries are currently on hold due to ongoing competitive work in the *Borrelia* field. We would have no objection to making these files available to the reviewers. However, the validation reports generated by the PDB are generally sufficient to assess the quality of the deposited structures.

In the ChiB structure (1.5 Å), only a few RSRZ outliers (18 residues out of 284) were detected. These are mainly located in surface regions with weak side-chain density, particularly in the N-terminal helix, which is consistent with the higher local B-factors.

The ChiA structure (2.7 Å) crystallized in the monoclinic space group C2 with two monomers in the asymmetric unit connected by a disulfide bond. As ChiA is rapidly oxidized, crystallization was

challenging; well-diffracting crystals were obtained only after seeding and in the absence of reducing agents. Minor flexibility between the monomers likely contributed to the moderate resolution. Space-group determination was initially ambiguous but confirmed unambiguously during final refinement. The data were processed with XDS, and quality was assessed with xtriage (Phenix). The relatively high Wilson B-factor (63.5; Supplementary Table 2) indicates increased overall flexibility, consistent with the higher number of RSRZ outliers (97 residues out of 530). These residues are mainly located in the N-terminal helix, the S-domain, and the connecting loops between helices. Nevertheless, the overall model is well defined, supported by an R_{free} of 30% and its close agreement with the AlphaFold2 prediction.

Line 180. Functional properties and redox regulation of Chi proteins...“All Chi proteins, except ChiC, bind to plasminogen”... In the previous sections the authors showed that ChiC binds plasminogen and assessed $K_d = 179\text{nM}$. This section and related Figures contain mostly speculations, should be shortened, edited and placed in the discussion.

We thank the reviewer for pointing out this discrepancy. We apologize for the incorrectness of our statement which has been removed in the revised version. As suggested, the paragraph has been shortened and partly moved to the Discussion section.

Line 152-171: “The region between $\alpha 7$ and $\alpha 8$ varies considerably among Chi proteins. In ChiA and ChiB, it appears disordered in the crystal structures, indicating high flexibility, and is also predicted with low confidence by AlphaFold2. ChiE possesses a markedly shorter linker (seven residues) compared to ~ 30 residues in the other homologs (**Fig. 2A, 3A; Fig. S4**), forming a distinct surface pocket that provides more open access to the hydrophobic pocket (**Fig. 3B**).

In ChiC and ChiE, four residues — C165, C260, N257, and D261 in ChiC; C156, C227, H224, and D228 in ChiE — form a cysteine–histidine–aspartate constellation reminiscent of a catalytic triad, located within a hydrophobic pocket (**Fig. 3C and D**). The two cysteines (C260/C227) are positioned in close proximity, potentially allowing disulfide bond formation.

To assess redox reactivity, Ellman’s assay was performed on oxidized and reduced Chi proteins (**Table S3**). In ChiA, one reactive thiol was detected, consistent with a surface-exposed cysteine forming a disulfide with a symmetry-related monomer. No free thiols were detected in ChiB, in line with its single cysteine being buried in a hydrophobic pocket. ChiD showed limited thiol reactivity, likely due to partial shielding of its cysteine by K166 and N265. In ChiC, one accessible thiol was observed in the oxidized state, corresponding to the surface-exposed C165, whereas in the reduced form only 1.5 thiols were detected instead of the expected three. This suggests that access to the buried cysteines is restricted by a flexible loop covering the pocket entrance (**Fig. 3A**). In contrast, ChiE showed two reactive thiols in the reduced form and 0.5 in the oxidized form, consistent with reversible redox switching of C156 and C227 within the hydrophobic pocket. Together, these findings indicate that ChiC and ChiE exhibit redox-responsive cysteine pairs that may undergo reversible disulfide formation.”

and 382-387: “ChiC and ChiE contain conserved cysteines that form a reversible disulfide bond, suggesting redox-sensitive regulation of ligand interaction, as confirmed by structural data and the Ellman assay. During infection, *B. recurrentis* may encounter reactive oxygen species such as hydrogen peroxide and hypochlorous acid, which could alter the redox state of these cysteines and thereby modulate protein conformation, potentially contributing to protection of the spirochetes under oxidative stress conditions in the human host.”

Line 210 and Figure S8. $\sim 9\text{kDa}$ product of digestion is out of range of presented SDS-PAGE picture.

We thank the reviewer(s) to draw our attention to this point. Now, we have included a revised figure showing the silver stain of both polyacrylamide gels in total. As clearly demonstrated in this figure, no 9 kDa C3a fragment could be detected.

Figure S8. ChiA lane contains double band for ChiA. Could this influence the results in the other presented experiments like, for example, problems with assessment of Kd for plasminogen binding?

We did not observed a double band for ChiA after purification using Ni-NTA affinity chromatography. The figure below shows a silver-stained gel of the purified proteins (500 ng protein was subjected to SDS-PAGE) used.

The double band in supplementary figure most likely occurred after the 2h incubation period at 37 °C before subjecting the reaction mixtures to SDS-PAGE. For all functional and binding assays as well as for crystallography, ChiA was either immobilized at 4 °C or analyzed at RT. Thus, it could be expected that ChiA was stable in all other assays performed.

Line 215. Structural comparison with surface proteins of LD borreliae. This section and related Figures contain mostly speculations, should be shortened, edited and placed in the discussion.

We respectfully disagree with the reviewer's assessment that this section is mainly speculative. The structural comparison presented here is based on quantitative DALI and Chimera MatchMaker analyses, which provide objective measures of structural similarity (Z-scores, RMSD values, aligned residues). These results are therefore data-driven rather than speculative. However, we agree that the interpretation of structural groupings may fit better within the Discussion section. We have accordingly shortened and edited this part and moved the interpretative statements to the Discussion, while retaining the essential results of the DALI-based comparison in the Results section.

Line 371-381: "Although Chi proteins adopt a CspA-like fold, they differ structurally from other Pfam54_60 members. As shown by the structural comparisons (**Fig. 4, S7**), Pfam54_60 proteins fall into two groups differing in the presence and organization of the S-domain. In contrast to these, ChiC and ChiE possess a well-defined S-domain and a hydrophobic pocket containing redox-active cysteines, features absent in other spirochetal complement-interacting proteins such as CspA or BBK32. The ChiB structure also reveals a bound phospholipid within this conserved pocket. In all Chi structures, the pocket is lined with positively charged residues that may interact with negatively charged phosphate groups, suggesting a common binding site for phospholipid-like molecules. Its absence in other Pfam54_60 members, likely due to a shift in helix α 7, reflects structural divergence

and potentially distinct functional roles (Fig. 1, 4). These structural differences may influence how *B. recurrentis* interacts with host factors beyond complement and plasminogen binding.”

Line 238. Selected residues potentially involved in the host-protein interaction – it is not specified on what basis these residues were selected

Figure 1 illustrates all mutated residues, highlighted in teal, and the C-terminal deletion variants that could not be produced in *E. coli*, shown in brown shades. The N-terminal helix and the distinct S-domain protrude from the compact core of the protein and are not present in other PFam54_60 members. These regions were therefore selected to test their contribution to complement interaction. In addition to testing their role in complement interaction, the C-terminal deletion variants were also designed to assess a possible role of the C-terminal helix in dimerization, as suggested by structural modeling and comparison with CspA, which binds factor H as a homodimer. Furthermore, we introduced single amino acid substitutions in residues lining the hydrophobic pocket (Y177A, R226E, and W255A), which are unique to Chi proteins. Tyrosine 177 and tryptophan 255 define the ends of the pocket, while arginine 226 interacts with the phosphate group of a bound lipid molecule. All single-point variants could be expressed and purified, whereas none of the C-terminal deletion constructs yielded soluble protein, suggesting that the C-terminal region is essential for structural stability. As no significant effects on complement inhibition, C3b/C5 binding, or C9 polymerization were observed (Fig. S12E–G), we kept this section concise in the manuscript. To clarify this point, we have revised this chapter in the results section.

Line 265-277: “To identify complement-interacting regions, we generated ChiB variants carrying targeted deletions or single amino acid substitutions (Fig. 1A, S12A). Residues were selected based on Chi-specific structural features, including the N-terminal helix, the distinct S-domain, residues forming the hydrophobic pocket (Y177, R226, W255), and the C-terminal helix possibly involved in dimerization. Larger deletions removed the N-terminal helix A (aa 1–58), the β -hairpin (aa 97–107), and the C-terminal helix E (Δ 248–284, Δ 273–284, Δ 277–284) to assess their contribution to ligand binding or dimerization. Despite extensive expression trials using different vectors, host strains, and conditions, none of the C-terminal deletion variants could be obtained in soluble form. None of the tested variants including deletion of the protruding N-terminus, the β -hairpin, or the point substitutions showed a measurable effect on complement inhibition (Fig. S12A–D). Only the Y177A substitution slightly reduced inhibitory activity, and no differences in C3b/C5 binding or C9 polymerization were observed (Fig. S12E–G). These findings indicate that several regions, or residues at the C-terminus, may jointly contribute to complement interaction.

Lines 250 to 273. Section: Chi homologs protect spirochetes from complement-mediated killing and confer serum resistance. This section presents really important findings on the biological relevance of the ChiA–E proteins. However, the results are described too casually and in a disorganized manner. The reader should be clearly informed about which *Borrelia* strain was used in each specific experiment.

As recommend by the reviewer(s), we added more information in the main text to specify the proteins and *Borrelia* strains used for these assays. To be honest, we do not find that this section is disorganized as we described step by step the findings with the purified proteins moving to the gene expression analyses to end up with the data collected with the gain-of-function strains which is, as mentioned by the reviewer(s) the most important findings. In sum, we strongly believe that these

flow of the text, help the readers to easily understand the relevance of these assays. This section was modified as follows:

Line 279-305: “To assess the immunomodulatory role of Chi homologs in protecting serum-sensitive *B. garinii* cells from complement-mediated killing, spirochetes were treated with 30 % NHS pre-incubated with 10 μ M purified Chi proteins (**Fig. 8**). By counting viable cells after 4 h incubation, all Chi homologs, except ChiA, conferred protection of susceptible spirochetes to NHS. Likewise, the motility and viability of spirochetes was not affected upon incubation with heat-inactivated NHS or BGA66 from *B. bavariensis* PBi used as a control as a known inhibitor of the CP, AP, and MAC¹³. Under the same conditions, approximately 80 % of the serum-sensitive cells were killed in the presence of NHS or NHS pre-incubated with 10 μ M BSA or Tris/HCl (buffer control) (**Fig. 8A**). These findings indicate that exogenous Chi proteins protect serum-sensitive *B. garinii* cells from human complement. To further elucidate the role of Chi proteins for facilitating serum resistance, serum-sensitive *B. garinii* G1 was used to generate a number of gain-of-function strains that ectopically produce individual Chi proteins as well as CihC or HcpA. To confirm expression of the *chi* homologous genes in *B. garinii* G1, RT-PCR was conducted. Initial expression analyses revealed that all borrelial genes analyzed were expressed in the surrogate strain but not in *B. garinii* G1 carrying the basic pKFSS1 shuttle vector (**Fig. S13**). Moreover, expression of all *chi* genes, as well as *cihC*, *hcpA*, and *flaB* could be demonstrated indicating that the entire *chi* gene cluster was expressed *in vitro* (**Fig. S14A**). Western blot analyses also confirm synthesis of CihC, HcpA, and ChiB in the WT strain *B. recurrentis* PAbJ cultivated in BSK medium supplemented with human or rabbit serum (**Fig. S14B**). Having demonstrated expression of *chi* genes, we assessed serum survival of gain-of-function *B. garinii* G1 strains by incubating spirochetes in 30 % NHS for 4 h (**Fig. 8B**). Significant levels of serum resistance comparable to HcpA-producing spirochetes could be observed for the ChiB-, ChiC-, and ChiD-producing spirochetes, respectively. In contrast, ChiA- and ChiE-positive spirochetes were considered serum susceptible as most of the cells were killed during the incubation period. CihC-producing spirochetes appeared to be less protected compared to spirochetes producing HcpA. As expected, no impact on motility and viability could be observed after heat-inactivated NHS (hiNHS) exposure. These findings revealed that ectopically-produced ChiB, ChiC, and ChiD facilitate resistance of serum-sensitive spirochete to human complement.”

Line 260. RNA expression not always means that the encoded protein is produced. Please be more careful when drawing conclusions and describe your results more precisely.

We completely agree with the reviewer(s) assessment that RNA expression does not mean that the protein of interest is also produced. To avoid any misunderstanding, we always used a clear terminology and described the expression of the respective genes in the gain-of-function strains as well as in the WT *B. recurrentis* strain under *in vitro* conditions but not the production of these Chi homologs.

Line 264. Please comment on the fact that the bands for the same proteins, CihC and HcpA, produced in different cells, show different molecular weights in Figure S12B

For the Western blot analyses, we used the purified CihC, HcpA, and ChiB proteins, respectively, as controls for demonstrating the reactivity of the respective antibodies. The two other lanes contain cell lysates obtained from *B. recurrentis* PAbJ cells cultivated in two different media *in vitro* (as already described in the figure legend). The slightly different molecular weight of the recombinant CihC protein might be explained by the His-tag fused to the N-terminus and used of protein purification. Regarding the double band of the purified HcpA protein, we again check the sequence of the vector to find any sequence difference between the PCR amplified and cloned gene fragment

and the WT gene but, expect for the His-tag, we cannot see any sequence variations which might account for the unusual migration in the SDS gel. AlphaFold prediction revealed that this molecule appears to be a globular protein which might explain why differences observed. Although highly speculative, post-translational modification in the WT *B. recurrentis* strain might also be an explanation. As these are speculations, we decided to not comment on the obvious differences shown between the purified and native proteins.

Lines 276-290. This section of the discussion includes overly long descriptions of the results. Please shorten these summaries and focus more on interpretation and implications.

As suggested by the reviewer(s) we shorten the paragraph and focus on the interpretation of our data. The paragraph now read as follows:

Line 308-317: “In In this study, we identified a cluster of five genes (**Fig. S1 and S2**) encoding for proteins that display anti-complement and anti-opsonic properties (**Figs. 5, 6, and Table S4**). The organization and architecture of this gene cluster are highly conserved among *B. recurrentis* strains isolated between 1994 and 2015 from clinically confirmed LBRF patients (**Fig. S1**)^{24,26} as well as ancient DNA⁴². The strong conservation over 20 years suggests that this gene cluster has been maintained by selective pressure. Also, the low genetic diversity of *B. recurrentis* might account for an extremely restricted pathogen-vector-host relationship. Comparative genomics identified multiple copies of Chi paralogous genes on the megaplasmid of *B. duttonii* and New World RFB (**Fig. S2**)^{25,43}. Due to their structural similarity to CspA orthologs of LD spirochetes, these encoding proteins were tentatively designated as “P35”-like proteins, Pfam54_60 proteins or CRASPs without further characterization^{25,44}.”

Line 302. ..”antibody responses to at least ChiB and ChiD have been detected in samples from LBRF”....Please provide more information helping the reader to identify the discussed literature data on ChiB and ChiD, concerning citation 40. For example, include in brackets the **corresponding ORF names** or other identifiers for ChiB and ChiD as described in reference 40.

We thank the reviewer(s) to point that out and added more information on the corresponding ORF proteins in the main text for more transparency as follows:

Line 329-331: “Of note, antibody responses to at least ChiB (provisionally termed ORF7) and ChiD (provisionally termed ORF9) have been detected in samples from LBRF patients indicating that certain Chi proteins were produced during infection⁴⁷.”

Line 356 – Was the serine protease activity of the Chi proteins was tested e.g. with another substrates eg. available commercially and based on fluorophores, to investigate if they are active ?

Our data showed that none of the Chi proteins displayed proteolytic activity against C3. Therefore, we did not investigate whether these proteins could cleave other substrates, as such activity was not indicated and is not claimed in this study. The potential enzymatic role of ChiE was inferred only from the structural arrangement of residues resembling a catalytic triad, without evidence of protease activity. It remains possible that ChiE may show protease-like activity only under physiological conditions, for example upon interaction with host proteins or membranes during infection.

Line 370 – This notion is not fully supported by the results from the Figure S4.

We agree with the reviewers assessment and have now clarify this point by rephrasing the sentences accordingly.

Line 403-405: “Mechanistically, the interaction of plasminogen with bacterial ligands is often mediated by lysines as also observed for all Chi proteins (**Fig. 7A and S11**).”

Minor comments:

Line 29, 66: 2.71A is medium resolution structure, whereas 1.5A is indeed a high resolution structure, maybe it would be better to write “X-ray structures of ChiA and ChiB”

That is correct, and we have revised the text accordingly (lines 29 and 90).

Line 58: CihC and HcpA, have been described so far in *B. recurrentis* as complement targeting proteins that bind C1-Inh, C4BP, and FH, respectively – what exactly bind what?

As mentioned above, we now add some sentence into the introduction to clarify this point as follows:

Line 68-71: “CihC and HcpA, both of which exhibit anti-complement properties and potentially contributing to the pathogenesis of *B. recurrentis* protect spirochetes from complement-mediated killing by binding of C1-Inh, C4BP (via CihC), and FH (via HcpA) to terminate CP and AP activation^{15,16}.”

Line 333: probably “as” not “than”, the whole sentence in the line 333 is also hard to understand

To clarify this point, we slightly rephrase the respective sentence as follows:

Line 361-363: “ChiE displaying inhibitory activity on all three pathways clearly protects susceptible spirochetes from complement-mediated killing comparable as ChiB, ChiC, and ChiD, however, the ChiE-producing strain was killed by human serum (**Fig. 8B**).”

Line 62: there is “chiC gene” which is probably a mistake as the gene corresponds to CihC protein , the same mistake is in the 263 line

We thank the reviewer(s) for pointing that out. We corrected the respective parts in the revised version as follows:

Line 85-86: “Here, we identified a unique gene cluster on megaplasmid lp190 adjacent to the tandemly arranged *chiC* and *hcpA* genes^{24,25}.”

and

Line 293-294: “Moreover, expression of all *chi* genes, as well as *cihC*, *hcpA*, and *flaB* could be demonstrated indicating that the entire *chi* gene cluster was expressed *in vitro* (**Fig. S14A**).”

Line 24-25: Sentence “Borrelia proteins protect susceptible spirochetes from complement-mediated killing and three molecules facilitate serum resistance” should be edited. The authors should name the proteins and molecules.

Edited as suggested: line 28-29: “*Borrelia* proteins protect susceptible spirochetes from complement-mediated killing and ChiB, ChiC, and ChiD facilitate serum resistance.”

Figure S10. The Figure legend needs correction.

The legend of the new **Figure S13** was corrected as follows:

“Gene expression analyses of Chi-encoded genes in gain-of-function strains. Expression of *chi* homologues genes in *in vitro* cultivated gain-of-function strains was determined by quantitative real-time PCR analyses. Total RNA was isolated from *in vitro* gain-of-function strains grown at 33 °C and transcribed to cDNA. Differences were calculated by comparing the C_T values with those obtained from transcribed cDNA of the serum-sensitive *B. garinii* G1 strain carrying the empty shuttle vector G1/pKFSS1. Data represent means of three independent experiments and differences were calculated by the $2^{-\Delta\Delta CT}$ method. An unpaired student t-test with a confidence interval of 95% was used to calculate the statistical significance. ***, $p \leq 0.001$; ****, $p \leq 0.0001$, n.s., no statistical significance.”

Reviewer #3 (Remarks to the Author):

I co-reviewed this manuscript with one of the reviewers who provided the listed reports. This is part of the *Nature Communications* initiative to facilitate training in peer review and to provide appropriate recognition for Early Career Researchers who co-review manuscripts.

We thank Reviewer #3 for their contribution to the review process.

Please note that the figures have been renumbered as follows:

Original submission	Revision
Fig. 4	Fig. 1 X-ray structures
Fig. 5	Fig. 2: Structural Comparison
Fig. 6	Fig. 3 Putative redox regulated sites
Fig. 7	Fig. 4: Structural comparison
Fig. 1	Fig. 5: Assessment of inhibitory capacity
Fig. 2	Fig. 6: Chi proteins TP inhibition & C9 polymerisation
Fig. 3	Fig. 7: Binding to plasminogen
Fig. 8	Fig. 8: Protection of Chi proteins
Fig. 9	Fig. 9: Schematic representation
Fig. S1	Fig. S1: Complement system (new)
Fig. S5	Fig. S2: Sequence analyses
Fig. S6	Fig. S3: Sequence comparison B. duttonii (new)
Fig. S7	Fig. S4: Sequence alignment and structure features
Fig. S9	Fig. S5: Surface representation
Fig. S2	Fig. S6: Overlay X-ray
Fig. S3	Fig. S7: Structure comparison of ChiB
Fig. S4	Fig. S8: Binding of complement to Chi proteins
Fig. S8	Fig. S9: Dose dependance assays
Fig. S10	Fig. S10: Proteolytic activity of Chi proteins
Fig. S11	Fig. S11: PLG activation assay
Fig. S12	Fig. S12: ChiB variants
	Fig. S13: Gene expression analyses
	Fig. S14: Gene expression & Western blot analyses of WT B. recurrentis

Reviewer comments are shown in black (Reviewers 2 and 3) and blue (Reviewer 4). Our responses are provided in magenta.

Reviewer #2 (Remarks to the Author):

1. The introduction has been significantly improved. The rest of the manuscript has been somewhat improved, but is still difficult to read, as mentioned in the first version of this review.

According to the reviewer's suggestion, we have re-structured the results, discussion as well as the materials and methods in the same sequence in which the data were presented in the results to make the whole text more reader-friendly. We hope that our revised manuscript now meets the reviewer's approval.

2. Panels A–C of Figure 1 should be moved to the Supplementary Data, as they duplicate content already shown in panels D–F. This recommendation was also made in the first review.

As suggested by this reviewer, panels A-C were moved to the supplements (now Figure S8) to avoid presentation of duplicate data in a single figure.

3. Figure 6 was not improved (the format of concentration units in the Figure was not unified).

We apologize for the confusion but the original gels incorporated in the Data Source File already contained the unified concentrations. Unfortunately, we did not upload the modified Figure 6 along with the revised manuscript. The modified figure has now been uploaded.

4. Supplementary Figure 8 was not improved and shows inconsistent data.

Supplementary Figure 8 (now Supplementary Figure 9) has now been improved to reflect the data presented in the Results section.

The controls are not clearly described. For example, panel S.Fig.8.C – BtcA is marked as significant. The authors write that “At the time when these assays were conducted, no control protein of bacterial origin was available that interact with FB, C1q, C4, and C4b, respectively”. Perhaps better controls are now available.

To improve the data presented in Supplementary figure S8, binding analyses of ChiE with C1q, C4, and C4b were repeated by including ChiC from *B. recurrentis* for C1q binding and CbiA from *Borrelia miyamotoi* as a control for C4 and C4b binding (Röttgerding et al., Sci Rep. 2017;7:303. doi: 10.1038/s41598-017-00412-4.) Regarding FB, however, no appropriate control was currently available. All data generated were incorporated into the new Supplementary figure S9.

The results presented in S. Figure 8 are main basis of Figure 9 which summarizes the proposed role of the studied proteins. In the light of the above, Figure 9 is not very credible.

As suggested by the reviewers, we assure that the binding profiles of each Chi protein summarized in Figure 9 and Table S4 correctly reflect the data of the complement binding analyses shown in Supplementary figure S9.

5. Supplementary Figure 9 was not improved. Supplementary Figure 9 G – data on ChiD binding to C5 are still inconsistent with the Supplementary Figure S8. The authors respond that they have no explanation for this observation. In my opinion, such inexplicable and inconsistent results should not be published. Both the tested proteins and BSA appear to bind to complement components nonspecifically.

Additional ELISAs were performed to analyse binding of C5 to Chi proteins confirming that only ChiB significantly bound C5. CbiA from *B. miyamotoi* was used as a positive control. The new data are shown in Supplementary Figure 10 (formerly Supplementary figure 9). Based on the new data, the dose-response assays with ChiD and ChiE were deleted from Supplementary figure 10 to ensure consistency of the figures as suggested.

The text was modified as follows:

Line 218 to 224: „To elucidate the molecular mechanism(s) of complement inhibition, interaction of Chi proteins with selected complement components of the AP [C3b, factor B (FB)] and the CP (C1q, C1r, C4, and C4b) as well as C5, FH, and FI was investigated. As demonstrated in **Fig. S9A**, all Chi proteins bound C3b but lower absorbance values were obtained employing ChiA. When binding to FB was assayed, all Chi homologs exhibited very low absorbance values (~0.25), even if statistically significant for ChiB and ChiE (**Fig. S9B**). In comparison to the C5-binding CbiA protein from *B. miyamotoi*¹² used as a positive control, only ChiB appears to bind C5 to some extent.”

Line 370 to 372: „Binding of Chi proteins to FB, the catalytic subunit of the AP C3 convertase, appears to be highly unlikely due to the very low absorbance values measured (**Fig. S9B**). Hence, the most effective mechanism of AP inhibition mediated by Chi’s targets C3b generation.”

The results presented in Supplementary Figure S9 also form the main basis of Figure 9, which summarizes the proposed role of the tested proteins. In the light of the above, Figure 9 is not very credible.

Figure 9 and Table S4 were improved and now show the data collected from the complement binding assays.

6. Supplementary Figure S11 and measurements in replicas. Although an Excel table with the raw data was included, the standard deviations in this table are still missing.

We thank this reviewer for bringing this point to our attention. The standard deviations have now been incorporated into the revised Data Source File as suggested.

7. Supplementary Figure S11 – the authors did not respond the question “Why in the assays with ChiA and ChiC, the proteolytic activity in the presence of TXA is almost as high as in its absence?” The authors still claim that: “Our new generated data revealed that TXA strongly influenced binding of plasminogen to ChiA, ChiB, and ChiD, and to some extent also to ChiC, and ChiE indicating that lysines do play a role in binding of plasminogen” and “Proteolytic activity of plasminogen was impaired when ChiB, ChiD, and ChiE were assayed indicating that lysines are involved in binding to all five Chi proteins “. In the discussion: “Mechanistically, the interaction of plasminogen with bacterial ligands is often mediated by lysines as also observed for all Chi homologs (Fig. 7A and S11)”. The results confirming these claims are missing. The Supplementary Figure 11 shows something opposite. The difference in signal recorded with and without TXA for controls is clear while there is no difference in signal for ChiA, ChiB and really weak difference for ChiD, so lysines are not involved in binding.

We thank all reviewers for pointing that out and agree with the reviewers’ comments that Supplementary figure S11 (now Supplementary figure S13) does not reflect what has been stated in the text. We greatly apologize for the confusion this may cause. Also, as suggested by reviewer #4, we reevaluated the new data submitted in our first revision and modified the respective part in the text and the discussion as follows:

Line 279 to 281: “Proteolytic activity of plasminogen was impaired in the presence of tranexamic acid when ChiC and ChiE were assayed but no impact could be observed for ChiA, ChiB, and ChiD, respectively.”

and

Line 417 and 419: „Mechanistically, the interaction of plasminogen with bacterial ligands is often mediated by lysines as also observed for ChiC and ChiE but not for ChiA, ChiB, and ChiD (**Fig. 7A and S13**).”

Points 4-7 point out some really serious shortcomings of this manuscript.

As already mentioned, all points raised by reviewer 1 and 2 were carefully addressed to the best of our knowledge.

8. The authors still claim that: “The ChiB structure was solved in 2018, at a time when AlphaFold2 had not yet been released. The first AlphaFold2 release appeared at the end of 2020, and public access to the prediction server became available only in mid-2021. Because of the low sequence identity to known structures, the homology model used for molecular replacement was not sufficient to solve the structure, so we performed experimental phasing using anomalous measurements, followed by AutoSol and AutoBuild. We fully agree that an AlphaFold2-based approach would have greatly facilitated structure determination, and we would have gladly used this tool had it been available at the time.”

Although the authors clarified the rationale behind their structure determination approach, they did not address the reviewer’s suggestion to use the AlphaFold model for MR. Given that it is now 2025, the authors should perform molecular replacement on the 2018 dataset using AlphaFold-generated model to assess whether the experimental model differs from the predicted one and, if so, in what ways. They should also articulate the possible added value provided by the experimental model. Readers are not aware that the structure was originally determined in 2018, and this inconsistency in the structural strategy is immediately noticeable. Moreover, refining predicted models using experimental data is now an established practice.

We thank the reviewer for this constructive comment. A comparison between the ChiA and ChiB crystal structures and their corresponding AlphaFold2 (AF2) models has been included since the initial submission (Fig. S8 and structural comparison section). The close agreement between the experimental structures and the AF2-predicted core folds supports the accuracy of both approaches.

To further illustrate the reliability of the AF2 models, we added a new figure showing per-residue confidence scores (pLDDT). Across most regions, pLDDT values exceed 90 and decrease only at the $\alpha 7$ – $\alpha 8$ junction and in the N-terminal regions (Fig. S6b), consistent with the increased flexibility observed in the crystal structures.

Notably, the AF2 model of ChiE, for which no experimental structure is currently available, suggested potential redox sensitivity, which we subsequently confirmed experimentally using Ellman assays. Together, the strong structural agreement with the experimental models and the high prediction confidence support the use of AF2 models for structural analysis of the remaining homologs. At the same time, our data illustrate that experimentally determined structures remain essential when mechanistically relevant biochemical features—such as ligand binding or redox-active cysteines—are involved.

While we carefully considered repeating molecular replacement using AF2-derived search models, we note that once refinement has converged, the final crystallographic model is defined by its agreement with the experimental diffraction data, as reflected by R_{work} and R_{free} . The ChiA and ChiB structures, with R_{free} values of approximately 30% and 20%, respectively, represent well-defined solutions that are not dependent on the initial search model.

The limitations observed for the ChiA structure are consistent with intrinsic crystal flexibility rather than the choice of starting model (see response to point 9 below). Importantly, several functionally relevant features—such as the hydrophobic pocket in ChiB and the disulfide bond linking ChiA monomers—are directly resolved in the crystallographic data and would not be accessible from prediction alone.

9. The authors still claim that: “We agree with the reviewer that access to both the MTZ files and the PDB files is important for evaluating the quality of the reported structures, especially when the public release is on hold until publication. The ChiA and ChiB structures have been deposited in the Protein Data Bank under accession codes 9FSD and 9FQQ, respectively, including the corresponding MTZ files. The entries are currently on hold due to ongoing competitive work in the Borrelia field.

We would have no objection to making these files available to the reviewers. However, the validation reports generated by the PDB are generally sufficient to assess the quality of the deposited structures.

In the ChiB structure (1.5 Å), only a few RSRZ outliers (18 residues out of 284) were detected. These are mainly located in surface regions with weak side-chain density, particularly in the N-terminal helix, which is consistent with the higher local B-factors.

The ChiA structure (2.7 Å) crystallized in the monoclinic space group C2 with two monomers in the asymmetric unit connected by a disulfide bond.

As ChiA is rapidly oxidized, crystallization was challenging; well-diffracting crystals were obtained only after seeding and in the absence of reducing agents. Minor flexibility between the monomers likely contributed to the moderate resolution. Space-group determination was initially ambiguous but confirmed unambiguously during final refinement. The data were processed with XDS, and quality was assessed with xtriage (Phenix). The relatively high Wilson B-factor (63.5; Supplementary Table 2) indicates increased overall flexibility, consistent with the higher number of RSRZ outliers (97 residues out of 530). These residues are mainly located in the N-terminal helix, the S-domain, and the connecting loops between helices. Nevertheless, the overall model is well defined, supported by an R_{free} of 30% and its close agreement with the AlphaFold2 prediction.”

I thought that this journal requires all raw data discussed in the manuscript to be available for review. The validation report for the ChiA structure indicates a model of suboptimal quality in terms of side-chain geometry and bond angles. I understand that the data are not ideal and that the structure displays substantial flexibility, but under such circumstances tightening the geometric restraints appears to be a reasonable approach to improve the model. Moreover such a high overall B-factor of the structure may just as well indicate an error in data processing or refinement. The validation report itself rather suggests that provided structure needs significant improvement.

We appreciate the reviewer’s careful evaluation of the ChiA structure and the corresponding validation report. The increased Wilson B-factor of ChiA (63.5 Å²) reflects intrinsic crystal flexibility

and is an experimentally determined property of the dataset. It is fully consistent with the average B-factor of the refined model (72 Å²) and is independent of refinement strategies. The RSRZ outliers are confined to highly mobile surface and loop regions. In such cases, residues can either be omitted or retained with high B-factors; we chose the latter to preserve amino-acid identity in regions with weak electron density.

As standard practice in crystallography, we first checked for possible errors in data processing using xtriage (Phenix) and XDS/XSCALE before initiating structure solution. No indications of data processing artefacts were detected. During refinement, we also tested several alternative refinement strategies, including stricter geometric restraints; however, these did not improve the global statistics and instead worsened the fit to the electron density. Refinement converged appropriately, as reflected in the R_{free} of 30%.

We would like to clarify that the crystallographic entries were re-deposited, resulting in new accession codes (28LI and 28LK). The atomic coordinates and structure factors were not modified. The entries are currently on hold and will be released upon publication. The manuscript and Data Availability statement have been updated accordingly.

Reviewer #3 (Remarks to the Author):

Reviewer #4 (Remarks to the Author):

We appreciate the time and effort reviewer #4 has dedicated to the revised manuscript and thank the reviewer for the valuable and positive comments. We have now incorporated all new data shown in the rebuttal to reviewer #1 into the new revised manuscript including the promoter analyses, and the degradation assays performed to assess surface localization of Chi proteins.

“1. There were no studies reported directly establishing the relevance of Chi proteins in protecting Br from being inactivated by compliment”

I agree with the authors on their rebuttal points related to the inherent difficulty in obtaining in vivo data due to numerous limitations in the vector biology, available animal model systems, and current tractability of genetic modifications of *B. recurrentis*. They have now provided new data in S14 that genes of interest are expressed, and proteins are produced, in wild-type *B. recurrentis* strains, although they were unable to identify in vitro growth conditions that regulate their production. ***The authors have appropriately addressed this concern.***

Note: The authors need to clarify the strain of *B. recurrentis* being evaluated in S14 as the rebuttal text and figure labels seem to indicate this was strain A17, but the legend indicates strain PABJ. If the image in the rebuttal represents an independent set of experiments on the other strain, it should be included in S14.

The strain used for the analyses has now been specified in Supplementary figure S15 as suggested

“2. Do all 5 Chi proteins regulated by different promoters responds to external signals in a similar fashion or is there a common signal that drive all of them or specifically one of the 5 Chi proteins?”

The authors provide extensive bioinformatic analyses in the rebuttal to address this question, however, it is not clear that this analysis was incorporated into the manuscript. The authors should include this information as a supplementary figure and discuss these results in the manuscript.

Assuming this is done, the authors will have appropriately addressed this reviewer question.

As suggested, the bioinformatic analyses of the chi promoters shown in our rebuttal to reviewer #1 have now been added to the manuscript and presented in Supplementary figure S16. The following text has been incorporated to the revised manuscript:

Line 494 to 500: „To recognize canonical promoter motifs within the *chi* gene cluster, the YAPP Eukaryotic Core Promoter Predictor (<https://www.bioinformatics.org>) and Promotech (R. Chevez-Guardado & L. Peña-Castillo, 2021, <https://doi.org/10.1186/s13059-021-02514-9>) were used. The YAPP tool is created for TATA boxes, initiator elements (INR), downstream core element (DPE) in upstream eukaryotic but also for prokaryotic promoter sequences. YAPP algorithm calculates matrix similarity score for matches with consensus sequences to qualify as promoter elements. True positives predictions display score values that tend to be around 0.5 or higher when using Promotech, a machine-learning-based method.“

Line 303 to 305: „Furthermore, sequence analyses identified canonical promoter elements in the upstream regions of each *chi* gene suggesting that these genes, in principle, are expressed *in vitro* (Fig. S16).“

“Since there is a sequence correlation to Pfam54 -60 paralogous gene family which are known to encode for genes that are differentially expressed in LD spirochetes, would there be a significant difference in the levels of expression of Chi proteins in Br.”

The authors have appropriately addressed the reviewer question. However, while the DALI-based structural comparisons being referenced here are performed and described in the work, the actual sequence relationship of Chi proteins (if any) to proteins encoded by tickborne relapsing fever spirochetes or Lyme disease associated borrelia is not made clear enough. If relevant, any phylogenetic relationship to proteins from TBRF or LD should be shown in Fig. S2, or S3 (or a new figure) and these relationships or lack thereof should be commented on in the manuscript. **Assuming this is done, the authors will have addressed this related concern.**

We thank this reviewer for his/her important remark. By conducting BlastP-based sequences comparisons between Chi proteins of *B. recurrentis* A17 and Pfam54 -60 proteins BBE31, BBA64, BBA66, BBA68 (CspA), BBA69, and BBA73 of *Borrelia burgdorferi* B31, no significant similarity for ChiA, ChiB, ChiD and ChiE could be found. Except BBA64 showed a rather weak identity (21% identity, 96% coverage) to ChiC. Thus, we decided to not include these additional bioinformatic analyses into the revised manuscript but commented on into the results as follows:

Line 191 to 192:“By using BlastP, these Pfam54_60 paralogous proteins did not show significant similarities to the five Chi proteins,,.

“Are the Chi proteins expressed on the surface of the Br and if so does enzymatic removal of surface exposed proteins enhance complement sensitivity of Br strains?”

The authors have appropriately addressed the reviewer question; however, it is confusing why this new data was seemingly not included in the revised manuscript. The data shown in the rebuttal

(proteinase K gels and complement deposition microscopy) should be included in the manuscript as supplementary data and commented on in the manuscript. **Assuming this is done, the reviewers will have fully addressed this concern**

As suggested, the data of the protease K degradation assay shown in our rebuttal to reviewer #1 has now been added to the manuscript and presented in Supplementary figure S17 and S18. The following text has been incorporated into the Discussion and the Materials and Methods:

Line 402 to 405 (Discussion): „Nevertheless, enzymatic removal of surface-exposed proteins including Chi proteins significantly enhances complement sensitivity of *B. recurrentis* as confirmed by an increase of deposited components on the *Borrelia* surface (Figs. S17 and S18).”

Line 816 to 835 (Materials and Methods):

„In situ proteinase K treatment and immunofluorescence microscopy

To obtain complement-susceptible *B. recurrentis* A17 cells, a protease accessibility assay was performed. Spirochetes (6×10^6 cells) were incubated with or without proteinase K (200 µg/ml) for 40 min at RT and the proteolytic activity was then terminated by adding Pefabloc® SC (3 mM). Following sedimentation, cells were carefully washed twice with PBS and resuspended in 100 µl PBS containing 1% BSA (PBSA). Proteinase K-treated spirochetes were either lysated by sonication for Western blot analysis or incubated for 30 min at 33 °C with either 50 µl NHS or 50 µl hiNHS. After sedimentation, spirochetes were diluted 1:20 in PBSA and aliquots of 12 µl were spotted on diagnostic slides (Waldemar Knittel Glasbearbeitungs GmbH, Braunschweig, Germany). Slides were allowed to air-dried overnight and thereafter incubated for 10 min at RT with 40% glyoxal solution. After fixation, slides were incubated for 1 h at 33 °C in a humidified chamber with either a polyclonal anti-C3 antibody (dilution of 1:1,000) or a monoclonal C5b-9 antibody (dilution of 1:50), respectively. Following four washes with PBS, the slides were incubated for 1 h at 33 °C with 1:2,000 dilutions of Alexa 488-conjugated secondary antibodies (Life Technologies, Carlsbad, CA, USA). To stain *Borrelia* DNA, slides were washed four times with PBS and incubated with 40 µl of a DAPI solution (2 µg/ml) for 10 min at 4 °C. After mounting with fluorescence mounting medium (Dako), complement components deposited on the spirochetal surface were visualized by using an Axio Imager M2 fluorescence microscope (Zeiss, Oberkochen, Germany) equipped with a Spot RT3 camera (Visitron Systems, Puchheim, Germany). Images were generated by using an objective lens (Zeiss Plan-Apochromat) with a 63x magnification and for digitalisation the Visiview® software (Visitron Systems GmbH, Puchheim, Germany). “

“5. What are the unique structural features of ChiC and ChiE that are different from those spirochetal proteins that interact with complement and how do these proteins alter interaction of Br with the host (other than complement and plasminogen binding). These aspects of the study can be expanded in the discussion part of the manuscript to focus the structural differences which is the mainstay of this study.”

The authors have appropriately addressed the reviewer question, however, on line 118: The definition of an “S-domain” should be given.

The paragraph has been revised and now reads (lines 120 to 122): “ChiB (31.8 kDa, 284 residues) consists of eight α-helices and one β-hairpin (P97–L107) forming a compact main domain (α2–α8) and a surface-exposed domain that we refer to as the S-domain. The latter comprises the β-hairpin and the N-terminal portion of helix α4 (A106–K122) (Fig. 1A)”.

“8. The authors still claim that: “The ChiB structure was solved in 2018, at a time when AlphaFold2

had not yet been released. The first AlphaFold2 release appeared at the end of 2020, and public access to the prediction server became available only in mid-2021. Because of the low sequence identity to known structures, the homology model used for molecular replacement was not sufficient to solve the structure, so we performed experimental phasing using anomalous measurements, followed by AutoSol and AutoBuild. We fully agree that an AlphaFold2-based approach would have greatly facilitated structure determination, and we would have gladly used this tool had it been available at the time.

Although the authors clarified the rationale behind their structure determination approach, they did not address the reviewer's suggestion to use the AlphaFold model for MR. Given that it is now 2025, the authors should perform molecular replacement on the 2018 dataset using AlphaFold-generated model to assess whether the experimental model differs from the predicted one and, if so, in what ways. They should also articulate the possible added value provided by the experimental model.

Readers are not aware that the structure was originally determined in 2018, and this inconsistency in the structural strategy is immediately noticeable. Moreover, refining predicted models using experimental data is now an established practice.”

While it should have been trivial for the authors to comply with the reviewer suggestion to solve the ChiB structure with the AF3 model by MR, I can see no reason to do this. The structure in question has an Rfree of ~21% and phasing was obtained using direct methods, still a gold standard. *In my view, there are no new experiments needed to address this concern, however, the evaluation of the AF models that is prompted by this reviewer question needs to be addressed as there is currently no way to assess the per-residue quality of any of models (most importantly for ChiC, D, and E for which no experimental structure was obtained). This can be done easily by inclusion of pLDDT values across ALL AlphaFold models presented in the study in Fig. S6 or a new supplementary figure, either as a per-residue plot or using the structures and a coloring scheme such as AlphaFolds standard pLDDT coloring.*

We appreciate the reviewer's suggestion to further evaluate the AlphaFold models. we have now extended this assessment by including per-residue confidence information for all five Chi homologs. Specifically, predicted local distance difference test (pLDDT) values are visualized using an AlphaFold-based color scheme in Fig. S6b. This addition allows assessment of per-residue confidence, particularly for ChiC, ChiD, and ChiE, for which no experimental structures are available.

We have added the following to the manuscript (line 141 to 143):”Per-residue confidence (pLDDT) exceeds 90 across most regions of the models and is reduced only at the $\alpha 7$ – $\alpha 8$ junction and N-terminal regions (Fig. S6b). Together, the strong structural agreement and high model confidence support the use of AF2 models for structural analysis of the remaining homologs.”

Minor NEW Comments that should be addressed:

The ChiB structure is repeatedly referred to in the manuscript as 1.6 Å limiting resolution, however Table S2 indicates the cut-off at 1.5 Å.

We thank this reviewer for bringing this to our attention. The main text now contains the corrected resolution of 1.5 Å as already mentioned in Table S2.

Line 542 – ChiB is referred to as an enzyme, this should be changed to protein.

Changed as suggested

There is no information given about the origin or nature of the “phospholipid” in the ChiB structure. It does not appear it was added exogenously as part of the crystallization experiments. This should be clarified in the methods and details about how the phospholipid was modeled must be given.

We thank the reviewer for this important comment. At 1.5 Å resolution, the initial Fo–Fc difference map calculated prior to ligand placement revealed strong, continuous, and well-defined electron density that could not be assigned to protein residues or solvent molecules. Alternative models corresponding to buffer components present during purification and crystallization (Tris, citrate, glycerol, and DTT) were systematically tested but none adequately accounted for the size and shape of the observed electron density.

Specifically, the Fo–Fc difference density contained a prominent peak consistent with a phosphate group, along with clear density extending into two elongated hydrophobic chains. At this resolution, two distinct carbonyl-like features were visible at positions consistent with acyl groups. The overall geometry and volume of the density supported assignment as a phospholipid-like molecule.

The ligand was therefore modeled as LPP and refined using standard restrained refinement procedures. The final model showed excellent agreement with the 2Fo–Fc map and resulted in minimal residual Fo–Fc difference density in this region. B-factors of the ligand were comparable to surrounding protein residues, further supporting the assignment. As no lipid was introduced during purification or crystallization, the most plausible explanation is co-purification of a bacterial phospholipid during heterologous expression in *E. coli*. These details have now been clarified in the revised Methods section as follows:

(Lines 547 to): “In addition to several solvent molecules, the initial Fo–Fc electron density map revealed clear density that could not be satisfactorily explained by buffer components present during purification and crystallization (Tris, citrate, glycerol, DTT) and was most consistent with a phospholipid-like molecule. The density contained a prominent peak consistent with a phosphate group and extended into two elongated hydrophobic chains with additional density features compatible with carbonyl groups. Based on its size and geometry, the ligand was modeled as LPP and refined with standard stereochemical restraints. As no lipid was introduced during purification or crystallization, the most plausible explanation is co-purification of a bacterial phospholipid during heterologous expression in *E. coli*.”

Fig S1 (Add factor D to the schematic)

A modified Figure S1 has been generated and factor D was added as suggested

Line 62: terminate should be terminated.

Line 64: Corrected as suggested

Line 66: block should be blocks.

Line 68: Corrected as suggested

Lines 81 and 83, I think “thether” should say tether.

Line 83 and 85: Corrected as suggested

Line 367-370: Might the CP specific nature of ChiE reported here and the lack of specific antibody being used in the assay be another potential reason for the incongruence in the data? This should be commented on.

We thank the reviewer for this helpful comment. In the serum susceptibility assay, spirochete survival was directly assessed after serum exposure and did not rely on antibody-based detection. Expression of ChiE in the gain-of-function strain was confirmed at the transcript level (Fig. S14G).

We agree that the absence of a ChiE-specific antibody limits verification at the protein level and prevents a direct assessment of surface exposure or stability, which may contribute to the observed discrepancy between purified ChiE activity and the lack of serum resistance in the ChiE producing strain. This point has now been clarified in the revised Discussion section as follows:

(Line 399 to 401): “Thus, employment of a ChiE-specific antibody would help clarify surface exposure and protein stability, and may therefore contribute to the observed phenotype in the surrogate strain.”

Line 384: encounters should be encounter

Corrected as suggested

Figure and Table Renumbering Overview

Former figure numbers correspond to Revision 1. Figures that were renumbered, or rearranged in Revision 2 are shown in **bold** and newly introduced figures were labeled as “new”.

Revision 1

Fig. 1 – X-ray structures
Fig. 2 – Structural comparison
Fig. 3 – Putative redox-regulated sites
Fig. 4 – Structural comparison
Fig. 5 – Assessment of inhibitory capacity
Fig. 6 – Chi proteins TP inhibition & C9 polymerization
Fig. 7 – Binding to plasminogen
Fig. 8 – Protection of Chi proteins
Fig. 9 – Schematic representation
Fig. S1 – Complement system
Fig. S2 – Sequence analyses
Fig. S3 – Sequence comparison *B. duttonii*

Revision 2

Fig. 1 – X-ray structures
Fig. 2 – Structural comparison
Fig. 3 – Putative redox-regulated sites
Fig. 4 – Structural comparison
Fig. 5 – Assessment of inhibitory capacity (dose dependence; former D–F) (new)
Fig. 6 – Chi proteins TP inhibition & C9 polymerization
Fig. 7 – Binding to plasminogen
Fig. 8 – Protection of Chi proteins
Fig. 9 – Schematic representation
Fig. S1 – Complement system
Fig. S2 – Sequence analyses
Fig. S3 – Sequence comparison *B. duttonii*

Fig. S4 – Sequence alignment and structural features

Fig. S5 – Surface representation

Fig. S6 – Overlay X-ray structures

—

Fig. S7 – Structure comparison of ChiB

Fig. S8 – Binding of C to Chi proteins

Fig. S9 – Dose dependence

Fig. S10 – Proteolytic activity of Chi proteins

Fig. S11 – PLG activation assay

Fig. S12 – Chi variants

Fig. S13 – Gene expression analyses (GOF)

Fig. S14 – Gene expression & Western blot analyses (WT)

—

—

—

—

Table S1

Table S2

Table S3

Table S4

Table S5

Fig. S4 – Sequence alignment and structural features

Fig. S5 – Surface representation

Fig. S6A – Overlay X-ray with AF2 structures

Fig. S6B – pLDDT-colored AF2 models (new)

Fig. S7 – Structure comparison of ChiB

Fig. S8 – Assessment of inhibitory capacity (dose dependence; former A–C) (new)

Fig. S9 – Binding of C to Chi proteins

Fig. S10 – Dose dependence

Fig. S11 – Proteolytic activity of Chi proteins

Fig. S12 – Chi variants

Fig. S13 – PLG activation assay

Fig. S14 – Gene expression analyses (GOF)

Fig. S15 – Gene expression & Western blot analyses (WT)

Fig. S16 – Promoter analyses (new)

Fig. S17 – Proteinase K assay (new)

Fig. S18 –IFM, deposition of complement on *Borrelia* surface (new)

Table S1

Table S2

Table S3

Table S4

Table S5